# Efficient Open Set Single Image Test Time Adaptation of Vision Language Models

**Manogna Sreenivas**                                                     *manognas@iisc.ac.in*
*Indian Institute of Science, Bengaluru*

**Soma Biswas**                                                           *somabiswas@iisc.ac.in*
*Indian Institute of Science, Bengaluru*

**Reviewed on OpenReview:** *https://openreview.net/forum?id=72YVabBErN*

## Abstract

Adapting models to dynamic, real-world environments characterized by shifting data distributions and unseen test scenarios is a critical challenge in deep learning. In this paper, we consider a realistic and challenging Test-Time Adaptation setting, where a model must continuously adapt to test samples that arrive sequentially, one at a time, while distinguishing between known and unknown classes. Current Test-Time Adaptation methods operate under closed-set assumptions or batch processing, differing from the real-world open-set scenarios. We address this limitation by establishing a comprehensive benchmark for *Open-set Single-image Test-Time Adaptation using Vision-Language Models.* Furthermore, we propose ROSITA, a novel framework that leverages dynamically updated feature banks to identify reliable test samples and employs a contrastive learning objective to improve the separation between known and unknown classes. Our approach effectively adapts models to domain shifts for known classes while rejecting unfamiliar samples. Extensive experiments across diverse real-world benchmarks demonstrate that ROSITA sets a new state-of-the-art in open-set TTA, achieving both strong performance and computational efficiency for real-time deployment. The code is released at https://github.com/manogna-s/ROSITA.git.

## 1 Introduction

Over the past decade, deep learning has revolutionized computer vision tasks such as image classification, object detection, and segmentation (Deng et al., 2009; Ren et al., 2015; He et al., 2017). However, these advancements are predominantly realized on the assumption that the training and test data follow the same distribution. In contrast, the real world is dynamic and ever-changing, making such assumptions often untenable. Distribution gaps between training and test data manifest in diverse forms Hendrycks & Dietterich (2019), including domain shifts and semantic shifts. Domain shifts could emerge from variations in lighting, weather, camera specifications, or geographical locations between the train and test datasets. Semantic shifts occur when a model, initially trained on a specific set of classes, encounters previously unseen classes during testing. Hence, navigating deep learning models through these dynamic test environments is imperative.

Test-Time Adaptation (TTA) addresses this challenge by enabling models to adapt during inference without access to the training data (Wang et al., 2021; Schneider et al., 2020; Döbler et al., 2023). TTA is characterized by three defining features: (1) no access to source data during adaptation, (2) the absence of ground truth labels for test data, and (3) an online adaptation scenario where test samples are encountered sequentially and accessible only once. These constraints reflect the dynamic and streaming nature of real-world applications. Existing TTA methods have predominantly focused on closed-set scenarios, assuming all test data belongs to known classes, hence falling short in real-world settings, where models are exposed to unseen categories beyond their training distribution. For example, an autonomous driving system trained to recognize urban vehicles like *car, truck, motorcycle* may incorrectly classify a *bicycle* as a *motorcycle* when deployed in a rural setting. In such scenarios, the model must not only adapt to domain shifts within known categories but also

identify unfamiliar elements as *unknown* rather than incorrectly classifying them as part of the known set. This highlights the critical need for **Open-set adaptation** in TTA. In addition, most TTA methods assume access to batches of test samples (Wang et al., 2021; Sreenivas et al., 2024), which are processed collectively for adaptation. However, this assumption is often impractical in real-time scenarios, where test samples arrive sequentially, one at a time, necessitating efficient **Single image Test-Time Adaptation** methods.

Large-scale Vision-Language Models (VLMs), such as CLIP (Radford et al., 2021), have demonstrated exceptional generalization capabilities across diverse domains, making them promising candidates for TTA. Recent works like TPT (Shu et al., 2022) and PromptAlign (Samadh et al., 2023) have explored prompt-tuning based adaptation of VLMs at the level of individual test samples, achieving improved zero-shot performance. TDA (Karmanov et al., 2024) employs a Training-free Dynamic Adapter to adapt VLMs during test-time. However, these methods are restricted to closed-set scenarios and do not address the challenges of more realistic open-set scenarios. Conversely, open-set TTA methods (Li et al., 2023; Lee et al., 2023) focus on adapting vision-only backbones (CNNs) trained on specific domains but require batch-wise processing of test samples, making them unsuitable for scenarios where test samples arrive one at a time. The combined challenges of open-set recognition and single-image adaptation remain largely unexplored.

To bridge these gaps, we establish a benchmark for **Open-set Single-image Test-Time Adaptation (OSTTA)** using VLMs, addressing both open-set recognition and single-image adaptation. We refer to the classes of interest (say CIFAR-10) as *desired* classes and the rest as *undesired* classes (say CIFAR-100). To identify whether a test sample belongs to a desired or undesired class, we employ a Linear Discriminant Analysis (LDA) Fisher (1936); Li et al. (2023) based class identifier. Samples identified as belonging to the desired classes are then classified accordingly into one of the desired classes. We equip closed-set single image TTA methods with this class identifier to handle open-set scenarios.

We propose a novel framework termed **ROSITA** designed for OSTTA using VLMs. At the core of this framework is the **ReDUCe** loss, which effectively leverages **Re**liable samples to enhance the separability between **D**esired and **U**ndesired classes through a **C**ontrastiv**e** loss. Additionally, moving beyond existing prompt-tuning approaches, we analyze the optimal set of parameters for adapting VLMs during test-time and identify that adapting LayerNorm parameters offers a lightweight yet effective solution for continuous adaptation. ROSITA dynamically updates these LayerNorm parameters using the ReDUCe loss, enabling it to adapt in open-set environments by accurately identifying unseen classes as "unknown" while maintaining the performance of VLMs on known categories. Our contributions are summarized as follows:

- To the best of our knowledge, we are the first to explore the capability of VLMs in addressing the challenging and realistic problem of *Open-set Single-image Test-Time Adaptation* (OSTTA), establishing a comprehensive benchmark for this setting.

- Our framework, ROSITA, introduces the ReDUCe loss to enhance separability between desired and undesired class samples, enabling reliable recognition of desired samples under domain shifts while effectively rejecting unfamiliar ones saying *"I don't know"*.

- We conduct a systematic analysis of parameter selection for VLM adaptation during test-time and identify LayerNorm parameters as the optimal choice for lightweight, *continuous adaptation of VLMs*.

- We demonstrate the effectiveness of ROSITA through extensive experiments across diverse domain adaptation benchmarks, simulating real-world test environments with single-domain shifts, continuous and frequent domain shifts, and varying proportions of desired and undesired class samples.

## 2 Open-set Single-image Test-Time Adaptation

### 2.1 Problem Setup

**Test stream.** The model encounters a single test sample $x_t$ at time $t$, sampled from $\mathcal{D}_t = \mathcal{D}_d \cup \mathcal{D}_u$ comprising of: (i) Desired class samples: $\mathcal{D}_d = \{x_t; y_t \in C_d\}$, with domain shift and belonging to one of the $\mathcal{C}_d$ desired classes, for example, $C_d = \{car, bus, ..., motorcycle\}$; (ii) Undesired class samples: $\mathcal{D}_u = \{x_t; y_t \in C_u\}$, which have semantic shift (irrelevant classes) such that $C_d \cap C_u = \phi$.

**Goal.** Given a test sample $x_t$ arriving at time $t$, the goal is to first recognize if it belongs to a desired class or not, constituting a binary classification task. If $x_t$ is identified as a desired class sample, a subsequent $|C_d|$-way classification is performed. Else, the prediction is *"I don't know"*. In essence, the overall process can be viewed as a $|C_d| + 1$ way classification problem.

**OSTTA scenarios.** We simulate several test scenarios inspired by the real world to evaluate the effectiveness of our method. (1) *Single domain*: We extend the standard TTA scenario where the test samples come from an unseen domain $D_d$ (say *snow* corruption of CIFAR-10C) by incorporating undesired samples $D_u$ (say CIFAR-100C). (2) *Continuously changing domains*: Here, $D_t$ changes with time as $(D_d^1 \cup D_u) \rightarrow (D_d^2 \cup D_u) \ldots \rightarrow (D_d^n \cup D_u)$, where $D_d^i$ is the $i^{th}$ domain encountered. (3) *Frequently changing domains*: Here, we significantly reduce the number of samples per domain in continuous open-set TTA. The fewer the samples per domain, the more frequently the test domain changes, simulating very dynamic open-set test scenarios. (4) *Varying sample ratio:* The proportion of samples from $C_d$ and $C_u$ in the test stream is varied.

## 2.2 Benchmark for OSTTA using VLMs

Here, we describe our motivation for using VLMs for the OSTTA problem and further describe how we establish a benchmark for the same.

**CNNs vs VLMs for OSTTA.** Test-Time Adaptation (TTA) traditionally focuses on CNNs, which are vision-only backbones trained on specific datasets. The goal is to adapt these CNNs to mitigate performance degradation when encountering unseen environments such as noisy or weather-affected conditions. These models usually require specific retraining for each dataset or desired classes. On the other hand, Vision-Language Models (VLMs) like CLIP (Radford et al., 2021) are pretrained on diverse image-text pairs from the web. These models demonstrate strong zero-shot generalization capabilities across diverse domains without any specific retraining. This makes VLMs a promising candidate for TTA scenarios. However, defining unseen classes or domains in the context of VLMs is non-trivial due to their exposure to diverse visual data. Although CLIP performs well in zero-shot classification for clean datasets, its performance on corrupted or style-shifted datasets like ImageNet-C/R remains suboptimal (Shu et al., 2022), making TTA still relevant. Moreover, CLIP can only classify an image by making a choice from the given set of desired classes. It lacks the ability to explicitly say "*I don't know*" when presented with a sample that does not belong to the set of desired classes, highlighting the need for open-set recognition. Noting these differences and advantages of VLMs over CNNs, we ask these questions: *1) How well can VLMs perform in open-set scenarios? 2) Can they be effectively adapted in a continuous manner? 3) How do we equip VLMs to handle domain shifts within desired classes while accurately rejecting unfamiliar samples?* To address these research questions, we establish a new benchmark and propose a framework termed ROSITA using VLMs.

**Classification using VLMs.** We evaluate our approach using Vision-Language Models (VLMs) such as CLIP (Radford et al., 2021) and MaPLe (Khattak et al., 2023) as backbones. CLIP consists of a Vision ($\mathcal{F}_V$) and Text ($\mathcal{F}_T$) encoder trained via contrastive learning on image-text pairs. The MaPLe backbone extends CLIP by incorporating multimodal prompts, enhancing its adaptability for downstream tasks. Given a test image $x_t$ and a set of desired classes $C_d = \{c_1, c_2, \ldots, c_N\}$, we construct text-based classifiers using predefined text prompts. Each class name is prepended with the prompt "A photo of a", creating class-specific text inputs $\{\boldsymbol{p}_T, c_i\}$. These inputs are passed through the text encoder to generate text embeddings $\boldsymbol{t}_i = \mathcal{F}_T(\{\boldsymbol{p}_T; c_i\})$ for each $c_i \in C_d$. The resulting classifier consists of text embeddings $\{\boldsymbol{t}_1, \boldsymbol{t}_2, \ldots, \boldsymbol{t}_N\}$. Class prediction for a sample $x_t$ is made by identifying the text embedding $t_i$ with the highest similarity to the image feature $f_t$ extracted from the vision encoder.

**Baseline Methods.** To establish a strong benchmark, we adapt several existing TTA methods designed for closed-set settings to the open-set Single Image TTA scenario. These include ZSEval (Radford et al., 2021), TPT (Shu et al., 2022), PAlign (Samadh et al., 2023), TDA (Karmanov et al., 2024) and DPE (Zhang et al., 2024). Furthermore, we extend TPT and PAlign to support continuous model updates by adapting prompts, referring to these variants as TPT-C and PAlign-C, respectively. We also adapt open-set TTA approaches originally designed for CNNs, such as (K+1)PC (Li et al., 2023) and UniEnt (Gao et al., 2024), to work with VLMs. These methods are described in detail in Appendix B. For a fair comparison, all baseline methods are equipped with a simple and efficient class identification mechanism based on the LDA objective (Li et al., 2023) to handle the open-set nature of the test stream.

**Desired vs Undesired Class Identifier.** In an open-set TTA setting, it is crucial for the model to distinguish between samples belonging to desired classes ($C_d$) and undesired classes ($C_u$) and appropriately reject samples from $C_u$. This problem can be viewed as a binary classification problem between desired and undesired samples based on the score $s_t$ (Equation 1). To achieve this, we equip all baseline methods with a parameter-free classifier based on Linear Discriminant Analysis (LDA) (Fisher, 1936; Li et al., 2023). This classifier uses the similarity score $s_t$, defined as the maximum cosine similarity between the image embedding $f_t$ of the test sample $x_t$ and the text embeddings $t_k$ of the desired classes $C_d$:

$$s_t = \max_k \text{sim}(f_t, t_k) \tag{1}$$

Rather than relying on a fixed threshold, which can be challenging to define in a streaming TTA scenario, we dynamically determine an optimal threshold using a continuously updated score bank $\mathcal{S}$. This bank stores the most recent $|\mathcal{S}|$ similarity scores, capturing the evolving distribution of the test stream. The optimal threshold $\tau_t^*$ is computed using 1D LDA to minimize intra-class variance, as follows:

$$\tau_t^* = \arg\min_\tau \frac{1}{|\mathcal{S}_d|} \sum_{s \in \mathcal{S}_d} (s - \mu_d)^2 + \frac{1}{|\mathcal{S}_u|} \sum_{s \in \mathcal{S}_u} (s - \mu_u)^2 \tag{2}$$

where $\mathcal{S}_d = \{s_i | s_i > \tau, s_i \in S\}$ and $\mathcal{S}_u = \{s_i | s_i < \tau, s_i \in S\}$ represent the scores of samples classified as desired and undesired, respectively, and $\mu_d$ and $\mu_u$ are their means. Using this threshold, the test sample $x_t$ is classified as:

$$\tilde{y}_t = \begin{cases} \text{desired} & \text{if } s_t \geq \tau_t^* \\ \text{undesired} & \text{if } s_t < \tau_t^* \end{cases} \tag{3}$$

This simple yet effective approach equips the model to handle open-set scenarios by explicitly rejecting undesired samples, thereby ensuring robust performance during adaptation.

We equip all the above described baseline methods with this LDA objective based *Desired vs Undesired class identifier* for a fair comparison in the Open-set Single Image TTA setting. In Appendix C.4, we demonstrate the superior performance of this method compared to naive confidence thresholding. With this comprehensive benchmark established, we now describe our proposed framework, ROSITA, which sets a new standard for Open-set single-image TTA.

## 3 Proposed ROSITA Framework

Given a single test sample $x_t$ at time $t$, to ensure effective TTA, we first characterize the sample based on the class and the quality of samples: (1) *Desired vs. Undesired Classes:* These refer to the ground truth class groups, where desired classes belong to $C_d$ and undesired classes belong to $C_u$. (2)*Reliable vs. Unreliable Samples:* A test sample $x_t$ is considered reliable if its score $s_t$ confidently places it in either the desired or undesired class distributions, estimated using LDA statistics ($\mu_d, \mu_u$). We leverage **Re**liable samples to differentiate **D**esired vs **U**ndesired class samples through a **C**ontrastiv**e** (**ReDUCe**) Loss for Open-set Single-image Test-time Adaptation (Figure 1). For this, we categorize the test samples as follows:

1. Reliable Desired Class Sample          :   $\tau_t^* < \mu_d < s_t$.
2. Unreliable Desired Class Sample       :   $\tau_t^* \leq s_t \leq \mu_d$.
3. Reliable Undesired Class Sample       :   $s_t < \mu_u < \tau_t^*$.
4. Unreliable Undesired Class Sample    :   $\mu_u \leq s_t \leq \tau_t^*$.

**ReDUCe Loss.** A contrastive objective typically needs positive and negative features, the goal being to maximize the similarity between a sample and its positives (could be augmentations (Chen et al., 2020) or nearest neighbours (Dwibedi et al., 2021)), while minimizing its similarity with the negatives. Such objectives (Chen et al., 2020; He et al., 2020; Khosla et al., 2020; Dwibedi et al., 2021) have been extensively used to learn good image representations in a self-supervised manner. While self-supervised learning assumes access to abundant data in an offline manner, giving the freedom to carefully choose positives and negatives, this

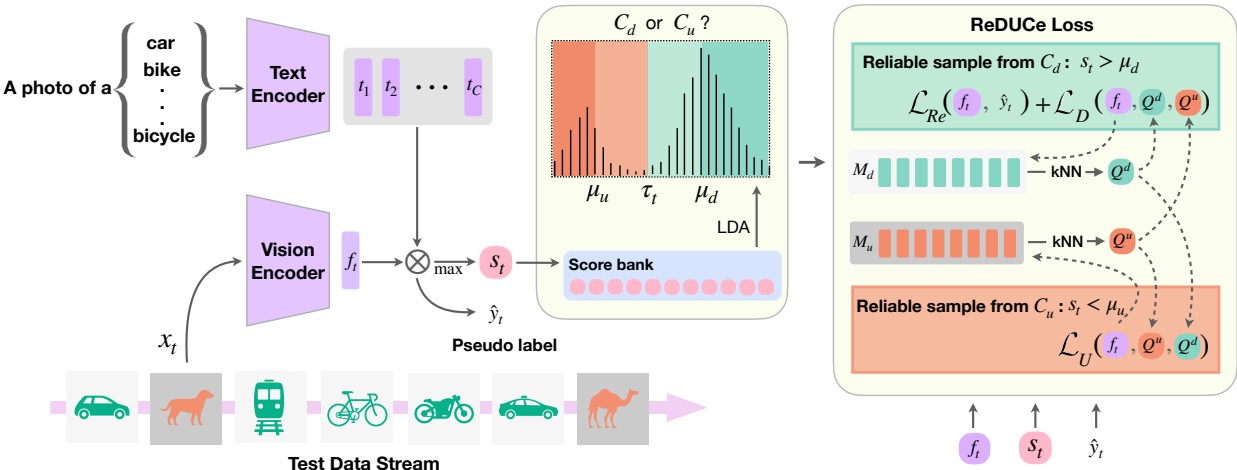

Figure 1: **ROSITA framework:** The test stream with samples from $C_d$ and $C_u$ arrive one at a time. An input image $x_t$ is recognized as a sample from $C_d$ and $C_u$ through an LDA based class identifier. Further, if a test sample is reliable, the respective feature banks are updated and the proposed ReDUCe loss is optimized to update the LayerNorm parameters of the Vision Encoder.

problem is set in an online scenario. Here, the test samples arrive one at a time and are accessible only at that instant. This challenging setting makes it non-trivial to use objectives like (Dwibedi et al., 2021). To circumvent this issue of lack of abundant test data, we propose to store two dynamically updated feature banks $\mathcal{M}_d$ and $\mathcal{M}_u$ of sizes $N_d$ and $N_u$, to store the features of reliable samples from $C_d$ and $C_u$ respectively. We propose ReDUCe loss to contrast a reliable sample from $C_d$ by choosing its positives and negatives as the $K$ nearest neighbours from $\mathcal{M}_d$ and $\mathcal{M}_u$ respectively and vice versa for a reliable sample from $C_u$. The buffer size for $\mathcal{M}_d$ is set as $|C_d| \times K$, where $|C_d|$ is the number of desired classes and $K$ is the number of neighbours retrieved. The feature banks $\mathcal{M}_d$ or $\mathcal{M}_u$ are updated with a feature $f_t$ if it is detected as a reliable sample from $C_d$ or $C_u$.

We fetch the $K$ nearest neighbours of a reliable test sample $x_t$ from each feature bank as follows.

$$Q_d = \text{kNN}(f_t; \mathcal{M}_d); \quad Q_u = \text{kNN}(f_t; \mathcal{M}_u) \tag{4}$$

**Case 1: Reliable sample from $C_d$.** If a test sample is identified as a reliable sample from $C_d$, we use a reliable pseudo-label loss on the sample $x_t$ and its augmentation $\tilde{x}_t$ as follows:

$$\mathcal{L}_{Re} = \mathcal{L}_{CE}(x_t, \hat{y}_t) + \mathcal{L}_{CE}(\tilde{x}_t, \hat{y}_t); \quad \hat{y}_t = \text{argmax}_i \, \text{sim}(f_t, t_i) \tag{5}$$

where sim represents cosine similarity. Further, we also propose to use a contrastive objective to enhance the clustering of desired class samples while pushing them apart from the undesired class samples.

As we aim to correctly classify the desired class samples, we select positives $z^+$ from $Q_d$ if its prediction $y^+$ matches with $\hat{y}_t$. The features $Q_u$ consisting of its kNN from $M_u$ act as its negatives. The following is the ReDUCe loss for a reliable sample from $C_d$:

$$\mathcal{L}_D = -\frac{1}{K^+} \sum_{z^+ \in Q_d} \mathbf{1}(y^+ = \hat{y}_t) \log \frac{\exp\left(\text{sim}\left(f_t, z^+\right)/\tau\right)}{\sum_{z^- \in Q_u} \exp(\text{sim}(f_t, z^-)/\tau)} \tag{6}$$

where $K^+ = \sum_{z^+ \in Q_d} \mathbf{1}(y^+ = \hat{y}_t)$, is the number of neighbours positively matched with $\hat{y}_t$.

**Case 2: Reliable sample from $C_u$.** If a test sample is identified as a reliable sample from $C_u$, we use the following contrastive objective by selecting positives $z^+$ from $Q_u$ and negatives $z^-$ from $Q_d$:

$$\mathcal{L}_U = -\frac{1}{K} \sum_{z^+ \in Q_u} \log \frac{\exp\left(\text{sim}\left(f_t, z^+\right)/\tau\right)}{\sum_{z^- \in Q_d} \exp(\text{sim}(f_t, z^-)/\tau)} \tag{7}$$

Table 1: Results with ImageNet-C/R as desired class data $D_d$, MNIST and SVHN for $D_u$.

| Method | IN-C/MNIST | | | IN-C/SVHN | | | IN-R/MNIST | | | IN-R/SVHN | | |
|---|---|---|---|---|---|---|---|---|---|---|---|---|
| | AUC ↑ | FPR ↓ | HM ↑ | AUC ↑ | FPR ↓ | HM ↑ | AUC ↑ | FPR ↓ | HM ↑ | AUC ↑ | FPR ↓ | HM ↑ |
| **CLIP** | | | | | | | | | | | | |
| ZS-Eval | 93.39 | 55.52 | 41.43 | 85.89 | 72.91 | 40.83 | 91.27 | 91.09 | 71.50 | 90.43 | 75.04 | 71.66 |
| TPT | 93.12 | 58.01 | 42.21 | 85.43 | 74.47 | 40.95 | 91.25 | 91.23 | 71.98 | 90.43 | 74.98 | 72.36 |
| TPT-C | 56.57 | 99.12 | 6.19 | 11.38 | 100.00 | 7.24 | 82.81 | 85.79 | 68.25 | 80.94 | 80.03 | 69.18 |
| (K+1) PC | 95.76 | 10.43 | 42.95 | 87.75 | 26.23 | 38.50 | 97.46 | 11.78 | 81.51 | 97.55 | 11.17 | 80.39 |
| UniEnt | 94.19 | 46.98 | 41.53 | 87.56 | 67.03 | 41.10 | 91.64 | 88.67 | 71.73 | 90.86 | 71.53 | 71.96 |
| TDA | 90.54 | 76.23 | 43.66 | 86.76 | 75.45 | 43.07 | 91.79 | 87.83 | 71.56 | 90.67 | 75.41 | 71.48 |
| DPE | 87.92 | 91.94 | 42.87 | 82.96 | 77.90 | 41.93 | 92.13 | 81.09 | 71.39 | 90.86 | 73.30 | 70.64 |
| **ROSITA** | **99.52** | **4.06** | **48.53** | **98.34** | **10.21** | **46.32** | **99.44** | **4.29** | **83.53** | **98.62** | **9.08** | **80.75** |
| | +6.13 | +51.46 | +7.10 | +12.45 | +62.70 | +5.49 | +8.17 | +86.80 | +12.03 | +8.19 | +65.96 | +9.09 |
| **MAPLE** | | | | | | | | | | | | |
| ZS-Eval | 81.49 | 92.95 | 41.70 | 83.26 | 71.15 | 42.77 | 90.15 | 83.54 | 74.42 | 92.74 | 65.70 | 75.71 |
| TPT | 81.38 | 93.17 | 39.92 | 83.18 | 71.52 | 40.93 | 90.14 | 83.58 | 74.00 | 92.74 | 65.68 | 75.23 |
| TPT-C | 83.25 | 87.60 | 42.81 | 83.18 | 70.60 | 42.86 | 90.35 | 81.49 | 74.73 | 92.79 | 65.20 | 75.59 |
| PAlign | 81.38 | 93.17 | 41.32 | 83.18 | 71.52 | 42.30 | 90.14 | 83.58 | 74.66 | 92.74 | 65.68 | 75.93 |
| PAlign-C | 71.22 | 86.32 | 27.14 | 32.17 | 94.32 | 15.44 | 92.20 | 59.70 | 75.23 | 93.54 | 54.59 | 75.67 |
| (K+1)PC | 98.58 | 3.35 | 48.69 | 77.17 | 39.74 | 38.10 | 99.01 | 3.16 | 84.23 | 95.14 | 13.77 | 80.16 |
| UniEnt | 81.53 | 93.45 | 41.50 | 83.41 | 70.84 | 42.78 | 90.14 | 83.49 | 74.48 | 90.14 | 83.49 | 74.48 |
| TDA | 76.79 | 99.02 | 42.98 | 82.46 | 91.75 | 44.63 | 90.43 | 86.56 | 73.66 | 92.92 | 64.63 | 74.16 |
| DPE | 73.97 | 99.59 | 41.39 | 80.06 | 87.10 | 44.05 | 90.44 | 78.77 | 72.67 | 93.48 | 55.74 | 76.74 |
| **ROSITA** | **99.56** | **1.66** | **51.30** | **98.68** | **5.09** | **50.67** | **99.39** | **2.95** | **84.70** | **97.85** | **12.98** | **83.07** |
| | +18.07 | +91.29 | +9.60 | +15.42 | +66.06 | +7.90 | +9.24 | +80.59 | +10.28 | +5.11 | +52.72 | +7.36 |

The LayerNorm parameters of the Vision Encoder are updated to minimize the following test-time objective to adapt the model one sample at a time in an online manner:

$$\mathcal{L}_{ReDUCe} = \begin{cases} \mathcal{L}_{Re} + \mathcal{L}_D & \text{if} \quad s_t > \mu_d \\ \mathcal{L}_U & \text{if} \quad s_t < \mu_u \end{cases} \tag{8}$$

This objective improves the proximity between the test sample and its positives, suitably chosen based on its score $s_t$, while also pushing apart the test sample and its negatives. This collectively encourages the model to adapt such that each of the desired classes and undesired classes are clustered and farther apart from each other, improving the overall classification performance of $C_d$ and $C_u$. We now perform Gradient Analysis on the loss function and theoretically justify how the proposed ReDUCe loss helps in enhancing the discriminability between desired and undesired class samples.

### 3.1 Gradient Analysis of the proposed REDUCE Loss

The key to understanding the behavior of the contrastive loss is to analyze its gradient. The softmax term in the denominator encourages $f_t$ to have lower similarity with negative samples, and the numerator encourages $f_t$ to have higher similarity with positive samples. We compute the gradient of the loss components $L_D$ and $L_U$ of the ReDUCe loss with respect to $f_t$ (Appendix A).

$$\begin{aligned} \frac{\partial \mathcal{L}_D}{\partial f_t} &= -\frac{1}{K^+} \sum_{z^+ \in Q_d} \mathbf{1}\left(y^+ = \hat{y}_t\right) \cdot \frac{1}{\tau}\left(z^+ - \sum_{z^- \in Q^u} p\left(z^-\right) z^-\right) \\ \frac{\partial \mathcal{L}_U}{\partial f_t} &= -\frac{1}{K} \sum_{z^+ \in Q_u} \frac{1}{\tau}\left(z^+ - \sum_{z^- \in Q_d} p\left(z^-\right) z^-\right) \end{aligned} \tag{9}$$

where $p\left(z^-\right)$ is the softmax probability of the negative samples defined as

$$p\left(z^-\right) = \frac{\exp\left(\text{sim}\left(f_t, z^-\right)/\tau\right)}{\sum_{z' \in Q^-} \exp\left(\text{sim}\left(f_t, z'\right)/\tau\right)} \tag{10}$$

where $Q^-$ is $Q_u$ for $\mathcal{L}_D$ and $Q_d$ for $\mathcal{L}_U$. The gradient of these contrastive loss formulations drives the following behavior in this context:

1. **Attraction to positive neighbors.** In the gradient of $\mathcal{L}_D$, the first term pulls the test feature $f_t$ towards its positives $z^+ \in Q_d$, representing the attraction force that encourages samples from desired classes to form $|C_d|$ tight clusters as the positives are chosen such that $\hat{y}_t = y^+$. Similarly, in the gradient of $\mathcal{L}_U$, the first term pulls $f_t$ towards its positives $z^+ \in Q_u$, encouraging all samples from $C_u$ to cluster together.

2. **Repulsion from negative neighbors.** The second term $p(z^-)z^-$ in the gradient pushes the test feature $f_t$ away from its negatives $z^- \in Q^-$ ($Q^-$ is $Q_u$ for $\mathcal{L}_D$ and $Q_d$ for $\mathcal{L}_U$). The strength of the repulsion is controlled by the softmax probability $p(z^-)$, where more similar negatives exert a stronger repulsive force on $f_t$, increasing the separation between samples from $C_d$ and $C_u$. As the negatives selected are its $K$ nearest neighbours of the opposite type, they are, in fact, hard negatives. Further, the contrastive objective inherently models the degree of hardness through the means of this probability $p(z^-)$. The closer the hard negative, the stronger the repulsion force.

We now present our analysis on the parameter choices for continuous adaptation of VLMs.

## 4 Analysis on parameters for Continuous Adaptation of VLMs

Test-Time adaptation methods using CNNs (Wang et al., 2021; Schneider et al., 2020; Liang et al., 2020; Chen et al., 2022) successfully leverage test domain data arriving in an online manner (in batches) to continuously update the model. In this work, we study TTA of VLMs like CLIP, which has only been explored very recently (Shu et al., 2022; Karmanov et al., 2024; Zhang et al., 2024) by adapting prompts independently for each image. While these methods show promise for on-the-fly adaptation in a zero-shot framework, it is not clear whether they can leverage the online data stream to continuously update the model parameters. Based on the evidence in prior TTA works (Wang et al., 2021; Chen et al., 2022), we analyze two aspects of VLMs for the TTA task: (1) Here, we question if VLMs can be continuously adapted in a similar manner, but using only a single test image at a time; (ii) If so, are prompts (Shu et al., 2022) the best parameters to continuously update?

**Experiment.** We choose six different parameter groups: (1) Prompts, (2) LayerNorm parameters (Zhao et al., 2023), (3) Full network, (4) First Attention Block of ViT, (5) Last Attention Block of ViT, (6) Prompts+LayerNorm(LN), 7) LoRA Adapters (Imam et al., 2024). We perform *single image TTA in a closed set scenario* on CIFAR-10C, by continuously adapting each of these parameter groups of CLIP, using reliable entropy loss, $L_{TTA} = \mathbf{1}(s_t > \tau)\mathcal{L}_{ent}(x_t)$, which is commonly used in several TTA methods (Wang et al., 2021; Niu et al., 2022) and VLM based prompt tuning methods like TPT, PAlign. Here, $x_t$ and $s_t$ refer to the test sample and its confidence, respectively. $\tau$ is the confidence threshold used to select reliable samples (Niu et al., 2022) for the model update, which we set to 0.7 in this analysis.

**Observations.** We find that continuous model adaptation can indeed improve VLMs performance based on our empirical analysis (Figure 2). (1) Using a high learning rate of $10^{-2}$ for any parameter group results in a severe drop in accuracy compared to the zero-shot performance of CLIP in this extreme setting of continuous single image model update. (2) The other extreme of low learning rate of $10^{-6}$ performs at par with ZSEval for all parameter groups, suggesting the model has not sufficiently changed. (3) Updating the Full Network results in an accuracy of about 10% across all learning rates, suggesting that giving the highest flexibility can cause the model to lose the inherent generalization ability of the VLM. (4) We also explore updating early attention layers and LoRA adapters. We observe these can be potential parameter choices on

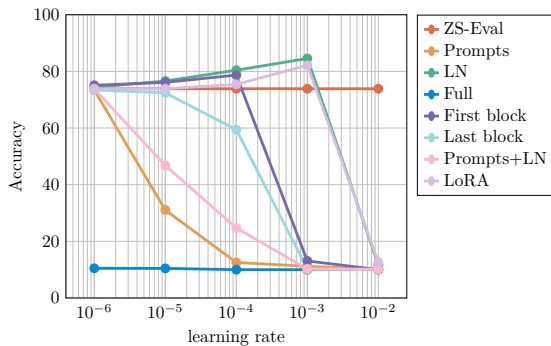

Figure 2: Accuracy on fine-tuning different parameter groups for single image TTA.

carefully choosing the learning rate (Appendix C.5). However, LoRA introduces additional parameters, scaling with the adapter rank. Unlike training-time scenarios where LoRA weights can be merged post-finetuning, continuous TTA cannot benefit from such merging, thereby increasing deployment complexity. Moreover, performance varies considerably with the choice of rank (Table 15), adding another hyper-parameter to be

tuned. (5) We find that LayerNorm tuning remains stable across optimizers, requires less hyper-parameter tuning, adds no additional model parameters, making it a more reliable and lightweight choice for our setting.

**Adapting Image encoder vs Text classifiers:** Most existing TTA approaches (Schneider et al., 2020; Wang et al., 2021; Chen et al., 2022) focus on adjusting image representations for domain shifts during test-time while keeping the classifiers fixed. This strategy helps retain class discriminative information. In contrast, in TPT and PAlign, the text-based classifiers that depend on learnable prompts are updated based on single images. While this does not impact zero-shot evaluation as the model weights are reset after each image, it can be detrimental during continuous updates.

Based on this analysis, we freeze the text-based classifiers and modify only the image representations using LayerNorm affine parameters. The rationale behind this approach is that text representations can be inherently more robust across domains. Text embeddings, often derived from a wide range of linguistic contexts, capture semantic meanings that are less susceptible to variations in visual data. Therefore, adapting the image encoder allows for more effective handling of domain shifts while retaining the class-level discriminative information from the text modality. This ensures that the model can be updated continuously without the need for resets, ultimately enhancing its performance in dynamic, real open-set environments.

## 5 Experiments

**Datasets.** We experiment with a diverse set of datasets to choose desired class data $D_d$ and undesired class data $D_u$. For $D_d$, we use CIFAR-10C (Hendrycks & Dietterich, 2019), CIFAR-100C (Hendrycks & Dietterich, 2019), ImageNet-C (Hendrycks & Dietterich, 2019), CCC Press et al. (2023) from the corruption category and ImageNet-R (Hendrycks et al., 2021), VisDA (Peng et al., 2017) and the Clipart, Painting, Sketch domains from DomainNet (Peng et al., 2019) as style transfer datasets. We introduce samples from MNIST (LeCun et al., 1998), SVHN (Netzer et al., 2011), CIFAR-10/100C (Hendrycks & Dietterich, 2019) and Tiny-ImageNet (Le & Yang, 2015) datasets as $D_u$ in the test stream. We describe the datasets in detail in Appendix B.3.

Table 2: $Acc_{HM}$ on VisDA dataset and Clipart, Painting, Sketch domains from DomainNet as $D_d$ and MNIST as $D_u$.

| Method | VisDA | Clipart | Painting | Sketch |
|---|---|---|---|---|
| ZSEval | 78.28 | 50.22 | 47.81 | 48.59 |
| TPT | 78.42 | 57.71 | 49.73 | 54.67 |
| TPT-C | 75.35 | 57.57 | 49.31 | 54.41 |
| (K+1)PC | 90.35 | 71.21 | 70.61 | 67.21 |
| UniEnt | 78.09 | 57.88 | 49.75 | 54.76 |
| TDA | 76.85 | 61.04 | 51.20 | 55.26 |
| DPE | 53.67 | 54.52 | 47.91 | 32.18 |
| ROSITA | 90.64 | 71.40 | 70.89 | 67.35 |
| | **+12.36** | **+21.18** | **+23.08** | **+18.76** |

**Implementation Details.** We use CLIP and MaPLe backbones with ViT-B16 architecture. For ROSITA, we use SGD optimizer with a learning rate of 0.001 to update the LayerNorm parameters of the Vision encoder. We set size of the score bank $\mathcal{S}$ to 512, number of neighbours $K$ to 5. The size of feature bank $M_d$ is set as $K \times C_d$ and that of $M_u$ to 64. Implementation details for all the baseline methods are presented in Appendix B.4. *We equip all methods with the same $C_d$ vs $C_u$ class identifier described in Section 2.2.* All experiments are done on a single NVIDIA A6000 GPU.

**Evaluation Metrics.** We employ standard metrics, namely Area Under the Receiver Operating Characteristic Curve (AUROC) and False Positive Rate at a True Positive Rate of 95% (FPR95), from the OOD detection literature (Lee et al., 2023; Li et al., 2023; Wang et al., 2023). Additionally, we compute the classification accuracy for desired class samples ($Acc_D$) and the binary classification accuracy for correctly recognizing samples from $C_u$ ($Acc_U$) as defined below. To gauge the overall performance, we compute $Acc_{HM}$ (HM), representing the harmonic mean of $Acc_D$ and $Acc_U$, which serves as a comprehensive metric capturing the trade-off between $Acc_D$ and $Acc_U$. Here, we summarily report AUROC (AUC), FPR95 (FPR) and $Acc_{HM}$ (HM) for all the datasets.

$$Acc_D = \frac{\sum_{(x_i,y_i)\in\mathcal{D}_d} \mathbf{1}\left(y_i = \hat{y}_i\right) \cdot \mathbf{1}\left(y_i \in C_d\right)}{\sum_{(x_i,y_i)\in\mathcal{D}_d} \mathbf{1}\left(y_i \in C_d\right)}; \quad Acc_U = \frac{\sum_{(x_i,y_i)\in\mathcal{D}_u} \mathbf{1}\left(\hat{y}_i \in C_u\right) \cdot \mathbf{1}\left(y_i \in C_u\right)}{\sum_{(x_i,y_i)\in\mathcal{D}_u} \mathbf{1}\left(y_i \in C_u\right)}$$

Table 3: Results with CIFAR-10C/100C as desired class data $D_d$ and four other datasets as $D_u$.

| | Method | MNIST | | | SVHN | | | Tiny-ImageNet | | | CIFAR-100C/10-C | | |
|---|---|---|---|---|---|---|---|---|---|---|---|---|---|
| | | AUC ↑ | FPR ↓ | HM ↑ | AUC ↑ | FPR ↓ | HM ↑ | AUC ↑ | FPR ↓ | HM ↑ | AUC ↑ | FPR ↓ | HM ↑ |
| **CIFAR-10C** — CLIP | ZS-Eval | 91.91 | 85.04 | 75.57 | 89.93 | 64.20 | 74.08 | 91.33 | 27.07 | 74.63 | 82.57 | 67.92 | 68.89 |
| | TPT | 91.89 | 85.55 | 75.81 | 89.93 | 64.41 | 74.36 | 91.31 | 27.23 | 75.17 | 82.57 | 68.06 | 69.17 |
| | TPT-C | 81.64 | 67.53 | 74.86 | 58.48 | 71.72 | 48.26 | 74.08 | 61.45 | 49.88 | 61.45 | 94.30 | 46.10 |
| | (K+1)PC | 98.05 | 12.50 | 83.27 | 80.74 | 50.33 | 70.10 | 87.09 | 52.29 | 73.98 | 62.55 | 91.68 | 56.46 |
| | UniEnt | 91.98 | 85.2 | 75.62 | 89.97 | 64.38 | 74.18 | 91.40 | 26.96 | 74.73 | 82.59 | 68.14 | 68.98 |
| | TDA | 92.94 | 71.11 | 77.06 | 92.02 | 52.68 | 76.64 | 91.68 | 25.37 | 75.94 | 83.54 | 66.06 | 70.13 |
| | DPE | 46.97 | 99.10 | 27.60 | 84.15 | 85.24 | 68.52 | 89.92 | 31.30 | 69.90 | 79.18 | 75.06 | 62.34 |
| | **ROSITA** | **99.10** | **7.63** | **84.17** | **94.79** | **32.59** | **78.80** | **96.43** | **12.10** | **80.06** | **82.99** | **62.89** | **69.56** |
| | | +7.19 | +77.41 | +8.60 | +4.86 | +31.61 | +4.72 | +5.10 | +14.97 | +5.43 | +0.42 | +5.03 | +0.6 |
| **CIFAR-10C** — MAPLE | ZS-Eval | 98.48 | 3.77 | 83.63 | 98.34 | **7.86** | 83.57 | 90.86 | 27.54 | 76.04 | 86.14 | 52.08 | 71.76 |
| | TPT | 98.15 | 5.67 | 81.56 | 98.34 | 7.89 | 82.73 | 90.86 | 27.61 | 75.46 | 86.15 | 52.14 | 70.94 |
| | TPT-C | 98.56 | **3.74** | 83.51 | 98.32 | 8.18 | 83.47 | 91.18 | 26.93 | 76.31 | 86.50 | 50.56 | 71.07 |
| | PAlign | 98.15 | 5.67 | 82.24 | **98.34** | 7.90 | 83.51 | 90.86 | 27.60 | 75.98 | 86.15 | 52.18 | 71.52 |
| | PAlign-C | 98.56 | 3.74 | 83.49 | 98.32 | 8.13 | 83.46 | 91.18 | 26.90 | 76.30 | 86.50 | 50.58 | 71.04 |
| | (K+1)PC | 98.34 | 9.63 | 86.52 | 71.01 | 78.78 | 68.70 | 71.20 | 85.81 | 68.29 | 62.35 | 88.44 | 61.89 |
| | UniEnt | 98.17 | 5.49 | 82.64 | 98.35 | 7.85 | 83.65 | 90.90 | 27.41 | 76.08 | 86.16 | 51.91 | 71.72 |
| | TDA | 98.42 | 4.13 | 81.97 | 98.60 | 6.20 | 83.95 | 91.27 | 27.00 | 76.84 | 86.72 | 51.40 | 72.61 |
| | DPE | 83.82 | 92.73 | 55.52 | 97.42 | 12.95 | 79.41 | 89.10 | 31.13 | 74.32 | 73.57 | 73.67 | 53.64 |
| | **ROSITA** | **99.34** | 5.22 | **87.63** | 97.80 | 13.15 | **84.17** | **91.67** | **25.31** | **77.67** | **86.82** | **50.33** | **73.15** |
| | | +0.86 | -1.45 | +4.00 | +0.54 | -5.29 | +0.60 | +0.81 | +2.23 | +1.63 | +0.68 | +1.75 | +1.39 |
| **CIFAR-100C** — CLIP | ZS-Eval | 77.78 | 99.93 | 48.39 | 64.70 | 98.68 | 45.85 | 67.31 | 73.89 | 45.80 | 63.28 | 93.25 | 44.04 |
| | TPT | 77.76 | 99.94 | 48.33 | 64.71 | 98.63 | 45.85 | 67.28 | 73.82 | 45.93 | 63.26 | 93.20 | 44.02 |
| | TPT-C | 51.57 | 100.00 | 27.04 | 9.40 | 99.98 | 5.74 | 59.74 | 79.76 | 18.41 | 55.86 | **86.35** | 13.64 |
| | (K+1)PC | 96.89 | 12.15 | 59.72 | 75.24 | 51.64 | 43.73 | 41.84 | 99.61 | 31.83 | 54.02 | 93.93 | 32.00 |
| | UniEnt | 77.94 | 99.93 | 48.32 | 64.78 | 98.61 | 45.84 | 67.40 | 73.77 | 45.83 | 63.28 | 93.18 | 44.04 |
| | TDA | 80.33 | 99.57 | 46.52 | 71.77 | 96.11 | 46.01 | 70.70 | 69.63 | 47.52 | 66.07 | 91.90 | 45.79 |
| | DPE | 67.06 | 99.88 | 42.54 | 43.23 | 99.79 | 35.69 | 61.42 | 80.62 | 42.80 | 60.08 | 92.80 | 42.21 |
| | **ROSITA** | **96.07** | **19.28** | **57.34** | **82.09** | **64.64** | **48.17** | **83.55** | **50.76** | **55.88** | **68.54** | 89.71 | **47.98** |
| | | +18.29 | +80.65 | +8.95 | +17.39 | +34.04 | +2.32 | +16.24 | +23.13 | +10.08 | +5.26 | +3.54 | +3.94 |
| **CIFAR-100C** — MAPLE | ZS-Eval | 87.43 | 64.19 | 54.97 | 92.98 | 40.51 | 56.42 | 68.80 | 74.35 | 48.24 | 66.93 | 87.94 | 46.06 |
| | TPT | 87.42 | 64.09 | 53.09 | 92.97 | 40.44 | 54.37 | 68.80 | **74.20** | 46.97 | 66.93 | 87.95 | 44.38 |
| | TPT-C | 87.65 | 63.08 | 55.14 | 93.09 | 40.30 | 56.31 | 68.85 | 74.71 | 48.53 | 66.97 | 87.94 | 46.30 |
| | PAlign | 87.42 | 64.11 | 53.98 | 92.97 | 40.48 | 55.37 | 68.80 | 74.23 | 47.69 | 66.93 | 87.93 | 45.16 |
| | PAlign-C | 88.25 | 57.31 | 55.69 | 93.45 | 39.39 | 57.39 | 68.76 | 78.12 | 48.15 | 66.82 | 87.80 | 47.01 |
| | (K+1)PC | 96.49 | 9.42 | 62.97 | 65.73 | 78.63 | 32.60 | 42.94 | 99.95 | 27.52 | 53.48 | 94.26 | 34.70 |
| | UniEnt | 87.40 | 64.02 | 54.86 | 92.99 | 40.36 | 56.42 | 68.84 | 74.26 | 48.41 | 66.93 | 87.96 | 46.09 |
| | TDA | 89.82 | 52.24 | 55.46 | 95.04 | 30.76 | 59.51 | 72.05 | 71.83 | 49.19 | 69.12 | 87.36 | 49.06 |
| | DPE | 39.05 | 98.88 | 33.66 | 84.29 | 76.13 | 52.20 | 63.74 | 82.75 | 45.74 | 65.61 | 90.67 | 46.36 |
| | **ROSITA** | **97.04** | **11.01** | **62.06** | **96.26** | **20.99** | **59.25** | **70.37** | 77.00 | **48.68** | **69.57** | **83.61** | **48.80** |
| | | +9.61 | +53.18 | +7.09 | +3.28 | +19.52 | +2.83 | +1.57 | -2.65 | +0.44 | +2.64 | +4.33 | +2.74 |

# 6 Research Questions

**1) How does ROSITA perform in comparison with prior methods in OSTTA setting?**

We observe, from Table 1, 2, 3 that TPT and PAlign perform similar to ZSEval in most datasets, as the prompts are reset after every single image update. On continuously updating prompts in TPT-C and PAlign-C, we observe a reduction in HM compared to ZS-Eval. The effect is more severe with CLIP when compared to MaPLe, as only the text prompts are updated keeping the vision encoder fixed (as also observed in Section 4). (K+1)PC and UniEnt, where LayerNorm tuning is done, perform better than prompt tuning methods. However, **ROSITA**, being equipped with a carefully designed objective to better discriminate between samples from $C_d$ and $C_u$ samples (Figure 3), results in overall better metrics in general.

**2) How does ROSITA perform in different real-world inspired OSTTA scenarios?**

**(a) Continuously changing domains:** We sequentially present 15 corruptions from CIFAR-10C, which form the domain $D_d$, alongside samples from four other datasets $D_u$. We also experiment with Continuously Changing Corruptions (CCC) (Press et al., 2023) benchmark where gradual domain changes are synthesized across 15 corruptions of ImageNet-C as $D_d$ and MNIST as $D_u$ and report the detailed results in Appendix D.1.
**(b) Frequently changing domains:** To further simulate more dynamic test environments, for CIFAR-10C/MNIST, we reduce the number of samples per corruption to 100, 250, 500, and 1000 in the continuously changing domain open-set TTA scenario. Reducing the sample count per corruption causes more frequent

Table 5: Performance in different Open-set TTA scenarios.

| Method | (a) Continuously changing domains | | | | | (b) Frequently changing domains | | | | (c) Varying ratio of $C_d/C_u$ | | | |
| | CIFAR-10C | | | | CCC | No. of samples per corruption | | | | Ratio | | | |
| | SVHN | MNIST | Tiny | C-100C | MNIST | 100 | 200 | 500 | 1000 | 0.2 | 0.4 | 0.6 | 0.8 |
|---|---|---|---|---|---|---|---|---|---|---|---|---|---|
| ZSEval | 64.33 | 64.04 | 66.50 | 58.49 | 31.01 | 61.41 | 61.87 | 61.42 | 63.30 | 75.56 | 75.59 | 75.57 | 75.56 |
| TPT | 64.26 | 64.03 | 66.50 | 58.47 | 31.04 | 61.33 | 62.32 | 61.59 | 63.24 | 75.67 | 75.75 | 75.81 | 75.83 |
| TPT-C | 33.05 | 46.44 | 59.38 | 37.24 | 13.56 | 60.62 | 61.30 | 57.16 | 34.88 | 72.70 | 74.31 | 74.79 | 75.16 |
| (K+1)PC | 65.13 | 62.52 | 66.93 | 57.46 | 33.84 | 60.90 | 60.76 | 61.40 | 63.26 | 62.31 | 68.85 | 81.70 | 82.90 |
| TDA | 66.02 | 66.44 | 67.64 | 59.44 | 33.87 | 60.17 | 61.43 | 63.22 | 64.82 | 72.45 | 75.04 | 77.54 | 77.91 |
| DPE | 23.36 | 50.12 | 58.96 | 35.56 | 31.16 | 47.48 | 46.22 | 39.83 | 46.52 | 65.67 | 66.12 | 56.38 | 29.98 |
| ROSITA | **66.86** | **65.26** | **68.89** | **59.16** | **34.84** | **61.64** | **66.82** | **67.97** | **73.24** | **82.96** | **83.97** | **84.51** | **84.37** |

domain changes, increasing the challenge for adaptation. **(c) Varying ratio of samples belonging to classes $C_d$ vs $C_u$:** We simulate real-world scenarios using the CIFAR-10C/MNIST dataset by varying the ratio of samples from the known classes $C_d$ versus unknown classes $C_u$ in the test stream by varying this ratio as 0.2, 0.4, 0.6, and 0.8. From results in Table 5, we observe that **ROSITA** demonstrates consistent superiority across all three open-set TTA scenarios, showcasing its capability to adapt effectively to both continuously and frequently changing domains, as well as varying class distributions.

**3) What is the importance of each loss component proposed in ROSITA?**

From Table 4, we observe that only using $\mathcal{L}_{Re}$ or $\mathcal{L}_D$ improves the metrics for CIFAR-10C dataset. For ImageNet-R (IN-R) as $D_d$, using $\mathcal{L}_{Re}$ or $\mathcal{L}_D$ is observed to increase FPR and decrease HM. IN-R has 200 classes making it a more challenging and confusing task compared to CIFAR-10C. This decrease in performance for IN-R can be attributed to the misclassification of some samples from $C_u$ as reliable desired class samples, increasing the confusion between $C_d$ and $C_u$ classes. Using $\mathcal{L}_U$ significantly reduces the confusion between samples from $C_d$ and

Table 4: Ablation study on loss components.

| $\mathcal{L}_{Re}$ | $\mathcal{L}_D$ | $\mathcal{L}_U$ | CIFAR-10C/MNIST | | | IN-R/MNIST | | |
| | | | AUC ↑ | FPR ↓ | HM ↑ | AUC ↑ | FPR ↓ | HM ↑ |
|---|---|---|---|---|---|---|---|---|
| ✗ | ✗ | ✗ | 91.91 | 85.04 | 75.57 | 91.27 | 91.09 | 71.5 |
| ✓ | ✗ | ✗ | 95.29 | 30.82 | 80.97 | 81.07 | 99.02 | 64.32 |
| ✗ | ✓ | ✗ | 95.23 | 28.91 | 79.71 | 87.73 | 94.67 | 67.28 |
| ✗ | ✗ | ✓ | 98.61 | 12.73 | 79.84 | 99.39 | 4.81 | 80.82 |
| ✓ | ✓ | ✗ | 96.23 | 22.73 | 79.24 | 76.78 | 99.22 | 62.54 |
| ✓ | ✗ | ✓ | 98.69 | 12.06 | 82.98 | 99.34 | 4.67 | 82.98 |
| ✗ | ✓ | ✓ | **99.27** | **4.15** | 80.69 | **99.48** | 4.40 | 81.92 |
| ✓ | ✓ | ✓ | 99.10 | 7.63 | **84.17** | 99.44 | **4.29** | **83.53** |

$C_u$, shown by the significant drop in FPR compared to ZSEval. The contrastive objectives $\mathcal{L}_D$ and $\mathcal{L}_U$ to separate the two types of samples, in conjunction with $\mathcal{L}_{Re}$ which aids to improve the $|C_d|$-way classification of desired class samples, gives the overall best results.

**4) What is the role of using reliable samples for OSTTA in ROSITA?**

To understand the role of selecting reliable samples for TTA, we do a simple experiment where we only use the threshold $\tau_t$ to distinguish between $C_d$ and $C_u$ samples. For all the samples with $s_t > \tau_t$ identified to belong to $C_d$, we perform TTA using $\mathcal{L}_{Re} + \mathcal{L}_D$ (Equation 6). Similarly, we use $L_U$ (Equation 7), for all samples identified to belong to

Table 6: Need for Reliable samples.

| Thresholds | $D_u$: MNIST | | | | |
| $\tau_u/\tau_t/\tau_d$ | C-10C | C-100C | IN-C | IN-R | VisDA |
|---|---|---|---|---|---|
| $\tau_t/\tau_t/\tau_t$ | **84.99** | 55.16 | 44.05 | 83.28 | **91.24** |
| $\mu_u/\tau_t/\mu_d$ | 84.17 | **57.34** | **48.53** | **83.53** | 90.64 |

$C_u$ based on the criterion $s_t < \tau_t$. From the results in Table 6, we see that, for CIFAR-10C and VisDA, this case performs slightly better than our case(last row in Table 6) where TTA is performed only on reliable samples. CIFAR-10C and VisDA dataset have 10 and 12 classes of interest respectively. The zero shot performance of these datasets being good, as the class confusion is less, using all samples for TTA can be helpful. On the other hand, the classification in CIFAR-100C, ImageNet-C and ImageNet-R is harder, due the inherent confusion arising due to the large number of classes. Using non reliable test samples, with scores in the range $\mu_u < s_t < \mu_d$ can adversely affect the adaptation process. Hence, using only reliable samples for TTA performs better for these datasets as seen in Table 6). In a real world test-time adaptation scenario, where we have no prior information about the difficulty of the classification task, in terms of severity of domain shift and class confusion, it is desirable to only use reliable samples for model updates.

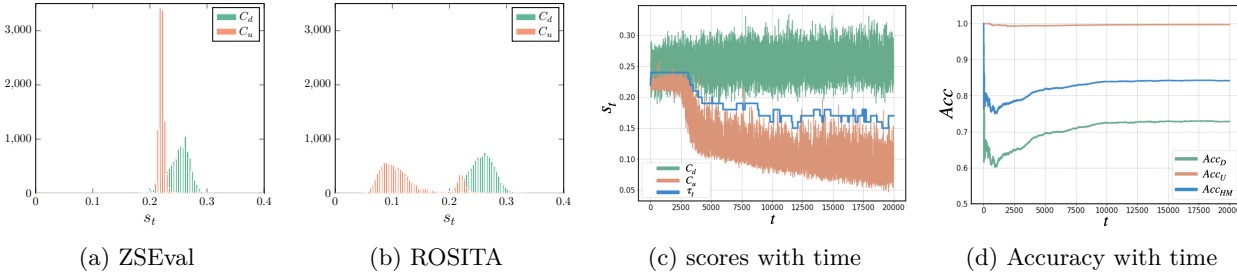

| (a) ZSEval | (b) ROSITA | (c) scores with time | (d) Accuracy with time |

Figure 3: Histograms of the scores $s_t$ for ZS-Eval (a) and ROSITA (b) on CIFAR-10C/MNIST dataset. (c) Change in scores for $C_d$ and $C_u$ class samples, the best threshold $\tau_t$; (d) Accuracy metrics on samples seen until time t. Samples from $C_d$ and $C_u$ separate better and the accuracy metrics improve with time.

**5) How do the scores $s_t$ and the performance of ROSITA vary with time?**

We plot the scores $s_t$ of samples from $C_d$ and $C_u$ over time, along with the threshold $\tau_t$, in Figure 3c using ROSITA. Initially, the scores of $C_d$ and $C_u$ overlap significantly ($t < 2500$), leading to unstable performance as shown in Figure 3d. During this phase, the threshold $\tau_t$ tends to classify most $C_u$ samples correctly, resulting in high $Acc_U$ but low $Acc_D$, as many desired class samples are incorrectly rejected. However, as the ReDUCe loss progressively improves class separability, $\tau_t$ adapts to the evolving score distribution, enhancing discrimination between $C_d$ and $C_u$. This refinement stabilizes the model's performance, yielding steady improvements in $Acc_D$ and $Acc_{HM}$ for $t > 2500$. The instability observed for $t < 1500$ is attributed to the initial learning process and the small sample size, as accuracy is measured on the cumulative number of samples seen up to time $t$, which is exactly $t$ in single image TTA.

**6) How does ROSITA fare in terms of memory required?**

Prompt tuning methods like TPT, PAlign do not require any memory buffer. TDA requires a memory buffer of size $(|C_d| \times (3+2)) \times F$ to store 3 features per desired class in the positive cache and 2 features per class in the negative cache. DPE requires a memory buffer of size $(|C_d| \times 3) \times F$ to store 3 features per desired class. ROSITA requires a memory buffer of size $(|C_d| \times K + |M_u|) \times F$ for the two feature banks.

Table 7: Memory overhead in ROSITA.

| Dataset | $|C_d|$ | No. of features | Memory (in MB) |
|---|---|---|---|
| CIFAR-10C | 10 | 5x10+64 | 0.758 |
| VisDA | 12 | 5x12+64 | 0.778 |
| CIFAR-100C | 100 | 5x100+64 | 1.679 |
| ImageNet-R | 200 | 5x200+64 | 2.703 |
| ImageNet-C | 1000 | 5x1000+64 | 10.89 |

For a ViT-B16 ($F = 512$) model with ImageNet-C ($|C_d| = 1000$), the required memory buffer size is $5 \times 1000 \times 512 + 64 \times 512$ (10.89MB) from Table 7. *The memory to store these features and computation required to compute feature similarity is as lightweight as performing a forward pass through a simple linear layer, demonstrating the memory and computational efficiency of* **ROSITA** *for real time applications.*

**7) How does ROSITA fare in terms of the GPU memory required and inference time?**

The GPU memory and time taken (secs/image) for prompt tuning methods TPT scales with the number of classes, as more memory is required to store the intermediate activations and gradients to backward pass through the text encoder. On the other hand, ROSITA requires two forward passes and one backward pass through the vision encoder for reliable test samples. Figure 4 compares the GPU memory and time complexity of ZS-Eval, TPT, and ROSITA representative of training-free methods (ZS-Eval, TDA), prompt-tuning (TPT/-C, PAlign/-C), and LayerNorm-tuning((K+1)PC, UniEnt, ROSITA) based methods in Figure 4. For e.g., for ImageNet-C dataset with 1000 classes, ZSEval, TPT and ROSITA

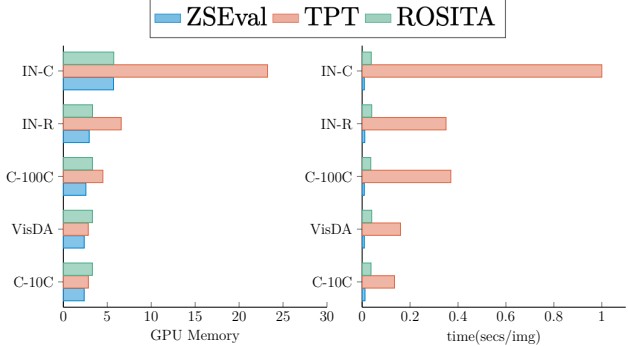

Figure 4: Complexity Analysis of different methods.

require 5.71 GB, 23.24 GB and 5.73 GB GPU memory to perform a single image based model update. Thus, **ROSITA** achieves computational efficiency comparable to training-free methods while being far more efficient

than prompt-tuning approaches. Despite its minimal computational overhead, ROSITA offers substantial performance gains, providing a balanced trade-off between efficiency and effectiveness for OSTTA scenarios.

**8) What are the key factors distinguishing ROSITA from prior works?**

1. *Enhanced use of LDA Statistics to identify reliable samples*: Apart from the threshold $\tau_t$, ROSITA leverages the score statistics $\mu_d$ and $\mu_u$ provided by the LDA class identifier, combined with the novel ReDUCe loss function, to adapt the model. This synergy enhances the discriminability between desired ($C_d$) and undesired ($C_u$) class samples, offering a clear advantage over baselines that use the same LDA identifier but fail to exploit this additional information (Figure 3).

2. *Bridging CNN and VLM-Based TTA insights*: ROSITA integrates key insights from CNN-based TTA methods such as normalization layer updates with vision-language models (VLMs) (Section 4). While simple in hindsight, this baseline was overlooked in prior VLM-based TTA works (Shu et al., 2022; Karmanov et al., 2024; Zhang et al., 2024). In this work, we attempt to highlight how these learnings can translate effectively to VLMs, underscoring their utility as a foundational approach for TTA.

3. *Holistic design for Open-set TTA*: ROSITA introduces the ReDUCe loss to distinctly separate desired ($C_d$) and undesired ($C_u$) class samples using compact feature banks. Although it is inspired by contrastive learning frameworks (Chen et al., 2020; 2022), it is specifically designed for open-set TTA: (i) Reliable samples from $C_u$ use nearest $C_u$ samples as negatives, and vice versa (ii) Unlike the $C_d$+1-way classification in (Li et al., 2023), ROSITA forces $C_d$ features to form distinct clusters and pushes $C_u$ features away. (iii) The feature banks are populated only with reliable samples, ensuring robust updates during adaptation. This approach specifically mitigates the significant overlap of scores $s_t$ between $C_d$ and $C_u$ in vision-language models, hence reducing misclassification and boosting discriminability.

**9) What are the limitations of ROSITA which can be addressed in future?**

Although we follow the dataset choices in (K+1)PC Li et al. (2023), the first work on open-set TTA, we acknowledge that using MNIST as undesired class samples may seem unrealistic for natural image scenarios. However, even in this seemingly simple open-set scenario, our observations (Tables 2 3) show that CLIP struggles to reject MNIST samples, with high FPR for most prior methods and significant overlap in similarity scores between desired and undesired classes (Figure 3a). We have also conducted experiments using CIFAR-10C as desired classes and CIFAR-100C as undesired ones, which consist of corrupted but semantically similar images. While gains are less pronounced than in simpler settings like MNIST, our findings underscore the core challenge of enabling VLMs to reliably say "I don't know.". Additionally, ROSITA was not benchmarked in standard TTA setting where there are no undesired class samples in the test stream. Future work could explore a unified TTA framework that adapts effectively regardless of the nature of test-time samples.

**Appendix.** We present more detailed experimental analysis in the Appendix: C.1 Analysis on error bars, C.2 Analysis of parameter $K$, C.3 Detailed analysis of ReDUCe Loss components, C.4 Comparison of different $C_d$ vs $C_u$ Class identifiers for Open-set TTA, C.5 Extensive analysis on parameter choice for continuous adaptation of VLMs, D Additional experiments demonstrating the generalizability of ROSITA to more datasets, scenarios and VLM backbones, E Failure case analysis, F Broader impact concerns.

## 7 Conclusion

In this work, we address the challenging problem of **Open-set Single-image Test-Time Adaptation** (OSTTA), where models must adapt continuously to shifting data distributions and distinguish between known and unknown classes, all while processing test samples one at a time. To advance research in this area, we establish a comprehensive benchmark for OSTTA using Vision-Language Models (VLMs), bridging the gap between open-set recognition and sequential adaptation in dynamic environments. We propose **ROSITA**, a novel framework specifically designed for OSTTA, overcoming the limitations of prior methods that assume closed-set conditions or batch-wise test processing. ROSITA leverages two dynamically updated feature banks to differentiate between desired and undesired class samples. At its core, the proposed ReDUCe loss facilitates effective model adaptation by leveraging reliable samples while mitigating the negative influence of undesired class samples. Extensive experiments across diverse domain adaptation benchmarks demonstrate that ROSITA consistently outperforms prior methods, achieving good accuracy with computational efficiency.

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

# APPENDIX

## A   Gradient Analysis of the ReDUCe Loss

Here, we delve deeper into the ReDUCe loss function in ROSITA, breaking down its key components and mathematically demonstrate why the proposed objective improves the separation of $C_d$ and $C_u$ samples. We'll focus on contrastive loss components $L_D$ and $L_U$ which are designed to improve discriminability.

**ReDUCe loss in a nutshell.**   A test sample $x_t$ arrives at time $t$ with feature representation $f_t$. Two feature banks, $\mathcal{M}_w$ and $\mathcal{M}_s$ store reliable sample features from $C_d$ and $C_u$ respectively. ReDUCe loss aims to pull the test sample's feature $f_t$ towards its positive samples $z^+$, which are its K nearest neighbors $Q_d = \mathrm{k}NN(f_t; M_d)$ if it is a reliable $C_d$ sample or $Q_u = \mathrm{k}NN(f_t; M_u)$ if it is a reliable $C_u$ sample. The feature $f_t$ is pushed away from its negative samples $z^-$, which are the K nearest neighbors from the undesired feature bank $M_u$ if it is a reliable $C_d$ sample or from the desired feature bank $M_d$ if it is a reliable $C_u$ sample. The features $f_t, z^+, z^-$ are all unit norm vectors. The key to understanding the behavior of the proposed loss is to analyze its gradient.

**Gradient of $L_D$ with respect to $f_t$:**

The contrastive loss for desired class samples $L_D$ is defined as:

$$
\begin{aligned}
\mathcal{L}_D &= -\frac{1}{K^+} \sum_{z^+ \in Q_d} \mathbf{1}(y^+ = \hat{y}_t) \log \frac{\exp\left(\mathrm{sim}\left(f_t, z^+\right)/\tau\right)}{\sum_{z^- \in Q_u} \exp(\mathrm{sim}(f_t, z^-)/\tau)} \\
\frac{\partial \mathcal{L}_D}{\partial f_t} &= -\frac{1}{K^+} \sum_{z^+ \in Q_d} \mathbf{1}(y^+ = \hat{y}_t) \frac{\partial}{\partial f_t} \log \frac{\exp\left(\mathrm{sim}\left(f_t, z^+\right)/\tau\right)}{\sum_{z^- \in Q_u} \exp(\mathrm{sim}(f_t, z^-)/\tau)}
\end{aligned}
\tag{11}
$$

The loss is of the log-softmax structure. Consider gradient of the following term:

$$
\frac{\partial}{\partial f_t} \log \frac{\exp\left(\mathrm{sim}\left(f_t, z^+\right)/\tau\right)}{\sum_{z^- \in Q} \exp(\mathrm{sim}(f_t, z^-)/\tau)} = \frac{\partial}{\partial f_t}\left(\frac{\mathrm{sim}\left(f_t, z^+\right)}{\tau}\right) - \frac{\partial}{\partial f_t} \log \sum_{z^- \in Q} \exp(\mathrm{sim}(f_t, z^-)/\tau)
$$

The gradients of the two terms involved are

$$
\begin{aligned}
\frac{\partial}{\partial f_t}\left(\frac{\mathrm{sim}\left(f_t, z^+\right)}{\tau}\right) &= \frac{z^+}{\tau} \\
\frac{\partial}{\partial f_t} \log \sum_{z^- \in Q} \exp(\mathrm{sim}(f_t, z^-)/\tau) &= \frac{\sum_{z^- \in Q} \frac{\partial}{\partial f_t} \exp(\mathrm{sim}(f_t, z^-)/\tau)}{\sum_{z^- \in Q} \exp(\mathrm{sim}(f_t, z^-)/\tau)} \\
&= \frac{1}{\tau} \cdot \frac{\sum_{z^- \in Q} \exp(\mathrm{sim}(f_t, z^-)/\tau)}{\sum_{z^- \in Q} \exp(\mathrm{sim}(f_t, z^-)/\tau)z^-} \\
&= \frac{1}{\tau} \cdot \sum_{z^- \in Q} p(z^-)z^-
\end{aligned}
\tag{12}
$$

The final gradient of the log-softmax term is

$$
\frac{\partial}{\partial f_t} \log \frac{\exp\left(\mathrm{sim}\left(f_t, z^+\right)/\tau\right)}{\sum_{z^- \in Q} \exp(\mathrm{sim}(f_t, z^-)/\tau)} = \left(z^+ - \sum_{z^- \in Q} p\left(z^-\right) z^-\right)
$$

where $p\left(z^-\right)$ is the softmax probability of the negative samples defined as

$$p\left(z^-\right) = \frac{\exp\left(\text{sim}\left(f_t, z^-\right)/\tau\right)}{\sum\limits_{z' \in Q^-} \exp\left(\text{sim}\left(f_t, z'\right)/\tau\right)}$$

Substituting Equation 12 in Equation 11, we get the gradient of the desired sample contrastive loss $L_D$ with respect to $f_t$ as

$$\frac{\partial \mathcal{L}_D}{\partial f_t} = -\frac{1}{K^+} \sum_{z^+ \in Q_d} \mathbf{1}(y^+ = \hat{y}_t)\left(z^+ - \sum_{z^- \in Q_u} p\left(z^-\right) z^-\right) \tag{13}$$

**Gradient of $L_D$ with respect to $f_t$:**

The contrastive loss for desired class samples $L_D$ is defined as:

$$
\begin{aligned}
\mathcal{L}_U &= -\frac{1}{K} \sum_{z^+ \in Q_u} \log \frac{\exp\left(\text{sim}\left(f_t, z^+\right)/\tau\right)}{\sum\limits_{z^- \in Q_d} \exp(\text{sim}(f_t, z^-)/\tau)} \\
\frac{\partial \mathcal{L}_U}{\partial f_t} &= -\frac{1}{K^+} \sum_{z^+ \in Q_u} \frac{\partial}{\partial f_t} \log \frac{\exp\left(\text{sim}\left(f_t, z^+\right)/\tau\right)}{\sum\limits_{z^- \in Q_d} \exp(\text{sim}(f_t, z^-)/\tau)}
\end{aligned}
\tag{14}
$$

Substituting Equation 12 in Equation 14, we get:

$$\frac{\partial \mathcal{L}_U}{\partial f_t} = -\frac{1}{K^+} \sum_{z^+ \in Q_u} \left(z^+ - \sum_{z^- \in Q_d} p\left(z^-\right) z^-\right) \tag{15}$$

**Interpretation of the Gradients:**

- Both the gradient terms in Equations 13 and 15 have two components: Positive term $z^+$ and Negative term $p\left(z^-\right) z^-$. The positives and negatives are suitably chosen from the desired and undesired feature banks.

- Positive term $z^+$: The term $z^+$ pulls the test feature $f_t$ closer to its feature vectors $z^+$. This term represents the attraction force that encourages $C_d$ samples to cluster together in $L_D$ and $C_u$ samples to cluster together in $L_U$.

- Negative term $p\left(z^-\right) z^-$: The negative samples $z^-$ exert a repulsive force, pushing $f_t$ away from them. The strength of this repulsion is controlled by the softmax probabilities $p\left(z^-\right)$, where higher similarity between $f_t$ and $z^-$ increases the repulsion force. This inherently models the degree of hard negatives from the negative feature bank.

- The overall gradient update encourages $f_t$ to move closer to its positives while moving away from its negatives, enhancing the separation between samples from $C_d$ and $C_u$ classes.

## B Baselines

### B.1 Vision Language Models

**CLIP** (Radford et al., 2021) is a multimodal VLM consisting of two modules: Vision encoder and Text encoder denoted as $\mathcal{F}_V$ and $\mathcal{F}_T$ respectively. During pre-training, the two modules are jointly trained in a contrastive self-supervised fashion to align massive amounts of web scrapped image-text pairs. CLIP has demonstrated impressive zero-shot performance across a wide variety of datasets.

**MaPLe** (Khattak et al., 2023) is a multimodal prompt learner model that simultaneously adapts both vision and text encoders while fine-tuning CLIP for downstream tasks. They use learnable text prompts $\boldsymbol{p}_T$ and bridge the two modalities using visual prompts obtained as $\boldsymbol{p}_V = \mathrm{Proj}(\boldsymbol{p}_T)$. Learnable tokens are also introduced in deeper layers of both image and text encoders, enabling progressive feature adaptation.

### B.2 Methods

**ZSEval (Radford et al., 2021):** Given a test image $x_t$, the image feature is extracted from the vision encoder as $f_t = \mathcal{F}_V(x_t)$. For a $C$-class classification problem, the classifier is obtained by prepending a predefined text prompt $\boldsymbol{p}_T$="A photo of a", with the class names $\{c_1, c_2, \ldots c_C\}$ to form class specific text inputs $\{\boldsymbol{p}_T, c_i\}$ for $i \in \{1, \ldots C\}$. These texts are then embedded through the text encoder as $\boldsymbol{t}_i = \mathcal{F}_T(\{\boldsymbol{p}_T; c_i\})$ to get the text classifiers $\{\boldsymbol{t}_1, \boldsymbol{t}_2, \ldots \boldsymbol{t}_C\}$. The class prediction is made by identifying the text feature $\boldsymbol{t}_i$ which has the highest similarity with the image feature $f_t$.

**TPT (Shu et al., 2022)** aims to improve the zero shot generalization ability of CLIP by providing custom adaptable context for each image. This is done by prepending learnable text prompts $\boldsymbol{p}_T$ to the class names. The text classifiers $\boldsymbol{t}_i = \mathcal{F}_T(\{\boldsymbol{p}_T; c_i\}), i \in \{1, 2, \ldots C\}$ are now a function of these learnable prompts, which are specially adapted for each test image using an entropy minimization objective as $\arg\min_{\boldsymbol{p}_T} \mathcal{L}_{\mathrm{ent}}$ . The entropy is obtained using the average score vector of the filtered augmented views.

**PromptAlign (PAlign) (Samadh et al., 2023)** leverages multimodal prompt learner model MaPLe (Khattak et al., 2023) to facilitate the adaptation of both vision and language encoders for each test sample. They align the token distributions of source and target domains, considering ImageNet as a proxy for the source dataset of CLIP. The vision and language prompts of MaPLe are optimized with the objective $\arg\min_{\{\boldsymbol{p}_V, \boldsymbol{p}_T\}} \mathcal{L}_{ent} + \mathcal{L}_{align}$ for each sample $x_t$.

**TPT-C (Shu et al., 2022)/PAlign-C (Samadh et al., 2023)**: We adapt TPT and PAlign for continuous model update, which we refer as TPT-C and PAlign-C respectively. The prompts $\{\boldsymbol{p}_T\}$ and $\{\boldsymbol{p}_V, \boldsymbol{p}_T\}$ in TPT and PAlign are continuously updated with the test stream with their respective test objectives.

**(K+1)PC (Li et al., 2023)**: This was the first work exploring open world TTA, however it was done in the context of CNNs and not VLMs. Also, the test samples come in batches, while we perform single image TTA. We adapt this method for our problem setting as follows: As we use VLMs, we use the text prototypes (instead of the source prototypes). The prototype pool is dynamically updated by adding features of reliable test samples recognized to belong to undesired classes. The vision encoder is updated using a (K+1) way prototypical cross entropy loss.

**TDA (Karmanov et al., 2024)**: TDA is a training-free dynamic adapter for TTA in vision-language models, utilizing a lightweight key-value cache for efficient pseudo label refinement without backpropagation.

**DPE (Zhang et al., 2024)**: DPE accumulates task-specific knowledge by dynamically evolving two sets of prototypes, textual and visual, during test time. These prototypes are refined to capture increasingly accurate multi-modal representations for target classes. To ensure consistency between modalities, DPE incorporates learnable residuals for each test sample, aligning textual and visual prototypes for improved representation alignment.

**UniEnt (Gao et al., 2024)**: This is a very recent work addressing open-set TTA in the context of CNNs. They use a Distribution Aware Filter (DAF) based on Gaussian Mixture Modeling of the scores to distinguish

between desired and undesired class samples. They employ entropy minimization and entropy maximization objectives for desired and undesired class samples respectively.

We equip all the baselines with the same LDA based Desired vs Undesired class identifier described in Section 2.2 for fair comparison of the TTA methods for this problem.

### B.3 Datasets

We experiment with diverse datasets encompassing corruption, style transfer and other common datasets.

**CIFAR10-C** (Hendrycks & Dietterich, 2019) is a small-scale corruption dataset of 10 classes with 15 common corruption types. It consists of 10,000 images for each corruption.

**CIFAR-100C** (Hendrycks & Dietterich, 2019) is also a corruption dataset with 100 classes and 15 corruption types. It also consists of 10,000 images for each corruption.

**ImageNet-C** (Hendrycks & Dietterich, 2019) is a large-scale corruption dataset spanning 1000 categories with a total of 50,000 images. 15 types of corruption images are synthesized from these 50,000 images.

**ImageNet-R** (Hendrycks et al., 2021) is a realistic style transfer dataset encompassing interpretations of 200 ImageNet classes, amounting to a total of 30,000 images.

**VisDA** (Peng et al., 2017) is a synthetic-to-real large-scale dataset, comprising of 152,397 synthetic training images and 55,388 real testing images across 12 categories.

**DomainNet** (Peng et al., 2019) is a large-scale domain adaptation dataset. We use the Clipart, Painting and Sketch domains with 345 categories from the DomainNet dataset for our experiments.

**CCC** Press et al. (2023) is a benchmark designed to assess long-term continual TTA behavior in a changing world, covering scenarios such as weather changing from foggy to rainy, day to night.

**MNIST** (LeCun et al., 1998) is a dataset of handwritten images consisting of 60,000 training and 10,000 testing images.

**SVHN** (Netzer et al., 2011) is also a digits dataset with house numbers captured from real streets. It consists of 50,000 training images and 10,000 testing images.

We perform experiments on eight domains $D_d$ for desired class samples. The corresponding $D_u$ are chosen such that there is no overlap between the classes $C_d$ and $C_u$ as described in Table 8. The 15 corruptions of CIFAR-10C/100C and ImageNet-C fall into four categories: synthetic weather effects, per-pixel noise, blurring, and digital transforms. *snow* corruption is a synthesized weather effect on which all the main experiments of CIFAR-10C, CIFAR-100C and ImageNet-C are done. To evaluate the robustness of our method across different corruption types, we do additional experiments with *impulse noise* , *motion blur* and *jpeg compression* corruptions from the categories per-pixel noise, blurring and digital transforms respectively and report the results in Section D.4.

Table 8: Details of desired and undesired class dataset combinations

| Datasets | | # images | | |
|---|---|---|---|---|
| $D_d$ | $D_u$ | $D_d$ | $D_u$ | Total |
| CIFAR-10C | MNIST, SVHN, Tiny ImageNet, CIFAR-100C | 10000 | 10000 | 20000 |
| CIFAR-100C | MNIST, SVHN, Tiny ImageNet, CIFAR-10C | 10000 | 10000 | 20000 |
| ImageNet-C | MNIST, SVHN | 50000 | 50000 | 100000 |
| ImageNet-R | MNIST, SVHN | 30000 | 30000 | 60000 |
| VisDA | MNIST, SVHN | 50000 | 50000 | 100000 |
| Clipart | MNIST, SVHN | 29208 | 29208 | 58416 |
| Painting | MNIST, SVHN | 43700 | 43700 | 87400 |
| Sketch | MNIST, SVHN | 41832 | 41832 | 83664 |

## B.4 Implementation Details

Here, we describe the parameters chosen for all the baseline methods and our proposed method.

**TPT (Shu et al., 2022):** The prompt is initialized with the default *A photo of a* text. The corresponding 4 tokens in the input text embedding space are optimized for each test image. The prompt is **reset** after each update. A single test image is augmented 63 times using random resized crops to create a batch of 64 images. The confident samples with 10% lowest entropy are selected. The test time loss is the entropy of the averaged prediction of the selected confident samples. AdamW optimizer with a learning rate of $5e^{-4}$ is used, following (Shu et al., 2022).

**PAlign (Samadh et al., 2023):** Following Promp-tAlign (Samadh et al., 2023), MaPLe (Khattak et al., 2023) model trained on ImageNet using 16-shot training data with 2 prompt tokens for a depth of 3 layers is used. The prompts on both the text and vision encoders are optimized on a single test image. Similar to TPT, 10% of 64 augmentations are selected to compute the entropy loss. The token distribution loss to align the token statistics of test with that of source data is computed for all 64 images. AdamW optimizer with a learning rate of $5e^{-4}$ to update the prompts for each image, following (Samadh et al., 2023). The prompts are **reset** to the ImageNet trained prompts after each update.

**TPT-C (Shu et al., 2022)/ PAlign-C (Samadh et al., 2023):** We create the continuous prompt update versions of TPT and PAlign as TPT-C and PAlign-C respectively. The only difference is that the prompts are continuously updated using the test stream of samples. If a sample is detected as reliable $C_d$ sample, the

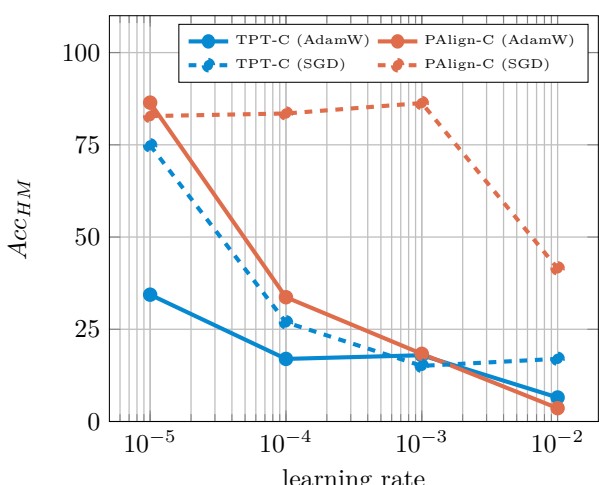

Figure 5: Performance of TPT-C and PAlign-C for CIFAR-10C/MNIST with AdamW and SGD optimizer on varying learning rates.

respective test time objectives are used to update the prompts. For this purpose, we vary the learning rate and optimizer to select the best optimizer for continuous prompt update. On performing experiments on CIFAR-10C/MNIST data, from Figure 5, we observe that SGD optimizer with learning rate $10^{-5}$ works the best for continuous prompt update and hence we use this for all the experiments of TPT-C and PAlign-C.

**(K+1)PC (Li et al., 2023):** The vision encoder is updated using a (K+1) way prototypical cross entropy loss . The prototypes are updated using the test stream of samples. The learning rate is set to 0.001.

**TDA (Karmanov et al., 2024)**: We use $\tau_t$ from the LDA based $C_d$ vs $C_u$ identifier to recognise the desired and undesired class samples. Following (Karmanov et al., 2024), we set the shot capacity to 3 and the number of key-value caches is $C_d$ as we use the adapter only for desired class samples.

**DPE (Zhang et al., 2024):** We use the same LDA based $C_d$ vs $C_u$ identifier to recognise the desired and undesired class samples. We use the same hyperparameters presented in (Zhang et al., 2024). A priority queue storing 3 visual features per class is used. The text and visual prototype residuals are updated with a learning rate of 0.0006 using AdamW optimizer.

**UniEnt (Gao et al., 2024)**: We use the UniEnt objective in combination with LDA based class indentifier. The entropy minimization and maximization objectives are used for desired and undesired class samples respectively. The LayerNorm parameters are updated with a learning rate of 0.001 using SGD optimizer.

**ROSITA:** We use SGD optimizer with a learning rate of 0.001 to update the LayerNorm affine parameters of the Vision encoder. We set the size of score bank $\mathcal{S}$ to 512, number of neighbours $K$ to 5 and the size of $M_u$ is set to to 64.

# C   Additional Analysis

In this section, in addition to the analysis done in Section 6, we study the robustness of the proposed method ROSITA more extensively, in the terms of (1) Error bars on different test data streams, (2) Role of the parameter $K$, the number of neighbours, (3) Analysis of the scores $s_t$ on using different combinations of the proposed loss components, (4) Comparison of different $C_d$ vs $C_u$ Class identifiers for Open-set TTA.

## C.1   Analysis on error bars

To study the robustness of our method for differently ordered test streams, we run ROSITA with five random seeds and report the Mean and Standard deviation of the $Acc_{HM}$ in Table 9 for CIFAR-10C/100C as $D_d$ and MNIST, SVHN, Tiny ImageNet, CIFAR-100C/10C as $D_u$ (corresponding to our results in Table 3 in the main paper). We observe that the variance in the performance of ROSITA is very low, reinforcing the robustness of the proposed method for different shuffled datasets and augmentations created.

Table 9: Performance (Mean and Standard deviation of $Acc_{HM}$) of ROSITA across 5 random seeds for CIFAR-10/100C as $D_d$ with 4 other datasets as $D_u$.

| $D_d \backslash D_u$ | MNIST | SVHN | Tiny | CIFAR-100/10C |
|---|---|---|---|---|
| CIFAR-10C | $84.07 \pm 0.023$ | $78.90 \pm 0.038$ | $80.10 \pm 0.014$ | $69.44 \pm 0.018$ |
| CIFAR-100C | $57.09 \pm 0.041$ | $47.90 \pm 0.047$ | $55.95 \pm 0.051$ | $48.10 \pm 0.024$ |

## C.2   Analysis on parameter K

Table 10: Performance ($Acc_{HM}$) on varying $K$ with MNIST as $D_u$.

| $D_d$ | $|C_d|$ | $K$ | | | | | |
|---|---|---|---|---|---|---|---|
| | | 0 | 1 | 3 | 5 | 7 | 9 |
| CIFAR-10C | 10 | 80.97 | 83.9 | 84.32 | 84.17 | 84.10 | 84.02 |
| ImageNet-R | 200 | 64.32 | 83.65 | 83.87 | 83.53 | 83.39 | 83.42 |
| ImageNet-C | 1000 | 42.05 | 48.35 | 47.17 | 48.53 | 48.37 | 47.73 |

We vary the hyperparameter $K$ which represents the number of positives and negatives chosen in Equation 6 and 7 and report the results ($Acc_{HM}$) in Table 10. The size of the feature bank $\mathcal{M}_d$ is set as $N_d = K \times C_d$. $N_d$ increases with the number of classes as well as the number of neighbours $K$. We set $K$ to be 5 in all main results reported, which corresponds to feature bank size $N_d$ of 50, 1000, 5000 respectively for the datasets CIFAR-10C, ImageNet-R and ImageNet-C respectively. In Table 10, we use the notation $K = 0$ to correspond to the case where only the reliable pseudo label loss $\mathcal{L}_{Re}$ is used. The results show that even with $K = 1$, there is a significant improvement in $Acc_{HM}$ when compared to the case where $\mathcal{L}_D, \mathcal{L}_U$ is not used ($K = 0$). On further increasing $K$, we observe improvement only for the CIFAR-10C as $D_d$, but the performance is similar for ImageNet-R and ImageNet-C for higher values of $K$ as well. Further, we investigate this observation that the performance of ROSITA is similar on significantly varying $K$ or the feature bank size. For $K = 5$, we check the average number of positives actually selected for $L_D$ in Equation 6. for each of these datasets. We find this to be $4.1, 2.5$ and $1.5$ for CIFAR-10C, ImageNet-R and ImageNet-C respectively. This agrees with the results in Table 10, where $K$ of 3, 5 works better compared to 1 as more neighbours have common pseudo label, aiding the clustering of classes of interest. For CIFAR-10C and ImageNet-R, using $K < 5$ suffices and for ImageNet-C as only 1-2 neighbours are matched for majority of reliable desired class samples, setting $K = 1$ suffices. For practical purposes, this observation suggests that the buffer size for $M_d$ can indeed be reduced based on storage budget available depending on the application and device the model is deployed on. For e.g., if the memory budget available can store only upto 1000 features, $K$ can be set flexibly depending on the number of classes of interest. For ImageNet-C with 1000 classes, $K$ can be set to 1.

### C.3 Detailed analysis of ReDUCe Loss components

We provide detailed results of Table 4, including all the five metrics in Table 11. Additionally, we visualise the histograms of the scores $s_t$ on using different combinations of the loss components of ReDUCe Loss in the Figures 6, 7, justifying their role in better discrimination of samples from $C_d$ and $C_u$.

Table 11: Detailed performance metrics analysing the ReDUCE Loss components.

| $\mathcal{L}_{Re}$ | $\mathcal{L}_D$ | $\mathcal{L}_U$ | CIFAR-10C/MNIST | | | | | ImageNet-R/MNIST | | | | |
|---|---|---|---|---|---|---|---|---|---|---|---|---|
| | | | AUC | FPR | $Acc_D$ | $Acc_U$ | $Acc_{HM}$ | AUC | FPR | $Acc_D$ | $Acc_U$ | $Acc_{HM}$ |
| ✗ | ✗ | ✗ | 91.91 | 85.04 | 60.82 | 99.77 | 75.57 | 91.27 | 91.09 | 55.67 | 99.90 | 71.50 |
| ✓ | ✗ | ✗ | 95.29 | 30.82 | 68.36 | 99.30 | 80.97 | 81.07 | 99.02 | 48.42 | 95.76 | 64.32 |
| ✗ | ✓ | ✗ | 95.23 | 28.91 | 66.93 | 98.52 | 79.71 | 87.73 | 94.67 | 51.13 | 98.34 | 67.28 |
| ✗ | ✗ | ✓ | 98.61 | 12.73 | 66.60 | 99.68 | 79.84 | 99.39 | 4.81 | 67.81 | 99.99 | 80.82 |
| ✗ | ✓ | ✓ | 99.27 | 4.15 | 67.76 | 99.73 | 80.69 | 99.48 | 4.40 | 69.38 | 99.98 | 81.92 |
| ✓ | ✓ | ✓ | 99.10 | 7.63 | 72.81 | 99.74 | 84.17 | 99.44 | 4.29 | 71.73 | 99.98 | 83.53 |

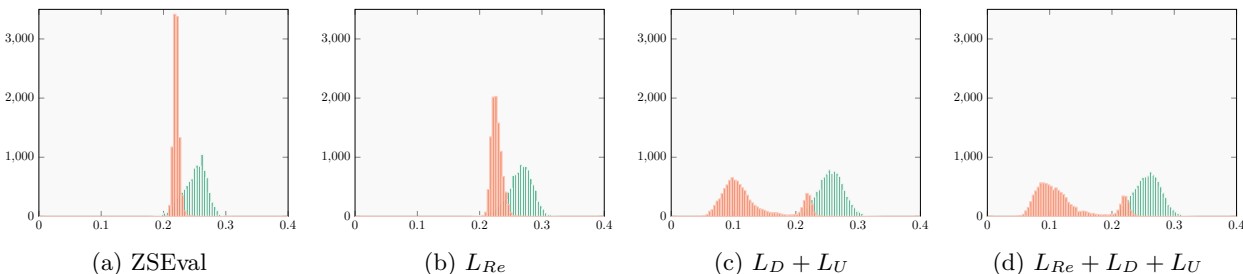

| (a) ZSEval | (b) $L_{Re}$ | (c) $L_D + L_U$ | (d) $L_{Re} + L_D + L_U$ |

Figure 6: Histograms of $C_d$ and $C_u$ class scores for ZS-Eval and on using different loss components of the proposed ReDUCe loss on CIFAR-10C/MNIST dataset with CLIP.

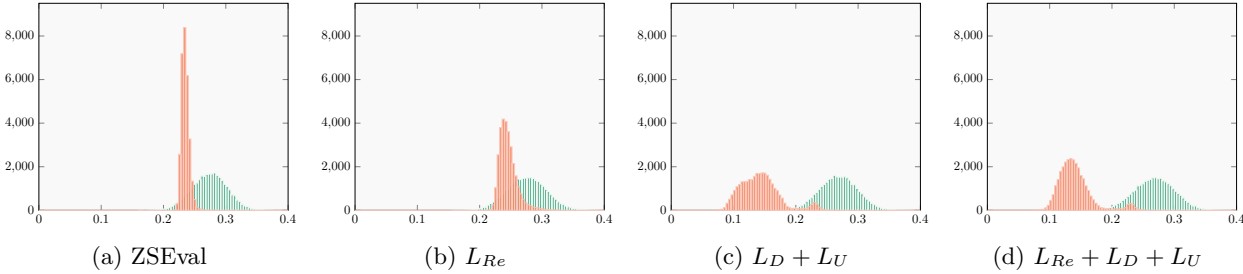

| (a) ZSEval | (b) $L_{Re}$ | (c) $L_D + L_U$ | (d) $L_{Re} + L_D + L_U$ |

Figure 7: Histograms of $C_d$ and $C_u$ class scores for ZS-Eval and on using different loss components of the proposed ReDUCe loss on ImageNet-R/MNIST dataset with CLIP.

From Figure 6 and 7, we observe that, on using just $\mathcal{L}_{Re}$, the scores of $C_d$ and $C_u$ classes still sufficiently overlap, similar to the case of ZSEval. The performance purely depends on the quality of pseudo labels of the detected reliable desired class samples. In CIFAR-10C, as there are only 10 classes and given that the performance of ZSEval in CIFAR-10C is fairly good, it ensures good quality pseudo-labels, hence resulting in overall better metrics even using $\mathcal{L}_{Re}$ as shown in Table 11. ImageNet-R dataset inherently has more confusion as it is a 200-way classification problem. This naturally could result in lower quality pseudo-labels, in turn degrading the performance compared to ZSEval. In addition, using $\mathcal{L}_{Re}$ for desired class samples that are misclassified as undesired class samples increases the FPR and results in a decrease in overall metrics compared to ZSEval. However, using $\mathcal{L}_D$ and $\mathcal{L}_U$ separates the scores $s_t$ of the samples from $C_d$ and $C_u$, resulting in two distinct peaks as seen in Figure 6 and 7, which in turn results in a significantly low FPR as reported in Table 11. Hence, the best results (Table 11) are obtained using the proposed ReDUCe loss, where

all the loss components help each other to better discriminate the desired classes $C_d$ from $C_u$ (measured by AUC, FPR) and also improving the $C_d$-way accuracy ($Acc_D$) on desired classes.

**Effect of ReDUCe loss in representation space:** Contrastive learning through ReDUCe loss influences the representation space by enhancing feature separability, improving the discriminability of desired vs. undesired classes. In addition to the accuracy metrics presented, here we compute the **inter-feature distance** between desired ($C_d$) and undesired ($C_u$) class features through the following distance metrics:

**(a) Mean Pairwise Distance** is a simple but effective way is to compute the pairwise Euclidean or cosine distance between all $C_d$ and $C_u$ samples and then take the mean:

$$d_{\text{Euclidean}} = \frac{1}{|F_d||F_u|} \sum_{f_d \in F_d} \sum_{f_u \in F_u} \|f_d - f_u\|_2; \quad d_{\text{Cosine}} = 1 - \frac{1}{|F_d||F_u|} \sum_{f_d \in F_d} \sum_{f_u \in F_u} \frac{f_d \cdot f_u}{\|f_d\|\|f_u\|} \tag{16}$$

where $F_d$ and $F_u$ are the feature banks for desired and undesired class samples, respectively.

**(b) Wasserstein Distance (Optimal Transport)** measures the optimal transport cost between the two feature distributions, making it ideal for comparing feature distributions of desired ($C_d$) and undesired ($C_u$) classes.

$$d_{\text{Wasserstein}} = \inf_{\gamma \in \Pi(F_d, F_u)} \mathbb{E}_{(f_d, f_u) \sim \gamma}[\|f_d - f_u\|] \tag{17}$$

Table 12: Inter-feature distance measures for ZSEval and ROSITA using CIFAR-10C as $D_d$.

| $D_u$ | Method | Euclidean ($\uparrow$) | Cosine ($\uparrow$) | Wasserstein ($\uparrow$) |
|---|---|---|---|---|
| MNIST | ZSEval | 0.5831 | 0.1731 | 0.5553 |
| | ROSITA | **1.2970** | **0.8623** | **1.2708** |
| SVHN | ZSEval | 0.6963 | 0.2456 | 0.6535 |
| | ROSITA | **1.0785** | **0.5918** | **1.0265** |
| TinyImageNet | ZSEval | 0.7522 | 0.2860 | 0.6583 |
| | ROSITA | **0.9939** | **0.4994** | **0.8852** |
| CIFAR-100C | ZSEval | 0.5587 | 0.1598 | 0.4265 |
| | ROSITA | **0.8194** | **0.3411** | **0.6594** |

The results clearly show that ROSITA improves feature separability across all metrics compared to ZSEval, reinforcing its effectiveness in refining the representation space for open-set recognition. The degree of separation varies with task difficulty for MNIST, the distances increase significantly, while for CIFAR-100C, the improvement is smaller. This aligns with intuition, as distinguishing between CIFAR-10C and CIFAR-100C is inherently more challenging than separating CIFAR-10C from MNIST. The quantitative results validate that contrastive adaptation enhances representation learning, but the extent of improvement depends on the complexity of the undesired dataset.

### C.4 Comparison of different $C_d$ vs $C_u$ Class identifiers for Open-set TTA

To study the role of the $C_d$ vs $C_u$ class identifiers in Open-set Single Image TTA, we experiment with three class identifiers, on five datasets as $D_d$ with MNIST as $D_u$ using CLIP backbone.

**(1) Simple thresholding based on Maximum Softmax Probability(MSP):** We set fixed thresholds $\tau_u, \tau_d$ to identify reliable samples from $C_d$ and $C_u$ classes respectively and $\tau_t$ to distinguish between $C_d$ and $C_u$ samples. We combine this class identifier with the ReDUCe loss of the proposed ROSITA framework.

**(2) Distribution Aware Filter (DAF) (Gao et al., 2024) :** We adopt the Distribution Aware Filter proposed in UniEnt (Gao et al., 2024), a very recent method on open-set TTA using CNNs, where they

model the scores $s_t$ (similarity between image feature and source prototype) as a Gaussian Mixture Model for each batch. In our case, as we do single image TTA, we use a score bank as described in Section 2.2 as a proxy for the batch of samples, to estimate the parameters of the GMM. As it is a 2-component GMM, we identify a sample as a desired class sample if the probability $\pi(x_t)$ of the sample belonging to the desired classes(component with higher mean estimated) is greater than 0.5 or vice versa. The GMM based class identifier is defined as follows:

$$\hat{y} = \begin{cases} \in C_d & \text{if } \pi(x_t) \geq 0.5 \\ \in C_u & \text{if } \pi(x_t) < 0.5 \end{cases} \tag{18}$$

We combine this class identifier with the Unified entropy objective and ReDUCe loss proposed by UniEnt (Gao et al., 2024) and our proposed ROSITA framework respectively.

**(2) Linear Discriminant Analysis (LDA) based (Li et al., 2023) :** As described in Section 2.2, we set $\tau_d$ to $\mu_d$ and $\tau_u$ to $\mu_u$ to identify reliable $C_d$ and $C_u$ samples to perform TTA. We set $\tau_t$ to $\mu_u$ to distinguish between $C_d$ and $C_u$ samples. The thresholds are estimated in an online manner using the score bank $\mathcal{S}$. The LDA based class identifier is defined as follows:

$$\hat{y} = \begin{cases} \in C_d & \text{if } s_t \geq \tau_t^* \\ \in C_u & \text{if } s_t < \tau_t^* \end{cases} \tag{19}$$

We combine this class identifier with the Unified entropy objective and ReDUCe loss proposed by UniEnt (Gao et al., 2024) and our proposed ROSITA framework respectively. The three thresholds for ReDUCe loss in Table 13 correspond to $\tau_u/\tau_t/\tau_d$ where $\tau_u$ and $\tau_d$ is used to identify reliable test samples and $\tau_t$ is used to distinguish between $C_d$ and $C_u$ samples. In the case of DAF with ReDUCe loss, we use the means $\mu_d^*$ and $\mu_*$ for the two gaussian mixture components to identify reliable samples.

Table 13: Comparison of $C_d$ vs $C_u$ class identifiers: MSP vs DAF vs LDA. The three thresholds for ReDUCe loss correspond to $\tau_u/\tau_t/\tau_d$ where $\tau_u$ and $\tau_d$ is used to identify reliable test samples and $\tau_t$ is used to distinguish between $C_d$ and $C_u$ samples. In the case of DAF with ReDUCe loss, we use the estimated means $\mu_d^*$ and $\mu_u^*$ of the two Gaussian mixture components to identify reliable samples.

| $C_d$ vs $C_s$ | Threshold | Test-time objective | $D_u$: MNIST | | | | |
| | | | C-10C | C-100C | IN-C | IN-R | VisDA |
|---|---|---|---|---|---|---|---|
| MSP | 0.4/0.6/0.8 | ReDUCe | 43.44 | 34.42 | 1.20 | 77.12 | 88.49 |
| | 0.3/0.5/0.7 | | 33.70 | 32.60 | 1.74 | 80.29 | 50.87 |
| | 0.5/0.5/0.5 | | 22.82 | 37.41 | 1.91 | 30.90 | 32.31 |
| LDA | $s_t > \tau_t$ | UniEnt | 75.62 | 48.31 | 41.53 | 71.73 | 78.09 |
| DAF | $\pi(x_t) > 0.5$ | | 79.43 | 50.12 | 46.52 | 79.30 | 86.79 |
| LDA | $\mu_u/\tau_t/\mu_d$ | ReDUCe | **84.17** | **57.34** | **48.53** | **83.53** | 90.64 |
| DAF | $\mu_u^*/0.5/\mu_d^*$ | | 83.56 | 55.37 | 48.33 | 83.32 | **90.97** |

Our key observations based on the results in Table 13 are as follows:

**Fixed vs Dynamic Thresholds:** The performance of both, DAF and LDA based class identifier is significantly better than the simple thresholding case on adaptation using ReDUCe loss. The thresholds estimated in an online manner using the score bank $\mathcal{S}$ are more reliable than fixed thresholds. The DAF and LDA-based class identifier is able to better discriminate between $C_d$ and $C_u$ samples, resulting in better performance.

**UniEnt vs ReDUCe loss:** The performance on using ReDUCe loss (with either DAF or LDA class identifier) is significantly better than using the Unified entropy objective proposed in UniEnt (Gao et al., 2024). The ReDUCe loss components aid each other to better discriminate the desired classes $C_d$ from $C_u$ (measured by AUC, FPR) and also improve the $C_d$-way accuracy ($Acc_D$) on desired classes.

**LDA vs DAF with ReDUCe loss:** The performance of LDA and DAF based class identifier perform very similarly when used in combination with ReDUCe loss. This suggests that ReDUCe loss in ROSITA is robust to the choice of a dynamically updating class identifier.

**Why is ReDUCe loss better than Unified entropy objective for Open-set TTA of VLMs?**

- Both LDA (Li et al., 2023) and DAF (Gao et al., 2024) were proposed for CNN based open-set TTA where a source model is trained on say clean data and is adapted to new domains, with the observation that the feature-prototype similarity scores $s_t$ can distinguish desired and undesired class samples. In the case of VLMs, the source model is trained on a large scale dataset and is adapted to potentially unseen/corrupted/covariate-shifted data. *The prior that the feature-prototype similarity scores $s_t$ can distinguish desired and undesired class samples does not translate to VLMs as the scores overlap significantly,* as observed in ZSEval histogram plots in Figures 6 and 7.

- In the case of CNNs, where the the initial scores are well separated and model has access to a batch of test samples at a time, UniEnt leverages this to further aid the separation of desired and undesired class samples in the batch through the UniEnt objective. In the case of VLMs, the scores are not well separated initially. This results in the means $\mu_d$ and $\mu_u$ in the case of LDA to be very close leading to misclassification of $C_d$ and $C_u$ class samples using the estimated threshold $\tau_t$. Similarly, in the case of DAF, the two components of GMM would not be very distinctive to well distinguish desired and undesired class samples. This misclassification can result in entropy minimization being applied on $C_u$ samples and entropy maximization on $C_d$ samples, which is undesirable. Employing UniEnt objective with several misclassified samples may not actually separate desired and undesired classes, as also empirically observed in Tables 1 2 3 (UniEnt has high FPR rate in general). Entropy maximization of $C_u$ samples does not explicitly enforce the separation of desired and undesired class samples in the feature space.

- The $\mathcal{L}_D$ and $\mathcal{L}_U$ loss components of ReDUCe loss explicitly enforce the separation desired and undesired class samples in the common VL latent space, while the $L_{Re}$ loss aims to only align the desired class samples to align with the text prototypes. With time, the model is adapted such that undesired class samples are away from the desired class samples and also the text prototypes. This ReDUCe loss addresses the challenges in single image open-set TTA in a holistic manner, resulting in better performance.

- On adopting UniEnt objective to single-image TTA, either entropy minimization or maximization loss would be active based on whether a test sample is identified as desired or undesired class sample, which is a limitation, as the objective cannot enforce distinction between the two types of features.

- In the case of CNNs, where the the initial scores are well separated and model has access to a batch of test samples at a time, UniEnt leverages this to further aid the separation of desired and undesired class samples in the batch through the UniEnt objective. In the case of VLMs, the scores are not well separated initially, hence the ReDUCe loss components (with the help of feature banks) acts as the driving force to better separate the desired and undesired class samples in the common latent space, resulting in lower FPR rates as a consequence.

## C.5 Extensive analysis on parameter choice for continuous adaptation of VLMs

Our initial experiments showed that updating LayerNorm parameters with simple entropy objective can effectively improve closed-set TTA performance. We illustrate this in Section 4 on CIFAR-10C dataset. Further, to justify our choice of updating LayerNorm parameters, we present the detailed experiments we conducted based on the following choices: (a) **Learnable parameters**: (1) Prompts, (2) Full network, (3) First Attention Block of ViT, (4) Last Attention Block of ViT, (5) Prompts+LayerNorm(LN), (6)LoRA Adapters (Imam et al., 2024), (6) LayerNorm parameters (Zhao et al., 2023) (b) **Datasets**: In addition to CIFAR-10C (Section 4), we experiment with ImageNet-R, a relatively large scale dataset consisting of 30,000 images from 200 classes. (c) **Optimizer**: Along with SGD, we experiment with AdamW optimizer also used in [1], with varying learning rates on both CIFAR-10C and ImageNet-R dataset.

Table 14: Accuracy on updating different parameter groups on CIFAR-10C and ImageNet-R datasets.

| Optimizer | Parameters | CIFAR-10C | | | | | ImageNet-R | | | | |
|---|---|---|---|---|---|---|---|---|---|---|---|
| | | $1e^{-6}$ | $1e^{-5}$ | $1e^{-4}$ | $1e^{-3}$ | $1e^{-2}$ | $1e^{-6}$ | $1e^{-5}$ | $1e^{-4}$ | $1e^{-3}$ | $1e^{-2}$ |
| SGD | Prompts | 73.40 | 31.04 | 12.53 | 11.18 | 10.19 | 73.97 | 74.17 | 74.71 | 25.68 | 10.63 |
| | Full | 10.48 | 10.44 | 9.99 | 10.00 | 10.01 | 14.18 | 7.19 | 0.65 | 0.65 | 0.42 |
| | First Block | 75.1 | 76.12 | 78.27 | 13.07 | 10.01 | 73.84 | 74.31 | 74.91 | 8.76 | 0.32 |
| | Last Block | 73.45 | 72.42 | 59.44 | 10.17 | 10.02 | 75.95 | 77.93 | 24.82 | 0.52 | 0.67 |
| | Prompts+LN | 73.82 | 46.77 | 24.71 | 10.24 | 10.18 | 73.76 | 75.09 | 76.35 | 28.72 | 11.74 |
| | LoRA Adapters | 73.86 | 73.90 | 75.42 | 83.15 | 12.58 | 73.51 | 73.57 | 74.22 | 77.39 | 34.83 |
| | LayerNorm | 74.35 | 76.61 | 80.41 | **84.58** | 11.69 | 74.13 | 74.35 | 75.23 | **76.92** | 33.07 |
| AdamW | Prompts | 72.40 | 18.6 | 12.83 | 10.04 | 10.08 | 74.4 | 75.17 | 27.93 | 6.82 | 4.37 |
| | Full | 10.32 | 10.03 | 10.00 | 10.00 | 9.97 | 14.83 | 0.95 | 0.28 | 0.52 | 0.66 |
| | First Block | 79.05 | 24.70 | 10.84 | 10.00 | 10.00 | 74.6 | 74.8 | 5.68 | 0.26 | 0.15 |
| | Last Block | 59.23 | 10.84 | 10.49 | 10.00 | 10.01 | 77.44 | 10.67 | 0.51 | 0.25 | 0.33 |
| | Prompts+LN | 75.01 | 72.10 | 21.92 | 13.33 | 10.01 | 74.52 | 76.45 | 12.99 | 8.87 | 5.55 |
| | LoRA Adapters | 77.64 | 81.55 | 14.01 | 10.25 | 10.02 | 74.34 | 76.14 | 18.63 | 2.26 | 0.62 |
| | LayerNorm | 76.10 | 81.57 | **85.9** | 85.27 | 10.03 | 73.96 | 75.64 | 78.28 | **78.81** | 31.47 |

**LoRA Adapters for Model adaptation:**

Following Imam et al. (2024), we employ LoRA adapters with rank 16 for multi-head self-attention layers in the vision encoder. We also vary its rank to study its sensitivity (Table 15).

Table 15: Change in HM, number of parameters for varying rank of LoRA adapters vs. LayerNorm.

| LoRA rank | 2 | 4 | 8 | 16 | **LayerNorm** |
|---|---|---|---|---|---|
| HM ($\uparrow$) | 27.49 | 34.08 | 79.24 | 83.15 | 84.58 |
| Additional params ($\downarrow$) | 110,592 | 221,184 | 442,368 | 884,736 | 0 |
| Learnable params ($\downarrow$) | 110,592 | 221,184 | 442,368 | 884,736 | 39,936 |

**(a) Sensitivity to Learning Rate:** As observed in Tables 14 and 15, LoRA-based adaptation can be effective when an appropriate learning rate is chosen. For instance, it performs well at $1e^{-3}$ with SGD and $1e^{-5}$ with AdamW. While it may require some tuning to achieve optimal performance, LoRA remains a viable option for test-time adaptation under the right hyperparameter settings.

**(b) Increased Model Complexity:** Unlike LayerNorm tuning, which does not introduce additional parameters, LoRA requires the addition of adapter modules during deployment. As shown in Table 15, the number of additional parameters scales with the rank of the LoRA adapter, increasing from 110K (rank=2) to 884K (rank=16). This added complexity can be a concern for lightweight real-time adaptation settings, especially in resource-constrained environments. A key distinction between using LoRA for training-time finetuning and test-time adaptation is that in the former, LoRA weights can be merged with the base model post-training, effectively eliminating the additional parameter overhead. However, in TTA, adaptation is a continuous process, meaning that the additional parameters cannot be merged with the backbone model. This makes LoRA-based adaptation more computationally and memory-intensive compared to LayerNorm tuning, which operates within the existing model structure.

**(c) Sensitivity to Rank Selection:** As seen in Table r8 above, the performance of LoRA is highly dependent on the rank selection. Lower-rank configurations (e.g., rank=2, 4) lead to poor adaptation, whereas higher-rank settings (e.g., rank=16) perform significantly better but at the cost of increased parameter overhead. This trade-off introduces another hyperparameter that must be carefully tuned.

While LoRA-based adaptation has been explored in recent works, our findings suggest that LayerNorm tuning remains a more efficient and robust choice for OSTTA due to its stability across learning rates, no additional parameter overhead, and suitability for continuous adaptation without requiring explicit rank selection.

# D    Additional Experiments

In addition to the results presented in the main paper, we perform additional experiments supporting the claims made and for more comprehensive understanding of the analysis presented in Section 6.

## D.1    Open Set Single Image CTTA Experiments

In addition to the Open-set CTTA experiments done on CIFAR-10C as desired class dataset, here we also experiment with CIFAR-100C as desired class dataset. We present the 15 corruptions of CIFAR-10C/CIFAR-100C sequentially as $D_d$, one sample at a time along with different datasets for $C_u$ samples, namely MNIST, SVHN, Tiny ImageNet, CIFAR-10C/CIFAR-100C and report the results in Table 16. We observe that the improvement in performance of ROSITA is agnostic to open-set scenarios including different combinations of $D_d$ (continuously changing domains) and $D_u$ datasets.

Table 16: Open-set CTTA performance for CIFAR-10C and CIFAR-100C as desired datasets.

| Method | CIFAR-10C | | | | CIFAR-100C | | | |
|---|---|---|---|---|---|---|---|---|
| | SVHN | MNIST | Tiny | C-100C | MNIST | SVHN | Tiny | C-10C |
| ZSEval | 64.33 | 64.04 | 66.50 | 58.49 | 39.00 | 36.29 | 38.41 | 35.04 |
| TPT | 64.26 | 64.03 | 66.50 | 58.47 | 39.00 | 36.24 | 38.38 | 34.45 |
| (K+1)PC | 65.13 | 62.52 | 66.93 | 57.46 | 40.64 | 37.05 | 38.23 | 34.55 |
| TDA | 66.02 | 66.44 | 67.64 | 59.44 | 40.49 | 40.35 | 39.92 | 35.42 |
| DPE | 23.36 | 50.12 | 58.96 | 35.56 | 30.75 | 19.23 | 35.85 | 27.62 |
| ROSITA | **66.86** | **65.26** | **68.89** | **59.16** | **41.64** | **38.02** | **40.44** | **36.05** |

### D.1.1    Performance on Open-set Continuously Changing Corruptions Benchmark

CCC benchmark (Press et al., 2023) was specifically introduced to assess the long-term continual adaptation behavior of TTA methods in a changing world, covering scenarios such as weather changing from foggy to rainy, day to night. We experiment with the CCC dataset where gradual domain changes are synthesized across 15 corruptions of ImageNet-C as $D_d$ and MNIST as $D_u$ for a sequence length of 300k samples. From Table 17, we observe that ROSITA consistently outperforms prior methods even in this challenging CCC dataset.

Table 17: Results on open-set CCC benchmark.

| Method | CCC/MNIST | | | | |
|---|---|---|---|---|---|
| | AUC ↑ | FPR ↓ | $Acc_D$ ↑ | $Acc_U$ ↑ | $Acc_{HM}$ ↑ |
| ZS-Eval | 88.45 | 88.24 | 18.37 | 99.53 | 31.01 |
| (K+1)PC | 95.82 | 20.16 | 20.37 | 99.97 | 33.84 |
| TDA | 87.42 | 85.83 | 20.42 | 99.35 | 33.87 |
| UniEnt | 89.90 | 82.86 | 18.99 | 99.62 | 31.90 |
| DPE | 84.68 | 87.56 | 18.47 | 99.40 | 31.16 |
| ROSITA | **96.02** | **19.96** | **21.14** | **99.11** | **34.84** |

## D.2    Varying OOD ratio

In addition to the results presented in Table 5, we perform experiments varying the OOD ratio using ImageNet-R as desired class dataset which is a relatively large scale dataset with 50,000 images from 200 classes. We use MNIST as undesired class dataset and vary the ratio of number of samples belonging to ImageNet-R and MNIST as 0.2, 0.4, 0.6, 0.8. From Table 5 and Table 18, we observe consistent improvements of ROSITA across datasets compared to the prior methods.

Table 18: Varying ratio for ImageNet-R/MNIST.

| Method | 0.2 | 0.4 | 0.6 | 0.8 |
|---|---|---|---|---|
| ZS-Eval | 65.46 | 67.13 | 69.25 | 70.77 |
| TPT | 65.67 | 67.73 | 70.12 | 71.54 |
| TPT-C | 64.83 | 64.55 | 48.97 | 63.86 |
| (K+1)PC | 78.09 | 81.08 | 81.35 | 82.61 |
| TDA | 67.90 | 71.33 | 71.54 | 71.47 |
| DPE | 66.89 | 68.47 | 69.72 | 70.87 |
| ROSITA | **82.22** | **83.32** | **83.59** | **83.84** |

### D.3 Performance of ROSITA on large Vision Language backbones

Here, in addition to CLIP ViT-B/16 (Radford et al., 2021) and MAPLE (Khattak et al., 2023) backbones, we perform experiments using large-scale Vision language backbones including CLIP ViT-L/14 by Open-AI (Radford et al., 2021) and Open-CLIP ViT-L/14 (Cherti et al., 2023) with CIFAR-10C/100C as $D_d$ and MNIST, SVHN, Tiny-ImageNet and CIFAR-100C/10C as $D_u$. From Table 19, we observe that ROSITA consistently outperforms even very recent baselines like (K+1)PC (Li et al., 2023), TDA (Karmanov et al., 2024), suggesting that the performance of ROSITA is agnostic to the choice of VL backbone.

Table 19: Comparison of ROSITA with prior methods on large scale Vision Language backbones.

| VL Backbone | Method | CIFAR-10C | | | | CIFAR-100C | | | |
|---|---|---|---|---|---|---|---|---|---|
| | | MNIST | SVHN | Tiny | C-100C | MNIST | SVHN | Tiny | C-10C |
| CLIP ViT-L/14 | ZSEval | 83.94 | 74.54 | 80.16 | 72.32 | 56.29 | 52.35 | 53.25 | 49.89 |
| | (K+1)PC | 85.43 | 80.60 | 81.65 | 71.90 | 64.14 | 55.18 | 54.53 | 47.90 |
| | TDA | 84.91 | 76.87 | 81.07 | 74.23 | 59.11 | 55.25 | 55.44 | 52.48 |
| | ROSITA | **89.46** | **83.42** | **83.61** | **75.63** | **65.41** | **60.31** | **57.55** | **54.66** |
| Open-CLIP ViT-L/14 | ZSEval | 80.64 | 76.90 | 84.10 | 75.40 | 62.96 | 59.38 | 61.10 | 59.57 |
| | (K+1)PC | 85.84 | 82.42 | 84.99 | 75.70 | 70.14 | 63.36 | 60.56 | 59.43 |
| | TDA | 80.57 | 77.92 | 84.60 | 75.79 | 64.90 | 60.70 | 62.01 | 61.20 |
| | ROSITA | **89.04** | **82.98** | **85.55** | **76.62** | **70.54** | **63.84** | **62.57** | **61.84** |

### D.4 Experiments using different corruption types

To evaluate the robustness of our method across different domains, we do additional experiments with *impulse noise*, *motion blur* and *jpeg compression* corruptions from the corruption categories per-pixel noise, blurring and digital transforms respectively and report the results here. From Table 20, Table 21 and Table 22, we observe that ROSITA either outperforms or at par with prior methods in most cases even on using the same set of hyperparameters. This demonstrates its robustness across a variety of corruption types.

Table 20: Results on CIFAR-10C/100C (Impulse Noise) as $D_d$ with other $D_u$.

| | | Method | MNIST | | | SVHN | | | Tiny-ImageNet | | | CIFAR-100C/10-C | | |
|---|---|---|---|---|---|---|---|---|---|---|---|---|---|---|
| | | | AUC ↑ | FPR ↓ | HM ↑ | AUC ↑ | FPR ↓ | HM ↑ | AUC ↑ | FPR ↓ | HM ↑ | AUC ↑ | FPR ↓ | HM ↑ |
| C-10C (Impulse noise) | CLIP | ZS-Eval | 86.34 | 97.77 | 57.67 | 84.40 | 79.43 | 56.80 | 88.97 | 31.86 | 61.11 | 78.61 | 67.88 | 54.40 |
| | | TPT | 86.35 | 97.83 | 59.80 | 84.43 | 79.52 | 58.97 | 88.96 | 31.99 | 64.48 | 78.60 | 68.24 | 56.38 |
| | | TPT-C | 62.34 | 87.66 | 39.90 | 59.71 | 83.29 | 35.42 | 81.30 | 38.59 | 37.02 | 66.22 | 89.92 | 30.86 |
| | | ROSITA | 98.87 | 9.43 | 71.31 | 82.85 | 56.82 | 61.03 | 93.36 | 21.47 | 64.47 | 78.69 | 69.45 | 57.87 |
| | MAPLE | ZS-Eval | 91.10 | 76.09 | 64.01 | 92.98 | 45.28 | 63.66 | 83.77 | 44.44 | 60.93 | 79.22 | 65.26 | 57.49 |
| | | PAlign | 91.10 | 76.01 | 65.76 | 93.00 | 45.13 | 65.28 | 83.78 | 44.42 | 62.75 | 79.22 | 65.24 | 58.80 |
| | | PAlign-C | 92.43 | 63.39 | 63.61 | 92.92 | 45.86 | 64.50 | 83.36 | 45.74 | 60.83 | 79.30 | 64.47 | 57.00 |
| | | ROSITA | 98.80 | 6.10 | 71.79 | 95.39 | 28.06 | 72.13 | 84.92 | 45.35 | 65.30 | 80.49 | 65.57 | 61.63 |
| C-100C (Impulse noise) | CLIP | ZS-Eval | 70.48 | 99.17 | 25.08 | 51.12 | 96.44 | 25.69 | 59.90 | 67.18 | 27.72 | 53.51 | 94.97 | 25.16 |
| | | TPT | 70.56 | 99.17 | 25.26 | 51.21 | 96.38 | 26.26 | 59.91 | 67.09 | 28.36 | 53.53 | 94.94 | 25.63 |
| | | TPT-C | 57.65 | 93.07 | 8.71 | 79.28 | 57.07 | 2.74 | 90.40 | 22.60 | 5.71 | 50.26 | 95.34 | 3.26 |
| | | ROSITA | 36.47 | 99.96 | 20.98 | 24.17 | 99.77 | 18.99 | 53.57 | 79.85 | 26.27 | 58.02 | 94.15 | 29.75 |
| | MAPLE | ZS-Eval | 69.29 | 89.49 | 33.66 | 81.03 | 73.94 | 34.99 | 49.57 | 84.71 | 26.09 | 57.84 | 94.44 | 29.34 |
| | | PAlign | 69.31 | 89.54 | 33.74 | 81.05 | 73.98 | 34.96 | 49.60 | 84.63 | 25.81 | 57.84 | 94.48 | 29.53 |
| | | PAlign-C | 71.14 | 73.63 | 34.38 | 82.08 | 68.24 | 35.11 | 47.27 | 87.87 | 25.95 | 57.79 | 93.54 | 30.73 |
| | | ROSITA | 95.38 | 8.80 | 43.06 | 80.25 | 41.21 | 34.88 | 42.77 | 97.15 | 19.70 | 49.73 | 96.72 | 12.62 |

Table 21: Results on CIFAR-10C/100C(Motion blur) as $D_d$ with other $D_u$.

| | | Method | MNIST | | | SVHN | | | Tiny-ImageNet | | | CIFAR-100C/10-C | | |
|---|---|---|---|---|---|---|---|---|---|---|---|---|---|---|
| | | | AUC ↑ | FPR ↓ | HM ↑ | AUC ↑ | FPR ↓ | HM ↑ | AUC ↑ | FPR ↓ | HM ↑ | AUC ↑ | FPR ↓ | HM ↑ |
| C-10C (Motion blur) | CLIP | ZS-Eval | 97.73 | 2.75 | 73.69 | 96.40 | 18.34 | 73.82 | 95.25 | 15.75 | 74.27 | 79.57 | 70.08 | 62.86 |
| | | TPT | 97.72 | 2.68 | 74.15 | 96.39 | 18.16 | 74.42 | 95.23 | 15.72 | 75.03 | 79.56 | 69.86 | 63.25 |
| | | TPT-C | 80.73 | 86.28 | 63.74 | 62.09 | 62.52 | 42.19 | 80.76 | 51.66 | 48.04 | 55.66 | 97.04 | 37.53 |
| | | ROSITA | 99.90 | 0.04 | 81.87 | 96.50 | 21.55 | 77.47 | 96.58 | 13.65 | 77.44 | 82.03 | 65.95 | 66.96 |
| | MAPLE | ZS-Eval | 96.52 | 18.33 | 78.68 | 97.08 | 14.78 | 78.15 | 88.45 | 33.15 | 71.19 | 84.00 | 57.94 | 66.93 |
| | | PAlign | 96.51 | 18.37 | 78.92 | 97.08 | 14.82 | 78.38 | 88.45 | 33.13 | 71.73 | 83.99 | 57.99 | 67.15 |
| | | PAlign-C | 97.17 | 13.47 | 78.49 | 96.89 | 15.87 | 78.09 | 88.80 | 32.94 | 72.09 | 84.29 | 56.80 | 67.40 |
| | | ROSITA | 98.49 | 10.01 | 83.26 | 92.61 | 44.87 | 78.93 | 87.48 | 38.23 | 73.24 | 84.27 | 57.60 | 70.67 |
| C-100C (Motion blur) | CLIP | ZS-Eval | 93.08 | 58.92 | 48.17 | 83.63 | 81.33 | 46.04 | 79.34 | 53.56 | 48.53 | 64.03 | 91.54 | 41.63 |
| | | TPT | 93.06 | 59.87 | 48.18 | 83.61 | 81.56 | 45.54 | 79.29 | 53.76 | 48.26 | 64.02 | 91.63 | 41.25 |
| | | TPT-C | 66.77 | 98.77 | 19.96 | 29.69 | 99.94 | 11.39 | 69.25 | 62.87 | 17.10 | 53.22 | 94.57 | 13.59 |
| | | ROSITA | 98.93 | 6.79 | 55.49 | 89.39 | 37.86 | 48.50 | 90.20 | 31.61 | 55.05 | 65.30 | 91.59 | 42.54 |
| | MAPLE | ZS-Eval | 81.21 | 80.28 | 45.66 | 89.04 | 60.73 | 46.98 | 60.84 | 80.63 | 40.60 | 64.01 | 90.18 | 42.30 |
| | | PAlign | 81.20 | 80.52 | 44.52 | 89.03 | 61.01 | 45.76 | 60.84 | 80.64 | 40.03 | 64.01 | 90.26 | 41.26 |
| | | PAlign-C | 82.72 | 68.08 | 49.92 | 90.48 | 53.83 | 51.87 | 62.00 | 82.85 | 41.66 | 64.47 | 89.05 | 43.58 |
| | | ROSITA | 97.12 | 7.78 | 57.30 | 85.13 | 56.16 | 49.89 | 63.85 | 80.20 | 42.65 | 62.55 | 94.62 | 41.54 |

Table 22: Results on CIFAR-10C/100C(JPEG Compression) as $D_d$ with other $D_u$.

| | | Method | MNIST | | | SVHN | | | Tiny-ImageNet | | | CIFAR-100C/10-C | | |
|---|---|---|---|---|---|---|---|---|---|---|---|---|---|---|
| | | | AUC ↑ | FPR ↓ | HM ↑ | AUC ↑ | FPR ↓ | HM ↑ | AUC ↑ | FPR ↓ | HM ↑ | AUC ↑ | FPR ↓ | HM ↑ |
| C-10C (JPEG) | CLIP | ZS-Eval | 68.16 | 100.00 | 53.92 | 67.04 | 99.93 | 55.69 | 79.44 | 65.02 | 59.66 | 73.65 | 85.60 | 56.30 |
| | | TPT | 68.07 | 100.00 | 54.16 | 66.97 | 99.93 | 56.06 | 79.37 | 65.11 | 60.09 | 73.64 | 85.58 | 56.87 |
| | | TPT-C | 68.28 | 99.37 | 53.12 | 54.76 | 98.97 | 35.64 | 66.70 | 72.20 | 39.02 | 59.82 | 94.78 | 32.78 |
| | | ROSITA | 81.83 | 58.81 | 60.34 | 82.85 | 61.38 | 61.87 | 95.06 | 15.84 | 67.87 | 71.19 | 86.62 | 51.98 |
| | MAPLE | ZS-Eval | 95.15 | 33.39 | 69.72 | 95.96 | 22.02 | 69.73 | 86.64 | 36.79 | 65.68 | 79.26 | 68.19 | 60.10 |
| | | PAlign | 95.13 | 33.57 | 69.62 | 95.95 | 22.01 | 69.31 | 86.63 | 36.82 | 65.62 | 79.26 | 68.18 | 59.86 |
| | | PAlign-C | 96.53 | 20.14 | 70.50 | 95.94 | 21.51 | 70.01 | 87.38 | 35.07 | 66.42 | 79.85 | 66.17 | 61.11 |
| | | ROSITA | 99.28 | 5.71 | 76.74 | 95.54 | 29.06 | 72.86 | 89.88 | 31.12 | 68.78 | 80.69 | 61.64 | 62.23 |
| CIFAR-100C (JPEG) | CLIP | ZS-Eval | 50.88 | 100.00 | 32.27 | 39.25 | 100.00 | 26.41 | 48.65 | 95.60 | 29.92 | 53.51 | 95.59 | 32.48 |
| | | TPT | 50.78 | 100.00 | 32.38 | 39.18 | 100.00 | 26.48 | 48.55 | 95.60 | 29.86 | 53.49 | 95.57 | 32.70 |
| | | TPT-C | 12.11 | 100.00 | 3.32 | 10.05 | 99.98 | 2.45 | 63.07 | 90.01 | 9.49 | 52.23 | 95.05 | 6.33 |
| | | ROSITA | 29.10 | 100.00 | 22.83 | 35.58 | 99.94 | 23.50 | 50.76 | 94.76 | 31.64 | 53.96 | 96.18 | 30.39 |
| | MAPLE | ZS-Eval | 78.86 | 80.60 | 37.60 | 87.72 | 61.14 | 39.18 | 58.31 | 80.75 | 34.03 | 54.50 | 95.49 | 34.02 |
| | | PAlign | 78.82 | 80.92 | 36.62 | 87.69 | 61.37 | 38.01 | 58.29 | 80.79 | 33.17 | 54.49 | 95.52 | 32.96 |
| | | PAlign-C | 81.85 | 63.37 | 40.87 | 89.96 | 49.09 | 41.89 | 59.33 | 81.48 | 33.84 | 53.82 | 95.17 | 33.28 |
| | | ROSITA | 97.68 | 7.87 | 46.51 | 92.14 | 34.44 | 42.71 | 66.63 | 75.00 | 37.43 | 51.33 | 96.68 | 25.41 |

# E   Failure Case Analysis

Our findings indicate that while ROSITA outperforms baselines, it struggles in cases where undesired classes are highly similar to desired ones (e.g., CIFAR-10C vs. CIFAR-100C), leading to a higher false positive rate (FPR). Harder datasets like ImageNet-C pose additional challenges as the Zero-shot classification accuracy of CLIP itself is poor due to severe domain shift and increased number of desired classes. Towards the goal of studying dataset specific challenges, we study the two major components of ReDUCe loss by measuring the following metrics:

**(1) kNN Retrieval accuracy:** We compute the average number of correctly matched neighbors $K^+$ in $\mathcal{L}_D$ (Equation 6), where the pseudo-label $y^+$ of the retrieved neighbors matches the test sample's pseudo-label $y_t$.

**(2)Pseudo label accuracy of Reliable samples:** We evaluate the pseudo-label accuracy of the reliable samples, as only these are stored in the feature bank for kNN retrieval and model adaptation. Higher accuracy of these reliable samples directly benefits adaptation.

Table 23: Analysis on quality of Reliable samples.

| Metric | CIFAR-10C | ImageNet-R | ImageNet-C |
|---|---|---|---|
| Average $K^+$ in $\mathcal{L}_D$ | 4.1 | 2.5 | 1.5 |
| Accuracy of reliable desired class samples (%) | 88.7 | 86.40 | 43.74 |
| Accuracy of reliable undesired class samples (%) | 99.75 | 100 | 99.95 |

For $\mathcal{L}_D$, the average number of correctly matched neighbors aligns with dataset difficulty; CIFAR-10C has the highest $K^+$ (4.1), while ImageNet-C has the lowest (1.5), reflecting the increased complexity of retrieval in more challenging datasets. Additionally, the high accuracy of reliable desired ($\geq 86\%$ for CIFAR-100C and ImageNet-R) and undesired (nearly 100%) class samples ensures that the retrieved neighbors serve as strong supervisory signals for adaptation. Since $\mathcal{L}_D$ relies only on correctly matched neighbors, the adaptation process remains robust, effectively leveraging reliable samples regardless of dataset difficulty. However, there is still significant scope of improvement of desired class accuracy in harder datasets like ImageNet-C, where CLIP accuracy itself is still poor, given the severe domain shift and the 1000-way classification task, making it very challenging. The proposed ReDUCe loss in ROSITA primarily focuses on separating desired and undesired class samples. We use simple reliable pseudo label loss $\mathcal{L}_{Re}$ and clustering objective $\mathcal{L}_D$ that aims towards improving desired class accuracy $Acc_D$. More sophisticated methods can be employed in addition to this to improve $Acc_D$.

# F   Broader Impact Concerns

Our work introduces **Open-set Single-image Test-Time Adaptation (OSTTA)** for Vision-Language Models (VLMs), a novel and realistic problem studied for the first time. While VLMs are powerful for open-world recognition, real-world deployment benefits from the ability to recognize unknown objects and adapt effectively. Our study provides a strong baseline for evaluating their robustness under distribution shifts. While this is a step forward, some considerations are to be aware of before practical deployment. In healthcare, careful adaptation is crucial to avoid misinterpretations; in surveillance, fairness needs attention; and in autonomous systems, ensuring reliability is key. That said, our method is designed with privacy in mind as it does not store images, only extracted features, minimizing risks. Future work could explore safeguards against potential adversarial attacks, but overall, we see this as a positive step toward making VLMs more adaptable and reliable in diverse settings.

