# OpenReview forum: "Efficient Open Set Single Image Test Time Adaptation of Vision Language Models"
_TMLR — Accepted by TMLR_

### Review · Reviewer_hu7J · 2025-02-10

**Summary Of Contributions:**

This work explores and identifies a special case of test-time adaptation (TTA), called open-set TTA, using a single image sample. Existing variants of TTA do not cater to realistic setups where only a single test image is available during inference. To provide a standardized setup, this work proposes a benchmark for this problem setting, called OSTTA. Secondly, a new learning-based OSTTA framework for VLMs is proposed, called ROSITA. The main formulation of ROSITA utilizes desired samples (from known classes) and undesired samples (from unknown classes) to adapt VLMs at test time using the proposed ReduCE loss. ReduCE utilizes contrastive learning and pseudo-label-based objectives to continuously adapt the model for test-time images. Specifically, to gather the positive and negative samples for the contrastive loss, separate memory feature banks are used, which are updated/collected in an online fashion during the inference lifecycle. Finally, it is shown that tuning the layer-norm leads to improved TTA performance compared to other fine-tuning configurations. Detailed experiments are conducted, and the performance of the proposed framework is thoroughly compared with previous leading methods. The paper concludes with extensive ablation studies to motivate the design choices of the proposed framework.

**Audience:**

Yes

**Broader Impact Concerns:**

Discussion on the broader impact concerns is not provided in the manuscript.

**Claims And Evidence:**

Yes

**Requested Changes:**

I have added my comments in the weaknesses section of the review. It is requested that the authors provide a clear response that addresses these issues, which is crucial for the overall positive review decision.

I am looking forward to the response from the authors!

**Strengths And Weaknesses:**

**Strengths**
- The formulation of single-image based TTA adaptation with open-set constraints is a valid problem setting and closely mirrors the realistic environment that foundational models would face when deployed in real-world settings.
- The idea of utilizing a two-step inference procedure and the use of contrastive loss in the proposed ROSITA setup is motivating.
- The study regarding VLM fine-tuning strategies at test time is valuable and can serve as a clear reference for anyone pursuing fine-tuning of VLMs at test time.
- Overall, extensive experiments have been provided, and the proposed solution leads to improved results compared to baselines and existing leading methods.
- Important ablation studies have been conducted in a timely manner, and main design choices are properly motivated using analysis experiments.

**Weaknesses**
- My major concern is the use of memory banks in the proposed adaptation framework. Memory banks indirectly mean storing the exact samples (albeit in latent space). This raises many questions, e.g., why not directly use batch samples? Therefore, this framework might violate the regulation that it requires only a single test image for adaptation. This is a very critical point, and this work must highlight/resolve this aspect in the framework.
- As the method is continuous, it is still not clear how the performance varies for the initial test samples when the memory features are not yet updated (or not present) since no samples have been seen at the beginning.
- Additionally, the method strongly relies on the quality/difficulty of the initial test samples. In my understanding, if initial samples are more tricky/difficult, then the resulting feature banks will have noisy estimates and may even degrade performance for future samples. Similarly, if initial test samples are easy to predict correctly, the method will be more effective for future test image predictions. Unfortunately, these important investigations are not studied/highlighted in the current manuscript. It would be very important to study the impact of such cases for the proposed framework.
- Ablation studies on the memory banks module should be included in the manuscript.
- Although this work has presented an in-depth study for evaluating various fine-tuning strategies, I believe it is important to also discuss the effect of LoRA adaptation strategies for test-time adaptation. I understand it is not the most important part, but as a contribution listed in your manuscript, it would be good to fully cover all aspects of this direction. Lastly, there are recent works that explore TTA using LoRA [1], which can be a good reference for discussion.

- Based on the results, it would be good to have an analysis of failure cases and qualitative results, especially on challenging datasets that contain many classes. This would be a good resource for follow-up works building on the proposed method.

[1] Test-Time Low Rank Adaptation via Confidence Maximization for Zero-Shot Generalization of Vision-Language Models



**Minor comments**
- There are a few spacing errors in the writing. E.g., TPT citation on page 2.
- As referenced by this work, there exists open-set TTA methods (vision-only) and benchmarks but they might not conform to the single-image case. Does the proposed benchmark use the same datasets and benchmarks as these previous open-set works? Actually, it would be good to have a discussion on the benchmarks, and if they can be unified together (because only exception is the use of single image) for easy benchmarking process.

---

> ### Author Response · Authors · 2025-02-23
> **Response to Reviewer hu7J (1/6)**
>
> We thank **Reviewer hU7J** for appreciating the strengths of the work, including the problem formulation, the motivation for the proposed ROSITA framework, extensive experiments, and ablation studies. Here, we address the concerns raised.
>
>
>
>
> >### **Use of memory banks**
>
>
> We acknowledge the reviewer’s concern regarding the use of memory banks and clarify that our approach remains faithful to the single-image test-time adaptation paradigm. Several prior works, including TDA[3] and DPE[4], have also employed feature banks to aid single-image TTA, reinforcing that their usage does not equate to batch processing. Not only in single image TTA, even open-set methods based on vision-only backbones like (K+1)PC[5] operating on test batches also leverage memory banks to aid open-set adaptation.
>
>
> Specifically, our adaptation framework still performs backpropagation on a single test image at a time. The feature bank is only used to aid the clustering process, not to accumulate samples for batch-wise updates. This distinction is critical because our framework aligns with real-world scenarios where real-time inference is required, making it infeasible to store and accumulate test samples for batch adaptation.
>
>
> Additionally, storing features is fundamentally different from storing raw images, both in terms of storage overhead and privacy implications. Feature representations are significantly more lightweight (typically a few megabytes, as shown in Table 7), whereas methods that introduce additional layers for fine-tuning can impose higher storage and computational costs. While inversion techniques exist to recover images from features, such processes are non-trivial and require sophisticated reconstruction methods. We operate under the assumption that maintaining a small feature bank is a reasonable tradeoff, particularly given its efficiency in comparison to more compute-intensive adaptation strategies.
>
>
> >### **Performance on initial test samples**
>
>
> Here, we analyze the performance of ROSITA in early time steps, reporting the HM on test samples seen until time $t$.
>
>
> **Table r1:** Performance (HM) of ROSITA and ZS-Eval for initial samples. We also report the number of features collected in the desired and undesired class memory banks $M_d$ and $M_u$, respectively for CIFAR-10C/MNIST dataset.
>
>
> | $t$   | $M_d$ | $M_u$ | ROSITA | ZS-Eval |
> |-------|-------|-------|--------|---------|
> | 50    | 16    | 11    | 78.84  | 78.28   |
> | 100   | 27    | 21    | 79.91  | 78.42   |
> | 150   | 34    | 35    | 78.65  | 77.75   |
> | 200   | 46    | 50    | 79.44  | 78.75   |
> | 250   | 50    | 50    | 79.01  | 78.46   |
> | 500   | 50    | 50    | 77.82  | 77.22   |
> | 750   | 50    | 50    | 76.69  | 75.86   |
> | 1000  | 50    | 50    | 75.34  | 74.15   |
> | 2500  | 50    | 50    | 78.53  | 74.34   |
> | 5000  | 50    | 50    | 81.87  | 75.26   |
> | 10000 | 50    | 50    | 83.90  | 75.03   |
> | 20000 | 50    | 50    | 84.17  | 75.57   |
>
>
> The performance of ROSITA is similar or only marginally better than ZS-Eval, as the memory banks are still being populated. This is expected since test-time adaptation (TTA) is applied only to desirable samples, which make up roughly half of the test samples seen by time $t$. Additionally, only the **LayerNorm** parameters of the vision encoder are updated with a small learning rate, ensuring **gradual and stable adaptation**. Consequently, ROSITA's early performance remains close to ZS-Eval, and accuracy metrics in this phase are less stable due to the limited number of samples. However, as more samples are processed and the **ReDUCe loss** guides adaptation, the distinction between desired and undesired class scores becomes more pronounced (Fig. 3c). After sufficient updates, ROSITA's performance surpasses ZS-Eval significantly, especially beyond $t = 2500$.
>
>
>
> **Performance of ROSITA with time:**
>
> To study this aspect (Research Question 5), we analyze how ROSITA's scores $s_t$ and accuracy metrics evolve over time (Fig. 3c, 3d). Initially ($ t < 2500 $), the scores for $C_d$ and $C_u$ samples overlap significantly, causing ROSITA's performance to resemble ZSEval. The threshold $\tau_t$ correctly identifies most $C_u$samples ($Acc_S \approx 100\% $) but misclassifies several $C_d$ samples, leading to low $C_d$-way accuracy ( $Acc_D$ ). As adaptation progresses with the ReDUCe loss, the model better separates $C_d$ and $C_u$, improving classification performance $(Acc_D$ and $Acc_{HM}$) beyond $t > 2500$.
> The early instability ($ t < 1500 $) arises due to: **(i) Initial learning**, where the feature banks are still being populated; and **(ii) Small sample size**, since accuracy is computed on only $t$ samples, leading to oscillations, especially for $t < 500$.

---

> ### Author Response · Authors · 2025-02-23
> **Response to Reviewer hu7J (2/6)**
>
> >### **Impact of the quality of initial test samples**
>
> It is indeed true that the performance of the proposed method relies on the quality of test samples used for model adaptation. We address this by choosing only reliable test samples to populate the feature banks to prevent the noisy adaptation process. We investigate the role of the quality of test samples in determining the final performance of ROSITA here:
>
>
> **(a) Reliable samples are indeed good quality samples:** In Research Question (4), we study the role of reliable samples. We report the results (Table 6 in the paper) here for ease of reading.
>
>
> **Table r2:** Role of reliable sample in ROSITA
>
>
> | Samples for TTA       | CIFAR-10C | CIFAR-100C | ImageNet-C | ImageNet-R | VisDA     | Average   |
> |-----------------------|-----------|------------|------------|------------|-----------|-----------|
> | All test samples      | **84.99** | 55.16      | 44.05      | 83.28      | **91.24** | 71.74     |
> | Only reliable samples | 84.17     | **57.34**  | **48.53**  | **83.53**  | 90.64     | **72.84** |
>
>
> In the above table, the first row corresponds to the case where all test samples are used for model adaptation, and the second row corresponds to the case where or only reliable test samples ($s_t>\mu_d$ or $s_t<\mu_u$) for model adaptation as described in the proposed framework (Section 3).  The number of classes and the severity of domain shift are two of the factors characterizing the difficulty of a classification task. The results in Table 6 suggest that, for datasets like CIFAR-10C (10 classes) and VisDA (12 classes) having fewer classes of interest, using all samples for TTA can be helpful. The performance is comparable for ImageNet-R for both cases, which can be attributed to good zero-shot accuracy. On the other hand, for CIFAR-100C and ImageNet-C, due to the inherent confusion arising due to the large number of classes and high severity in domain shift, using all test samples, including non-reliable samples($μ_u < s_t < μ_d$), can adversely affect the adaptation process. Being selective about the test samples is crucial for effective TTA. In a real world test time adaptation scenario, where we have no prior information about the difficulty of the classification task, in terms of severity of domain shift and class confusion, it is desirable to only use reliable samples for model updates.
>
>
> **(b) Role of the quality of samples in the initial phase:** We consider the two cases: (i) All test samples are used for model adaptation; (ii) Only reliable samples are used for model adaptation. For both cases, we report the binary classification metrics (TPR, FPR, TNR, FNR) of the LDA class identifier, which recognizes a test sample to belong to the desired or undesired class. We report these metrics only for the samples used for TTA in the initial phase ($t=1000$).
>
>
> **Table r3:** Detailed analysis of reliable samples on CIFAR-100C/MNIST.
>
>
> | Samples for TTA       | No. of samples for TTA | TPR$\uparrow$ | FPR$\downarrow$ | TNR$\uparrow$ | FNR$\downarrow$ | $Acc_D$      | $Acc_U$ | $Acc_{HM}$ |
> |-----------------------|------------------------|---------------|-----------------|---------------| ----------------|--------------|---------|------------|
> | All test samples      | 1000/1000              |  57.92        | 17.23         | 82.83         | 41.92           | 35.07        | 82.83   | 49.27      |
> | Only reliable samples | 461/1000               |  57.89        | 0.44          | 99.57         | 41.20           | 33.47        | 94.61   | 49.44      |
> | All test samples      | 20000/20000            |  75.49        | 6.86          | 93.14         | 24.51           | 39.18        | 93.14   | 55.16      |
> | Only reliable samples | 10645/20000            |  91.08        | 0.03          | 99.75         | 6.95            | 40.63        | 97.41   | 57.34      |
>
>
>
>
> The results indicate that using all test samples for adaptation leads to higher FPR and FNR, as many unreliable samples introduce noise into the adaptation process. In contrast, adapting only on reliable samples significantly reduces FPR (from 17.23\% to 0.44\% at  $t = 1000$, and from 6.86\% to 0.03\% finally at $t = 20000$ ), leading to a more stable adaptation process. This reduction in FPR and FNR translates to better accuracy metrics ($Acc_D$, $Acc_U$ and $Acc_{HM}$), confirming that selective adaptation prevents undesirable model drift.
>
>
> Additionally, only 53.25% (10645/20000) of test samples are selected as reliable, which are actually used for adaptation, hence reducing computational overhead while still maintaining superior performance. This efficiency is particularly valuable in real-world settings, where computational constraints make adapting every test sample impractical. Thus, **selecting reliable (good quality) samples not only enhances adaptation quality but also improves efficiency, making the proposed framework more effective for real-world TTA scenarios.**

---

> ### Author Response · Authors · 2025-02-23
> **Response to Reviewer hu7J (3/6)**
>
> In addition to the above analysis, we also randomly shuffle the data, synthesizing five data streams to observe the robustness of the proposed method. We compute the mean and variance of the HM across the five test streams and report the results here:
>
>
> **Table r4:** Performance (Mean and Standard deviation of $Acc_{HM}$ for 5 random test streams ) of ROSITA for CIFAR-10/100C as desired class dataset $D_d$ with 4 other datasets as undesired class dataset $D_u$.
>
>
> | $D_d$\ $D_u$ | MNIST         | SVHN          | Tiny          | CIFAR-100/10C |
> |--------------|---------------|---------------|---------------|---------------|
> | CIFAR-10C    | 84.07 ± 0.023 | 78.90 ± 0.038 | 80.10 ± 0.014 | 69.44 ± 0.018 |
> | CIFAR-100C   | 57.09 ± 0.041 | 47.90 ± 0.047 | 55.95 ± 0.051 | 48.10 ± 0.024 |
>
>
> We observe that the variance in the performance of ROSITA is very low, reinforcing the robustness of the proposed method for different shuffled datasets and augmentations created.

---

> ### Author Response · Authors · 2025-02-23
> **Response to Reviewer hu7J (4/6)**
>
> >### **Ablation study on memory banks**
>
> Thanks for the suggestion. We have indeed performed the ablation study on memory banks, in terms of the parameter $K$ in Appendix C.2. The size of the feature bank depends on $K$, the number of neighbours (Table 7), which we set to 5. The performance of ROSITA for varying $K$ (and hence the size of the memory bank) is reported in Table 10. The results show that even
> with $K=1$, there is a significant improvement in $Acc_{HM}$ when compared to the case where $L_D$, $L_U$ is not used ($K = 0$). On further increasing $K$ in the range 3 to 9, the performance does not vary much, suggesting that choosing multiple positives and negatives plays a crucial role, but is not sensitive to the choice of $K$. We investigate this observation that the performance of ROSITA is similar to significantly varying $K$ or the feature bank size. For $K=5$, we check the average number of positives actually selected for $L_{D}$ in Eqn.(6) for each of these datasets. We find this to be $4.1, 2.5$, and $1.5$ for CIFAR-10C, ImageNet-R and ImageNet-C, respectively. This agrees with the results in Table 7 where $K$ of $3, 5$ works better compared to 1 as more neighbours have common pseudo label, aiding the clustering of classes of interest. For CIFAR-10C and ImageNet-R, using $K<5$ suffices, and for ImageNet-C as only $1$ or $2$ neighbours are matched for the majority of reliable desired class samples, setting $K=1$ suffices. For practical purposes, this observation suggests that the buffer size for $M_d$ can indeed be reduced based on the storage budget available depending on the application and device the model is deployed on.
> For e.g., if the memory budget available can store only up to 1000 features, $K$ can be set flexibly depending on the number of classes of interest. For ImageNet-C with 1000 classes, $K$ can be set to 1.
>
>
> **Table r5:** Accuracy on varying  $K$.
>
>
> | Dataset                 |   0   |   1   |   2   |   3   |   4   | 5     |
> |-------------------------|-------|-------|-------|-------|-------|-------|
> |CIFAR-10C    | 80.97 | 83.90 | 84.32 | 84.17 | 84.10 | 84.02 |
> |ImageNet-R   | 64.32 | 83.65 | 83.87 | 83.53 | 83.39 | 83.42 |
> |ImageNet-C   | 42.05 | 48.35 | 47.17 | 48.53 | 48.37 | 47.73 |
>
>
> >### **LoRA Adapters for model adaptation**
>
> We appreciate the reviewer’s suggestion to discuss LoRA-based adaptation further. We extend the analysis in Section 4 by performing LoRA adaptation. Following [1], we employ LoRA adapters with rank 16 for multi-head self-attention layers in the vision encoder of CLIP. We perform TTA on CIFAR-10C using SGD and AdamW optimizers. We also vary the rank of LoRA adapters to study its sensitivity to the choice of the rank. We report the results below:
>
>
> **Table r6:** Accuracy on CIFAR-10C with SGD optimizer and varying learning rates.
>
>
> | Method      | #params   | 1e-6  | 1e-5  | 1e-4  | 1e-3      | 1e-2    |
> |-------------|-----------|-------|-------|-------|-----------|---------|
> | Prompts     | 2048      | 73.4  | 31.04 | 12.53 | 11.18     | 10.19   |
> | Full        | 124436481 | 10.48 | 10.44 | 9.99  | 10.0      | 10.01   |
> | First Block | 7097088   | 75.10 | 76.12 | 78.27 | 13.07     | 10.01   |
> | Last Block  | 7097088   | 73.45 | 72.42 | 59.44 | 10.17     | 10.02   |
> | Prompts+LN  | 41984     | 73.82 | 46.77 | 24.71 | 10.24     | 10.18   |
> | LoRA        | 884736    | 73.86 | 73.90 | 75.42 | 83.15     | 12.58   |
> | LayerNorm   | 39936     | 74.35 | 76.61 | 80.41 | **84.58** | 11.69   |
>
>
> **Table r7:** Accuracy on CIFAR-10C with AdamW optimizer and varying learning rates.
>
>
> | Method      | 1e-6  | 1e-5  | 1e-4     | 1e-3  | 1e-2  |
> |-------------|-------|-------|----------|-------|-------|
> | Prompts     | 72.4  | 18.6  | 12.83    | 10.04 | 10.08 |
> | Full        | 10.32 | 10.03 | 10       | 10    | 9.97  |
> | First Block | 79.05 | 24.7  | 10.84    | 10    | 10    |
> | Last Block  | 59.23 | 10.84 | 10.49    | 10    | 10.01 |
> | Prompts+LN  | 75.01 | 72.1  | 81.92    | 13.33 | 10.01 |
> | LoRA        | 77.64 | 81.55 | 14.01    | 10.25 | 10.02 |
> | LayerNorm   | 76.1  | 81.57 | **85.9** | 85.27 | 10.03 |

---

> ### Author Response · Authors · 2025-02-23
> **Response to Reviewer hu7J (5/6)**
>
> **Table r8:** Change in HM, number of parameters for varying rank of LoRA adapters in comparison with LayerNorm
>
>
> | LoRA rank             | 2      | 4      | 8      | 16     | LayerNorm |
> |-----------------------|--------|--------|--------|--------|-----------|
> | HM                    | 27.49  | 34.08  | 79.24  | 83.15  | 84.58     |
> | Additional parameters | 110592 | 221184 | 442368 | 884736 | 0         |
> | Learnable parameters  | 110592 | 221184 | 442368 | 884736 | 39936 |
>
>
>
>
> Based on our empirical analysis, we find that while LoRA can be a viable adaptation strategy, it introduces certain challenges that make it less favorable compared to LayerNorm tuning in the context of Open-set Single Image Test-Time Adaptation (OSTTA):
>
>
> **(a) Sensitivity to Learning Rate:** As observed in Tables r6 & r7 above, LoRA-based adaptation exhibits significant sensitivity to the choice of optimizer and learning rate. While it performs reasonably well at specific learning rates (e.g., 1e-3 for SGD in Table 1 and 1e-5 for AdamW in Table 2), its performance degrades drastically when the learning rate is not well-tuned. This instability makes LoRA less reliable for real-world TTA, where learning rate tuning is not always feasible.
>
>
> **(b) Increased Model Complexity:** Unlike LayerNorm tuning, which does not introduce additional parameters, LoRA requires the addition of adapter modules during deployment.  As shown in Table r8, the number of additional parameters scales with the rank of the LoRA adapter, increasing from 110K (rank=2) to 884K (rank=16). This added complexity can be a concern for lightweight real-time adaptation settings, especially in resource-constrained environments. A key distinction between using LoRA for training-time finetuning and test-time adaptation is that in the former, LoRA weights can be merged with the base model post-training, effectively eliminating the additional parameter overhead. However, in TTA, adaptation is a continuous process, meaning that the additional parameters cannot be merged with the backbone model. This makes LoRA-based adaptation more computationally and memory-intensive compared to LayerNorm tuning, which operates within the existing model structure.
>
>
> **(c) Sensitivity to Rank Selection:** As seen in Table r8 above, the performance of LoRA is highly dependent on the rank selection. Lower-rank configurations (e.g., rank=2, 4) lead to poor adaptation, whereas higher-rank settings (e.g., rank=16) perform significantly better but at the cost of increased parameter overhead. This trade-off introduces another hyperparameter that must be carefully tuned, adding complexity to the adaptation process.
>
>
> While LoRA-based adaptation has been explored in recent works, our findings suggest that LayerNorm tuning remains a more efficient and robust choice for OSTTA due to its stability across learning rates, zero additional parameter overhead, and suitability for continuous adaptation without requiring explicit rank selection. We will include this discussion in the manuscript to reflect these insights.
>
>
> >### **Failure case analysis and Qualitative results**
>
> We appreciate the reviewer’s suggestion. Based on our previous experiments and the additional analyses conducted during the rebuttal, we will include a discussion on failure cases and qualitative results in the manuscript revision.
> Our findings indicate that while ROSITA outperforms baselines, it struggles in cases where undesired classes are highly similar to desired ones (e.g., CIFAR-10C vs. CIFAR-100C), leading to a higher false positive rate (FPR). This suggests room for improvement in handling fine-grained or semantically related class distinctions.
> Moreover, while our work focuses on identifying undesired class samples as “I don’t know” in many real-world applications, these new classes could be valuable and need incremental inclusion in the desired set. Exploring an incremental learning approach within OSTTA, where the desired class set expands over time, is an exciting direction for future research. We believe this will provide valuable insights and serve as a useful resource for future works building on our method.

---

> > ### Author Response · Authors · 2025-02-23
> > **Response to Reviewer hu7J (6/6)**
> >
> > >There are a few spacing errors in the writing. E.g., TPT citation on page 2.
> >
> > Thanks for pointing it out. We will correct them in the manuscript revision.
> >
> >
> > >### **Benchmarks and Datasets**
> >
> >
> > We agree with the reviewer's suggestion to include the vision-only methods for a unified benchmark.
> >
> >
> > **Benchmarks:** We have indeed made a sincere effort to unify two closely related problem settings to establish a strong benchmark for Open-set Single-image Test-Time Adaptation (OSTTA):
> > 1. **Open-set TTA benchmarks (vision-only, batchwise adaptation):** We modify open-set TTA approaches originally designed for CNNs, such as (K+1)PC[5] and UniEnt[8] , to work with VLMs in a single-image setting.
> > 2. **Closed-set TTA benchmarks (VLM-based, single-image adaptation):** We adapt several existing single-image TTA methods designed for VLMs, including ZSEval , TPT[6], PromptAlign[7] ,TDA[3] and DPE[4]. Additionally, we introduce **continuous model updates** for TPT and PAlign by adapting prompts, referring to these variants as TPT-C and PAlign-C, respectively.
> >
> >
> > For a fair comparison, all baseline methods incorporate a simple and efficient class identification mechanism based on the LDA objective  to handle the open-set nature of the test stream. A detailed description of these methods is provided in Appendix B.
> >
> >
> > **Datasets:** We evaluate on datasets used in prior open-set TTA works, including CIFAR-10C, CIFAR-100C, ImageNet-C, ImageNet-R, and VisDA for desired classes, and MNIST, SVHN, CIFAR-100C, and CIFAR-10C for undesired classes. Additionally, we incorporate Clipart, Painting, and Sketch from DomainNet to further test generalization. A detailed dataset description is provided in Appendix B.3.
> >
> >
> > >### **Broader Impact Concerns**
> >
> > Our work introduces **Open-set Single-image Test-Time Adaptation (OSTTA)** for Vision-Language Models (VLMs), a novel and realistic problem studied for the first time. While VLMs are powerful for open-world recognition, real-world deployment benefits from the ability to recognize unknown objects and adapt effectively. Our study provides a strong baseline for evaluating their robustness under distribution shifts. While this is a step forward, some considerations are to be aware of before practical deployment. In healthcare, careful adaptation is crucial to avoid misinterpretations; in surveillance, fairness needs attention; and in autonomous systems, ensuring reliability is key. That said, our method is designed with privacy in mind as it does not store images, only extracted features, minimizing risks. Future work could explore safeguards against potential adversarial attacks, but overall, we see this as a positive step toward making VLMs more adaptable and reliable in diverse settings.
> >
> > We will include this in the manuscript revision.
> >
> >
> > [1] Imam et al. "Test-Time Low Rank Adaptation via Confidence Maximization for Zero-Shot Generalization of Vision-Language Models", arXiv 2024.
> >
> >
> > [2] Press et al. “RDumb: A simple approach that questions our progress in continual test-time adaptation”, NeurIPS 2023.
> >
> >
> > [3] Karmanov et al., "Efficient test-time adaptation of vision-language models", CVPR 2024.
> >
> >
> > [4] Zhang et al., "Dual Prototype Evolving for Test-Time Generalization of Vision-Language Models", NeurIPS 2024.
> >
> >
> > [5] Li et al. " On the robustness of open-world test-time training: Self-training with dynamic prototype expansion", ICCV 2023.
> >
> >
> > [6] Shu et al., " Test-time prompt tuning for zero-shot generalization in vision-language models", NeurIPS 2022.
> >
> >
> > [7]  Hassan et asl. "Align Your Prompts: Test-Time Prompting with Distribution Alignment for Zero-Shot Generalization", NeurIPS 2023.
> >
> >
> > [8] Z. Gao et al. "Unified Entropy Optimization for Open-Set Test-Time Adaptation", CVPR 2024.

---

> > > ### Comment · Reviewer_hu7J · 2025-03-09
> > > **Final Decision: Accept**
> > >
> > > Dear Authors,
> > >
> > > Thank you for providing a detailed review response. I have read the authors response, and almost all of my comments and concerns have been adequately addressed and answered.
> > >
> > > For the concern of using external feature banks, it is important to make it distinctive in the main manuscript how it is different compared to saving the batch of images.
> > >
> > > I am leaning to give acceptance decision, given that authors include all the experiments, discussions and analysis presented during the rebuttal process for all reviewers.
> > >
> > > Thank you and kind regards!

---

### Review · Reviewer_63vr · 2025-02-18

**Summary Of Contributions:**

This paper tackles the challenge of Open-set Single Image Test Time Adaptation (OSTTA) in dynamic real-world environments, where models must adapt to sequential test samples and distinguish between known and unknown classes. The authors introduce ROSITA, a novel framework that leverages dynamically updated feature banks and a contrastive learning objective to improve the separation between desired and undesired classes. The key contributions include the establishment of a benchmark for OSTTA using Vision-Language Models (VLMs), the introduction of the ReDUCe loss to enhance class separability, and the identification of LayerNorm parameters as an efficient adaptation strategy for VLMs in open-set scenarios. Extensive experiments show that ROSITA achieves state-of-the-art performance, offering strong accuracy and computational efficiency for real-time deployment.

**Audience:**

Yes

**Broader Impact Concerns:**

The paper does not address the ethical implications of the proposed method, especially regarding real-world applications like healthcare, surveillance, or autonomous systems.

**Claims And Evidence:**

Yes

**Requested Changes:**

1. Provide More Detailed Explanations of Symbols and Terminology: The paper would benefit from more detailed explanations of the various symbols used, particularly in the loss function. For instance, it would be helpful to specify what each positive and negative sample refers to in the context of the loss. Clarifying these elements would enhance the paper's readability and ensure that readers can more easily follow the methodology.
2. Emphasize the Difference Between "Desirable" and "Reliable" Samples: Given the importance of both "desirable" and "reliable" samples in the proposed method, a clearer distinction between these two concepts is needed. These terms have a conceptual overlap, and further discussion of their differences and mechanisms would help avoid confusion. Expanding on this distinction and providing a more in-depth explanation of their roles in the method would improve both the clarity and depth of the paper.

**Strengths And Weaknesses:**

### Strength
1. Innovative Approach to Open-set TTA: The paper introduces a novel approach for Open-set Single Image Test Time Adaptation (OSTTA) using Vision-Language Models (VLMs). By distinguishing between "desired" and "undesired" classes, it addresses a real-world scenario where models must not only adapt to domain shifts within known classes but also reject unfamiliar samples, providing significant improvements over previous methods.
2. Efficient Adaptation: The framework emphasizes computational efficiency, especially through the identification of LayerNorm parameters as optimal for continuous adaptation of VLMs. The paper provides a comprehensive benchmark for OSTTA using VLMs and compares ROSITA with a variety of existing methods, demonstrating clear superiority in terms of accuracy and efficiency.
3. Theoretical Justification and Gradient Analysis: The gradient analysis for the ReDUCe loss adds strong theoretical grounding to the method, making the approach more rigorous and robust. This detailed analysis helps clarify how contrastive loss works to improve the separation between classes.
4. Realistic Test Scenarios: The authors simulate several real-world-inspired test scenarios (e.g., changing domains, varying sample ratios), which provides a practical context for evaluating the method’s effectiveness.
### Weakness
1. The robustness in real-world application: The tuning of VLM is highly sensitive to the learning rate, which can lead to unstable performance, and the experiments are based on relatively small-scale datasets, limiting the method's ability to handle larger, more complex data distributions in practical deployment scenarios. These factors should be addressed to improve its reliability in dynamic, real-world environments.

---

> ### Author Response · Authors · 2025-02-23
> **Response to Reviewer 63vr (1/3)**
>
> We thank **Reviewer 63vr** for acknowledging the strengths of this work, including the innovative approach for OSTTA, efficient model adaptation, theoretical analysis,and effectiveness of the method across various realistic test scenarios. Here, we address the concerns raised:
>
>
> >### **Sensitivity to learning rate**
>
>
> We agree that the tuning of VLM can be sensitive to learning rate, which is the case with neural networks in general as well. This was the motivation to identify the best possible choice of parameters in terms of sensitivity and efficiency for fine-tuning VLMs, especially in the single-image test scenario. Based on the empirical analysis done (Section 4), we identify LayerNorm parameters as a reliable choice for model adaptation. For learning rates less than $1e^{-3}$, the performance is better than zero-shot performance, unlike the other parameter choices. Based on the suggestion from Reviewer hu7J, we consider LoRA adapters as another potential choice of parameters and report the results below:
>
>
> **Table r1:** Accuracy on CIFAR-10C with SGD optimizer and varying learning rates.
>
>
> | Method      | #params   | 1e-6  | 1e-5  | 1e-4  | 1e-3      | 1e-2    |
> |-------------|-----------|-------|-------|-------|-----------|---------|
> | Prompts     | 2048      | 73.4  | 31.04 | 12.53 | 11.18     | 10.19   |
> | Full        | 124436481 | 10.48 | 10.44 | 9.99  | 10.0      | 10.01   |
> | First Block | 7097088   | 75.10 | 76.12 | 78.27 | 13.07     | 10.01   |
> | Last Block  | 7097088   | 73.45 | 72.42 | 59.44 | 10.17     | 10.02   |
> | Prompts+LN  | 41984     | 73.82 | 46.77 | 24.71 | 10.24     | 10.18   |
> | LoRA        | 884736    | 73.86 | 73.90 | 75.42 | 83.15     | 12.58   |
> | LayerNorm   | **39936** | 74.35 | 76.61 | 80.41 | **84.58** | 11.69   |
>
>
> The above table shows that LayerNorm is a very good choice of parameters to update both in terms of complexity and sensitivity to learning rate choices with zero additional parameter overhead.
>
>
>
> >### **Performance in complex scenarios**
>
>
> We agree with the reviewer that a reliable method should handle complex data distributions mimicking real-world environments. Towards this goal, we have performed several experiments across various datasets in terms of the type of domain shift, the number of samples, and several dynamic scenarios. In addition, we tested our method on the Continuously Changing Corruptions (CCC) benchmark[2] to study the performance in very long-term adaptation scenarios. We summarize them all below:
>
>
>
>
> **(a) Datasets:** The datasets we experiment on cover small to large-scale test scenarios, ranging from 10k samples to 50k samples (Appendix B.3 and Table 8). The following table summarizes the datasets used:
>
>
> **Table r2:** No. of images for datasets used for desired classes $D_d$.
>
>
> | Dataset       | CIFAR-10C | CIFAR-100C | Clipart | Painting | Sketch | ImageNet-R | ImageNet-C | VisDA |
> |---------------|-----------|------------|---------|----------|--------|------------|------------|-------|
> | No. of images | 10k       | 10k        | 29.2k   | 43.7k    | 41.8k  | 30k        | 50k        | 50k   |
>
>
> **(b) Changing domains:**  We emphasize that, in Open-Set CTTA scenarios (Table 5a and Appendix D.3 Table 18), we evaluate the performance on a sequence of 15 domains, each domain containing 10k samples, covering **long-range test scenarios** of 150k samples. We also evaluate a more complex setting where the domains are changed very frequently (Table 5b), simulating another practical deployment scenario.
>
> **(c) Varying ratio:** We change the number of samples from desired classes and undesired classes, creating practical test environments where this ratio is unknown (Table 5c).
>
> **(d) Corruption types:** In addition to the main experiments on CIFAR-10 and CIFAR-100 done with snow-corrupted images, we also demonstrate the performance of our method for various types of corruption, including Impulse noise (Table 15), Motion  Blur (Table 16), JPEG Compression (Table 17).
>
> **(e) Large vision language backbones:** In addition to CLIP ViT-B/16, in order to test the robustness across the complexity of backbones, we perform experiments using ViT-L/14 and OpenCLIP ViT-L/14 (Table 14).

---

> > ### Author Response · Authors · 2025-02-23
> > **Response to Reviewer 63vr (2/3)**
> >
> > **(f) Continuously Changing Corruptions (CCC), another practical scenario:** CCC benchmark[2] was specifically introduced to assess the long-term continual adaptation behavior of TTA methods in a changing world, covering scenarios such as weather changing from foggy to rainy, day to night. We experiment with the CCC dataset where gradual domain changes are synthesized across 15 corruptions of ImageNet-C as desired class dataset $D_d$ and MNIST as undesired class dataset $D_u$. The results are reported below. We evaluate the performance on the open-set CCC benchmark for a total test sequence length of 300k samples and report the results below:
> >
> >
> > **Table r3:** Performance of different methods on CCC benchmark.
> >
> >
> > | Method     | $AUC \uparrow$ | $FPR \downarrow$ | $Acc_D \uparrow$ | $Acc_U \uparrow$ | $Acc_{HM} \uparrow$ |
> > |------------|----------------|------------------|------------------|------------------|---------------------|
> > | ZS-Eval    | 88.45          | 88.24            | 18.37            | 99.53            | 31.01               |
> > | (K+1)PC    | 95.82          | 20.16            | 20.37            | 99.97            | 33.84               |
> > | TDA        | 87.42          | 85.83            | 20.42            | 99.35            | 33.87               |
> > | UniEnt     | 89.90          | 82.86            | 18.99            | 99.62            | 31.90               |
> > | DPE        | 84.68          | 87.56            | 18.47            | 99.40 | 31.16 |
> > | **ROSITA** | **96.02**          | **19.96**            | **21.14**            | **99.11**            | **34.84**               |
> >
> >
> >
> >
> >
> > [1] Imam et al. "Test-Time Low-Rank Adaptation via Confidence Maximization for Zero-Shot Generalization of Vision-Language Models", arXiv 2024.
> >
> >
> > [2] Press et al. “RDumb: A simple approach that questions our progress in continual test-time adaptation,” NeurIPS 2023.
> >
> >
> > >### **Detailed explanation of symbols and Terminology**
> >
> >
> > We appreciate the reviewer’s suggestion to provide clearer explanations of the symbols and terminology used. We clarify them here:
> >
> >
> > **Positives and Negatives in the loss terms:**
> >
> >
> > **(i) Positives:** In $L_D$, positives refer to the k nearest neighbours of the test feature $f_t$ from the desired class feature bank $M_d$ that belong to the same class and should be pulled closer together in the embedding space. The goal is to reinforce their similarity, ensuring that intra-class variations are minimized. Similarly, in $L_U$, positives are the k nearest neighbours of the test feature $f_t$ from the undesired class feature bank $M_u$, which are pulled closer together in the embedding space, ensuring all the undesired class samples are clustered together.
> >
> >
> > **(ii) Negatives:** In $L_D$, negatives refer to the k nearest neighbours of the test feature $f_t$ from the undesired class feature bank $M_u$, which appear in the denominator term of $L_D$ with the goal of pushing desired class sample feature $f_t$ apart from undesired class features, with the goal of separating desired and undesired class samples. Similarly, in $L_U$, negatives are the k nearest neighbours from the desired class feature bank $M_d$, the objective being to push the undesired class test sample $f_t$ away from the desired class samples.
> >
> >
> > **Difference between "Desired" and "Reliable" samples:**
> >
> >
> > The primary difference is that **Desired vs Undesired** are defined in the context of classes, whereas **Reliable vs Unreliable** are identified based on the quality of the sample for TTA.
> >
> >
> > **Desired vs Undesired:** These terms are defined in the context of classes. Desired classes refer to the classes of interest into which we aim to classify a sample. A sample belonging to one of the desired classes is referred to as a desired class sample. Any class that is not of interest is referred to as an undesired class, and correspondingly, a sample belonging to it is called an undesired class sample. A test sample $x_t$ is identified to belong to a desired or undesired class based on the score $s_t$ as follows:
> >
> >
> > (a) Desired class sample: $s_t \ge \tau_t^*$
> >
> >
> > (b) Undesired class sample: $s_t < \tau_t^*$
> >
> >
> >
> >
> > **Reliable vs Unreliable:** A test sample is said to be reliable if it is very confident so that it can reliably be used for TTA. Reliable samples can belong to both desired and undesired classes. In our work, we characterize test samples based on the score $s_t$ as follows:
> >
> >
> > (a) Reliable sample from desired class: $\tau_t^* < \mu_d < s_t $
> >
> >
> > (b) Unreliable sample from desired class: $\tau_t^* \le s_t \le \mu_d $
> >
> >
> > (c) Reliable sample from undesired class: $s_t < \mu_u < \tau_t^*$
> >
> >
> > (d) Unreliable sample from udesired class: $\mu_u \le s_t \le \tau_t^*$
> >
> >
> > where
> >
> >
> > $s_t$: score of sample $x_t$ at time $t$.
> >
> >
> > $\tau_t^*$: Best threshold identified by LDA class identifier.
> >
> >
> > $\mu_d$: Mean of the scores of desired class samples.
> >
> >
> > $\mu_u$: Mean of the scores of undesired class samples.
> >
> >
> > We will refine the manuscript for better clarity in these aspects.

---

> > > ### Author Response · Authors · 2025-02-23
> > > **Response to Reviewer 63vr (3/3)**
> > >
> > > >### **Broader Impact Concerns**
> > >
> > > Our work introduces Open-set Single-image Test-Time Adaptation (OSTTA) for Vision-Language Models (VLMs), a novel and realistic problem studied for the first time. While VLMs are powerful for open-world recognition, real-world deployment benefits from the ability to recognize unknown objects and adapt effectively. Our study provides a strong baseline for evaluating their robustness under distribution shifts. While this is a step forward, some considerations are to be aware of before practical deployment. In healthcare, careful adaptation is crucial to avoid misinterpretations; in surveillance, fairness needs attention; and in autonomous systems, ensuring reliability is key. Our method is designed with privacy in mind as it does not store images, only extracted features, minimizing risks. Future works could explore safeguards against potential adversarial attacks, but overall, we see this as a positive step toward making VLMs more adaptable and reliable in diverse settings.
> > >
> > > We will include this in the manuscript revision.

---

### Review · Reviewer_cGsT · 2025-02-18

**Summary Of Contributions:**

- This research explores **Open-set Single image Test Time Adaptation (OSTTA)** for Vision-Language Models, addressing the complex task of continuous model adaptation to individual test samples while maintaining the ability to identify unknown classes—a critical challenge in real-world applications where test samples arrive sequentially and may contain unseen categories.

- The proposed solution, **ROSITA**, combines a contrastive learning-based **ReDUCe** loss with optimized LayerNorm parameter adaptation, enabling efficient separation between desired and undesired classes without requiring batch processing or closed-set assumptions that limit existing approaches. The framework not only establishes new benchmarks for OSTTA but also shows remarkable performance in managing domain shifts and unknown class recognition across diverse adaptation scenarios, demonstrating its practical utility in dynamic real-world environments.

- Through extensive experiments, the authors validate ROSITA's effectiveness in both adapting to domain shifts within known categories and accurately identifying unseen classes as "unknown," while maintaining computational efficiency suitable for real-time deployment.

**Audience:**

Yes

**Broader Impact Concerns:**

There are no concerns in this aspect.

**Claims And Evidence:**

Yes

**Requested Changes:**

**Questions**
- What is the conceptual justification for Equation 7?
- What insights does the gradient analysis provide, and how does it contribute to understanding the method's effectiveness?

**Writing Suggestions**
- Equation-related issues:
  - Always include "," or "." after equations by considering each equation is part of a complete sentence.
  - Fix missing equal signs "=" in Equations 17 and 18 (Page 22).
- Notation standardization:
  - $Q^d$, $Q^u$ and $Q_d$, $Q_u$ are used interchangeably throughout the paper - maintain consistency with one format.
  - Fix inconsistent set notation: Change "Cd = {c1, c2, . . . cN} / {t1, t2, . . . tN }" to "Cd = {c1, c2, . . ., cN} / {t1, t2, . . ., tN }."
- Terminology consistency:
  - "open-set" vs "open set"
  - "test-time" vs "test time"
  - Recommend using hyphenated versions ("open-set" and "test-time") consistently throughout
- Citation formatting:
  - Use \citep instead of \cite for most cases (e.g., in TPTShu et al. (2022))

- Specific page corrections:
  - Page 8: "Appendix B.4 We" should be "Appendix B.4. We."
  - Page 10: "Similarly, we use LU ( Equation 7)" should be "Similarly, we use LU (Equation 7)."
  - Page 22: Add missing equal signs in Equations 17 and 18.

**Strengths And Weaknesses:**

**Strength**
- The method consistently improves performance while effectively distinguishing undesired classes in open-set recognition situations.
- Beyond overall performance metrics, Table 1 effectively demonstrates the method's capability in discriminating between desired and undesired classes.

**Weaknesses**

Methodology
- Limited Methodological Innovation:
  - The proposed method primarily combines existing techniques rather than introducing innovative components.
  - The Desired vs Undesired class identifier employs pre-existing 1D Linear Discriminant Analysis (LDA).
  - Test-time adaptation using kNN and contrastive learning has been previously explored in existing TTA methods [1, 2].
  - LayerNorm tuning cannot be considered a methodological component as it has been treated as a hyperparameter in prior work including TPT. A more innovative approach or stronger theoretical justification is needed.

- Technical Concerns:
  - The retrieval of K nearest neighbors using image features (Equation 4) may be unreliable, as recent literature indicates that the CLIP image encoder can ambiguously encode semantically distinct images with high cosine similarity.
  - Why does LayerNorm tuning outperform other parameter tuning approaches?
  - How does contrastive learning affect the representation space? Comprehensive analysis and justification are needed.

Experiments
- In Tables 1 and Table 2, the performance improvements are marginal. The superiority of the proposed method over baselines is not convincing. In particular:
  - The source of significant performance gain compared to ZSEval requires more explanation.
  - Performance improvement compared to (K + 1)PC appears only marginal.
- Table 3 shows occasional drops in FPR under certain settings but does not analyze these variations. The authors should provide potential explanations for these inconsistencies.
- The choice of MNIST as undesired samples is highly unrealistic for natural image scenarios, as handwritten digits rarely appear among natural images in real-world applications. Using corrupted but semantically similar images would provide a more realistic evaluation scenario.
- The ablation study in Table 4 is incomplete, focusing solely on AUC/FPR/HM metrics while omitting crucial analysis of how each component affects desired class accuracy.
- Experiments in Table 5 would benefit from a broader evaluation using various backbones and datasets beyond CIFAR-10C to demonstrate generalizability.
- Table 7 and Figure 4 require more comprehensive baseline comparisons for memory/time assessment, as relative efficiency metrics are crucial for practical deployment.

[1] Liu, Yuejiang, et al. "TTT++: When Does Self-Supervised Test-Time Training Fail or Thrive?." NeurIPS, 2021.

[2] Chen, Dian, et al. "Contrastive Test-Time Adaptation." CVPR, 2022.

---

> ### Author Response · Authors · 2025-02-27
> **Response to Reviewer cGsT (1/7)**
>
> We thank Reviewer CGsT for recognizing the significance of OSTTA and ROSITA, acknowledging our new benchmark and extensive experiments validating its effectiveness across diverse adaptation scenarios. We now address the concerns here:
>
> >### **On Limited Methodological Innovation**
>
> We appreciate the feedback regarding the novelty of our approach. While our method builds upon existing techniques, one of our primary goal is to establish a strong and practical benchmark for **Open-set Single-image Test-Time Adaptation (OSTTA)** using VLMs, a problem that has not been systematically explored before. We believe this is a necessary step to bridge the gap between conventional TTA research and real-world challenges. Below, we clarify our contributions in light of the reviewer’s points:
>
> **1. Motivation for OSTTA**
> Most TTA research focuses on closed-set adaptation, assuming test samples belong to a predefined set of classes. However, real-world deployment is inherently open-world, and models will encounter previously unseen classes, which closed-set TTA methods will misclassify. Instead of incorrectly labeling a bicycle as a car, an intelligent system should be able to say "I don’t know." This motivates the need to study Open-set TTA, which has only been explored in batch-based setting and using CNN [3,4].
>
> **2. Establishing a Benchmark for Open-set Single-image TTA with VLMs**
> **Why VLMs?** Traditional CNN-based TTA assumes that a model is trained on a specific dataset (e.g., CIFAR-10), which is then adapted to its corrupted test set (CIFAR-10C) using batches of test data. In contrast, Vision-Language Models (VLMs, e.g., CLIP) are pretrained on large-scale image-text pairs, enabling them to generalize across diverse domains without requiring dataset-specific retraining, leveraging text prompts for zero-shot classification. This makes them natural candidates for TTA, as demonstrated in recent closed-set single-image TTA works[5,6,7,8]. However, their robustness to corruption and distribution shifts remains suboptimal [5], and they **cannot explicitly reject out-of-distribution (OOD) or undesired class samples.** CLIP can only classify an image from a given set of desired classes, making open-set recognition a critical challenge.
>
> This raises the following research questions:
> 1. **How well can VLMs perform in open-set scenarios?**
> 2. **Can they be effectively adapted in a continuous manner?**
> 3. **How do we equip VLMs to handle domain shifts within desired classes while accurately rejecting unfamiliar samples?**
>
> To address these questions, we establish a **new benchmark** for Open-set Single-Image TTA, formally defining the problem, structuring evaluation, and enabling future research in this crucial direction. To systematically bridge closed-set and open-set TTA, we adapt and unify various baselines:
> - **Closed-set single-image VLM-TTA methods** (TPT [5], PromptAlign [6], TDA [7], DPE[8])
> - **Open-set CNN-based TTA methods** ((K+1)PC [3], UniEnt [4])
>
> **3. Addressing Concerns on Individual Methodological Components**
>
> - **Combination of Existing Techniques:** We recognize that our method is inspired from prior ideas, such as LDA-based class identification and test-time adaptation with kNN and contrastive learning. However, our first step was to evaluate how well-established prior works perform when adapted to OSTTA for VLMs. To this end, we set up the benchmark, conduct rigorous parameter update analysis, and employ an LDA-based class identifier. Building on the insights from CNN and VLM-based TTA methods, we design a unified framework, ROSITA for the OSTTA setting.
>
> - **LDA-Based Class Identifier:** While methods like MaxLogit could be used for class identification, they require pre-determined thresholds, which are **non-trivial to set for single-image TTA.** We adapt the dynamic thresholding mechanism from [3] for our setting and consistently use it across all baselines. In **ROSITA**, we further leverage its score statistics to **select reliable samples** for the proposed ReDUCe loss, ensuring robust adaptation.
>
> - **LayerNorm Tuning as a Baseline:** We agree that updating LayerNorm parameters has been explored in prior works[9]. However, prior works on VLM-based TTA[5,6,7,8] did not explicitly validate this design choice, which we rigorously ablate in our study. Our objective is to systematically analyzing role of parameter choices for VLM-based TTA, highlighting the contrast between CNN-based and VLM-based adaptation strategies. We clarify that we do not claim this as a novel contribution; rather, we highlight that LayerNorm tuning serves as a simple yet strong baseline that warrants study, given the well-established effectiveness of normalization updates in CNNs.

---

> ### Author Response · Authors · 2025-02-27
> **Response to Reviewer cGsT (2/7)**
>
> **4. ReDUCe Loss as a Key Contribution**
> While contrastive learning is widely used, the proposed **ReDUCe loss is designed specifically for OSTTA.** Unlike prior contrastive TTA approaches [1,2], it introduces the following key innovations:
>
> - **Separates desired and undesired classes:**
>   - $L_D$ encourages tight intra-class clustering for desired class samples.
>   - $L_U$ ensures undesired samples remain distinct from the desired classes.
> - **Dynamically updating feature banks:** Instead of static memory banks, we maintain feature banks that evolve during adaptation.
> - **Reliable sample selection:** Only high-confidence samples update the model to prevent catastrophic forgetting of the source VLM knowledge. This design choice is extensively ablated (Appendix C.5).
>
>
> We acknowledge that contrastive objectives often share a similar mathematical form [2, 10, 11], but the **way they are adapted to specific problems is critical.** For example, AdaContrast [2] adapts MoCo [11] for TTA, while MoCo itself was adapted from SimCLR [10]. Similarly, **our loss is specifically designed to handle open-set adaptation challenges in single-image settings.**
>
> By establishing a structured benchmark and evaluation framework, we provide a **strong foundation for future research on Open-set TTA of VLMs**. We hope this addresses your concern. We would be happy to clarify any further questions regarding this.
>
> ----------------
>
> >### **Reliability of kNN retrieval using CLIP**
>
> We have conducted a detailed analysis of kNN retrieval accuracy to assess the reliability of using image-features for retrieving neighbors in ROSITA.
>
> 1. **Ablation on Memory Bank Size:** We performed an ablation study on the number of neighbors $K$ (Appendix C.2). As shown in Table 7, we set $K=5$ based on empirical performance. Even with $K=1$, ROSITA significantly improves $Acc_{HM}$ compared to the case where $L_D, L_U$ are not used ($K=0$), indicating that incorporating even a single reliable neighbor aids adaptation. Further increasing $K$ (3 to 9) has minimal impact, suggesting that selecting multiple positive and negative samples contributes to adaptation but is not highly sensitive to $K$.
>
> 2. **Empirical Analysis of kNN Retrieval Accuracy:**
>    (i) We compute the average number of correctly matched neighbors $K^+$ in $L_D$ (Equation 6), where the pseudo-label $y^+$ of the retrieved neighbors matches the test sample’s pseudo-label $y_t$.
>    (ii) We evaluate the pseudo-label accuracy of the reliable samples, as only these are stored in the feature bank for kNN retrieval and model adaptation. Higher accuracy of these reliable samples directly benefits adaptation.
>
> **Table r1:** Analysis on kNN retrieval
>
> | Metric                                        | CIFAR-10C | ImageNet-R | ImageNet-C |
> |-----------------------------------------------|-----------|------------|------------|
> | Average $K^+$ in $L_D$                | 4.1       | 2.5        | 1.5        |
> | Accuracy of reliable desired class samples    | 88.7%     | 86.40%     | 43.74%     |
> | Accuracy of reliable undesired class samples  | 99.75%    | 100%       | 99.95%     |
>
>
> For $L_D$, the average number of correctly matched neighbors aligns with dataset difficulty; CIFAR-10C has the highest $K^+$ (4.1), while ImageNet-C has the lowest (1.5), reflecting the increased complexity of retrieval in more challenging datasets. Additionally, the high accuracy of reliable desired (≥ 86% for CIFAR-100C and ImageNet-R) and undesired (nearly 100%) class samples ensures that the retrieved neighbors serve as strong supervisory signals for adaptation. Since $L_D$ relies only on correctly matched neighbors, the adaptation process remains robust, effectively leveraging reliable samples regardless of dataset difficulty. However, there is still significant scope of improvement of desired class accuracy in harder datasets like ImageNet-C, where CLIP accuracy itself is still poor, given the severe domain shift and the 1000-way classification task, making it very challenging. Nevertheless, ROSITA still successfully separates desired and undesired class samples as observed from the AUC, FPR metrics in Table 1.
>
>
> For $L_U$ (Equation 7), the goal is to separate undesired class features from desired ones, so we utilize all five retrieved neighbors. Notably, the accuracy of reliable undesired samples is nearly 100% across datasets, ensuring a strong negative signal for contrastive separation. These findings confirm that ROSITA effectively balances retrieval-based adaptation by leveraging high-confidence samples, reinforcing its robustness across different datasets.

---

> > ### Author Response · Authors · 2025-02-27
> > **Response to Reviewer cGsT (3/7)**
> >
> > >### **Effectiveness of LayerNorm tuning:**
> >
> > Our analysis in Section 4. highlights that similar to BatchNorm in CNNs, LayerNorm (LN) plays a crucial role in mitigating covariate shift[12], making it particularly effective for test-time adaptation in VLMs. However, in the context of single-image TTA, where most prior methods relied on prompt tuning, an extensive analysis was necessary to determine the most effective parameter for fine-tuning. This study systematically evaluates different adaptation strategies, leading to the following key insights:
> >
> > - **Feature Normalization:** Normalization layers play a crucial role in adapting to domain shifts[12]. Prior works on CNN-based TTA have shown that updating normalization layers is a strong and simple adaptation strategy [12, 13]. Given that VLMs also heavily rely on normalization, our results confirm that LayerNorm tuning effectively calibrates features without overfitting, making it a good choice for test-time adaptation. Our observations also comply with the claims in [9] where they extensively evaluate the effectiveness of tuning LayerNorms with attention in large models.
> > - **Minimal Overhead:** LayerNorm tuning requires significantly fewer trainable parameters compared to fine-tuning the entire model, making it computationally very efficient.
> > - **Empirical Justification:** Our experiments systematically compare LayerNorm tuning with alternative parameter tuning strategies (e.g., tuning the classifier head, tuning later layers, or full fine-tuning). The results consistently show that LayerNorm tuning achieves the best trade-off between adaptation effectiveness and stability, making it the preferred choice.
> >
> > While LayerNorm tuning itself is not novel, its effectiveness for VLM-based TTA was not well established. We hope this analysis serves as a valuable reference for future TTA research.
> >
> > ----
> > >### **The effect of Contrastive learning**
> >
> > Contrastive learning influences the representation space by enhancing feature separability and improving the discriminability of desired vs. undesired classes. In ROSITA, the ReDUCe loss ensures that only reliable samples are used for model adaptation, refining the VLM's embedding space for improved open-set recognition.
> >
> > In addition to the accuracy metrics presented, here we compute the **inter-feature distance** between desired ($C_d$) and undesired ($C_u$) class features through the following distance metrics:
> >
> > **Mean Pairwise Distance:**
> > A simple but effective way is to compute the pairwise Euclidean or cosine distance between all $C_d$ and $C_u$ samples and then take the mean:
> >
> > - **Euclidean Distance**
> >
> > $d_{\text{Euclidean}} = \frac{1}{|F_d||F_u|} \sum_{f_d \in F_d} \sum_{f_u \in F_u} \| f_d - f_u \|_2$
> >
> > - **Cosine Distance**
> >
> > $
> > d_{\text{Cosine}} = 1 - \frac{1}{|F_d||F_u|} \sum_{f_d \in F_d} \sum_{f_u \in F_u} \frac{f_d \cdot f_u}{\|f_d\| \|f_u\|}
> > $
> >
> > where $F_d$ and $F_u$ are the feature banks for desired and undesired class samples, respectively.
> >
> > **Wasserstein Distance (Optimal Transport):** This measures the optimal transport cost between the two feature distributions, making it ideal for comparing feature distributions of desired ($C_d$) and undesired ($C_u$) classes.
> >
> > $
> > d_{\text{Wasserstein}} = \inf_{\gamma \in \Pi(F_d, F_u)} \mathbb{E}_{(f_d, f_u) \sim \gamma} [\| f_d - f_u \|]
> > $
> >
> > **Table r2:** Inter-feature distance measures for ZSEval and ROSITA using CIFAR-10C as $D_d$.
> >
> > | Undesired dataset | Method | Euclidean Distance (↑) | Cosine Distance (↑) | Wasserstein Distance (↑) |
> > |-------------------|--------|------------------------|---------------------|--------------------------|
> > | MNIST             | ZSEval | 0.5831                 | 0.1731              | 0.5553                   |
> > |                   | ROSITA | **1.2970**             | **0.8623**          | **1.2708**               |
> > | SVHN              | ZSEval | 0.6963                 | 0.2456              | 0.6535                   |
> > |                   | ROSITA | **1.0785**             | **0.5918**          | **1.0265**               |
> > | TinyImageNet      | ZSEval | 0.7522                 | 0.2860              | 0.6583                   |
> > |                   | ROSITA | **0.9939**             | **0.4994**          | **0.8852**               |
> > | CIFAR-100C        | ZSEval | 0.5587                 | 0.1598              | 0.4265                   |
> > |                   | ROSITA | **0.8194**             | **0.3411**          | **0.6594**               |
> >
> > ROSITA consistently improves feature separability over ZSEval, demonstrating its effectiveness in open-set recognition. The improvement varies with task difficulty; MNIST shows larger distance gains, while CIFAR-100C sees smaller improvements, reflecting the inherent challenge of distinguishing complex datasets. These results validate that contrastive adaptation enhances representation learning, with its impact depending on dataset complexity.

---

> > > ### Author Response · Authors · 2025-02-27
> > > **Response to Reviewer cGsT (4/7)**
> > >
> > > >### **Regarding Performance Improvements and source of gains**
> > >
> > > The primary source of performance gain over ZSEval comes from our proposed loss functions, particularly the ReDUCe loss, in combination with reliable sample selection and LayerNorm tuning. Our ablation studies (Table 4.) clearly demonstrate that each of these components plays a crucial role in enhancing adaptation. By selectively updating the model using only reliable samples, our approach ensures stable and meaningful adaptation, avoiding detrimental updates from unreliable pseudo-labels. Additionally, LayerNorm tuning effectively calibrates feature distributions, further improving performance. The experiments presented in Tables 1-3. consistently support these claims, showing that our method systematically outperforms ZSEval across multiple benchmarks.
> > >
> > > ------
> > >
> > > >### **Regarding Comparison with (K+1)PC:**
> > >
> > > We modify (K+1)PC [3] to perform TTA using VLMs as described in Appendix B.2. Since both (K+1)PC and ROSITA update the LayerNorm parameters in vision encoder, they achieve superior performance compared to methods that do not adapt the encoder. The relative performance between ROSITA and (K+1)PC varies across datasets, depending on the severity of domain shift and accuracy of reliable test samples used for adaptation. Specifically, the two methods achieve similar results on ImageNet-R (Table 1), VisDA and DomainNet (Table 2), while ROSITA demonstrates a more noticeable advantage on corruption benchmarks like ImageNet-C (Table 1), CIFAR-10C, CIFAR-100C (Table 3). Table 5. further highlights ROSITA's effectiveness in more challenging open-set TTA scenarios, where domain shifts are continuous and frequent. Compared to (K+1)PC, ROSITA consistently achieves superior performance across all settings: (a) continuously changing domains, (b) frequently changing domains, and (c) varying ratios of $C_{d}/C_{u}$. These scenarios are inherently more difficult due to dynamic distribution shifts and an increasing presence of unknown categories. While (K+1)PC improves over non-adaptive baselines, its reliance on prototypical cross-entropy loss for every test sample makes it vulnerable to unreliable pseudo-labels, especially in harder scenarios. In contrast, ROSITA’s selective sample reliability mechanism ensures robust updates, leading to significant performance gains. The strongest improvements are observed in scenario (c), where ROSITA outperforms (K+1)PC by a large margin, particularly at lower ratios of $C_{d}/C_{u}$, demonstrating its effectiveness in handling higher uncertainty.
> > >
> > > ------
> > >
> > > >### **Regarding occasional drops in FPR**
> > >
> > > The primary metric to rely on in OSTTA is **HM**, which reflects the harmonic mean of desired and undesired class accuracy. While **AUC** and **FPR** are widely used in OOD detection[14] to measure the distinction between ID and OOD classes (desired and undesired in our case), their interpretation in OSTTA differs due to the online nature of test-time adaptation.
> > >
> > > In standard OOD detection, AUC and FPR are computed **offline** using fixed model weights. However, in OSTTA, the scores used to compute these metrics are collected  in an **online manner**, as the model continuously evolves as it adapts to test samples. The final reported FPR is computed at the end of adaptation. This way of collecting scores and the adaptation dynamics introduces minor inconsistencies in FPR. To demonstrate this, consider Figure 3b, where we observe the scores for some undesired samples overlap with the mode of desired samples ($s_t>0.2$). These scores correspond to the initial test samples corresponding to the time steps $t<2500$ where the model still behaves very much like ZSEval. The scores for desired and undesired class samples overlap significantly for t<2500$ as observed in Figure 3c. Although the scores for desired and undesired class sample become distinctive with time, due to the online nature of accumulation of scores, the scores of initial samples also influence the FPR metric, causing discrepancies at times. While AUC and FPR provide a measure of separation, the absolute trends should be perceived with respect to the online nature of the problem in this case.
> > >
> > > Despite these variations, our approach consistently maintains a low FPR across diverse test scenarios, demonstrating its robustness. We hope this explanation clarifies the observed trends.

---

> ### Author Response · Authors · 2025-02-27
> **Response to Reviewer cGsT (5/7)**
>
> >### **On the choice of MNIST as undesired samples**
>
> We acknowledge that using MNIST as undesired class samples may seem unrealistic for natural image scenarios. However, we followed the dataset choices in (K+1)PC, the first work on open-set TTA, which despite using batch-based adaptation and vision-only models, established a standard evaluation setup. Even in this seemingly trivial open-set case, our observations (Table 1,3) show that CLIP struggles to reject MNIST samples (high FPR for most prior methods), with a significant overlap in scores between desired and undesired classes (Figure 3a).
> We have also conducted experiments using CIFAR-10C as desired classes and CIFAR-100C as undesired class samples, which consist of corrupted but semantically similar images. While improvements in this case are relatively less pronounced than in simpler settings like MNIST, our findings highlight fundamental challenges in equipping VLMs with the ability to say *"I don't know."* This study serves as an extensive exploration of the primary issues in preventing misclassification of novel samples, a crucial step toward more reliable open-set recognition.~~
>
> ------
>
> >### **Accuracy metrics for ablation study on loss components**
>
> We appreciate the reviewer’s suggestion and acknowledge that Table 4. primarily focuses on AUC, FPR, and HM metrics. However, we would like to clarify that the analysis of how each loss component affects desired class accuracy ($ Acc_D $) has been included in **Appendix C.3.**, which we also report here for convenience.
>
> **Table r3:** Detailed metrics for ablation study on loss function.
>
> | $\mathcal{L}_{Re}$ | $\mathcal{L}_{D}$ | $\mathcal{L}_{U}$ | AUC       | FPR      | $Acc_D$   | $Acc_U$   | $Acc_{HM}$ |
> |----------------|----------------|----------------|-----------|----------|-----------|-----------|------------|
> | ❌           | ❌           | ❌           | 91.91     | 85.04    | 60.82     | **99.77** | 75.57      |
> | ✅           | ❌           | ❌           | 95.29     | 30.82    | 68.36     | 99.30     | 80.97      |
> | ❌           | ✅           | ❌           | 95.23     | 28.91    | 66.93     | 98.52     | 79.71      |
> | ❌           | ❌           | ✅           | 98.61     | 12.73    | 66.60     | 99.68     | 79.84      |
> | ❌           | ✅           | ✅           | **99.27** | **4.15** | 67.76     | 99.73     | 80.69      |
> | ✅           | ✅           | ✅           | 99.10     | 7.63     | **72.81** | 99.74     | **84.17**  |
>
> ----
>
> >### **Broader evaluation to demonstrate generalizability.**
>
> In addition to the experiments presented in Table 5, we perform the following experiments, which further strengthen the effectiveness of ROSITA.
>
> **(a) Open-set CTTA:** In addition to open-set CTTA experiments done on CIFAR-10C, here we experiment with CIFAR-100C, a harder dataset with 15 continuously changing domains. The results below show that ROSITA clearly outperforms prior methods in this long range open-set scenario.
>
> **Table r4: Open-set CTTA performance for CIFAR-100C as desired dataset.**
>
> | Method  | MNIST     | SVHN      | Tiny      | CIFAR-100C |
> |---------|-----------|-----------|-----------|------------|
> | ZS-Eval | 39.00     | 36.29     | 38.41     | 35.04      |
> | TPT     | 39.00     | 36.24     | 38.38     | 34.45      |
> | (K+1)PC | 40.64     | 37.05     | 38.23     | 34.55      |
> | TDA     | 40.49     | 40.35     | 39.92     | 35.42      |
> | DPE     | 30.75     | 19.23     | 35.85     | 27.62      |
> | ROSITA  | **41.64** | **38.02** | **40.44** | **36.05**  |
>
> **(b) Continuously Changing Corruptions (CCC) in Open-set scenario:** CCC benchmark[15] was specifically introduced to assess the long-term continual adaptation behavior of TTA methods in a changing world, covering scenarios such as weather changing from foggy to rainy, day to night.  We evaluate the open-set CCC benchmark for a total test sequence length of 300k samples with MNIST as $D_u$ and report the results below.
>
> **Table r5:** Performance of different methods on open-set CCC.
>
> | Method     | $AUC \uparrow$ | $FPR \downarrow$ | $Acc_D \uparrow$ | $Acc_U \uparrow$ | $Acc_{HM} \uparrow$ |
> |------------|----------------|------------------|------------------|------------------|---------------------|
> | ZS-Eval    | 88.45          | 88.24            | 18.37            | 99.53            | 31.01               |
> | (K+1)PC    | 95.82          | 20.16            | 20.37            | 99.97            | 33.84               |
> | TDA        | 87.42          | 85.83            | 20.42            | 99.35            | 33.87               |
> | UniEnt     | 89.90          | 82.86            | 18.99            | 99.62            | 31.90               |
> | DPE        | 84.68          | 87.56            | 18.47            | 99.40 | 31.16 |
> | **ROSITA** | **96.02**          | **19.96**            | **21.14**            | **99.11**            | **34.84**               |
>
> ROSITA outperforms prior methods even in a challenging dataset such as CCC.

---

> ### Author Response · Authors · 2025-02-27
> **Response to Reviewer cGsT (6/7)**
>
> **(c) Varying OOD ratio:** In addition to the results presented in Table 6, we perform experiments using ImageNet-R as desired class dataset which is a relatively large scale dataset with 50,000 images from 200 classes. We report the results below, from which we observe consistent improvements of ROSITA compared to the prior methods.
> **Table r6: Varying ratio of $C_d/C_u$ for ImageNet-R/MNIST.**
>
> | Method  | 0.2       | 0.4       | 0.6       | 0.8        |
> |---------|-----------|-----------|-----------|------------|
> | ZS-Eval | 65.46     | 67.13     | 69.25     | 70.77      |
> | TPT     | 65.67     | 67.73     | 70.12     | 71.54      |
> | TPT-C   | 64.83     | 64.55     | 48.97     | 63.86      |
> | (K+1)PC | 78.09     | 81.08     | 81.35     | 82.61      |
> | TDA     | 67.90     | 71.33     | 71.54     | 71.47      |
> | DPE     | 66.89     | 68.47     | 69.72     | 70.87      |
> | ROSITA  | **82.22** | **83.32** | **83.59** | **83.84**  |
>
> ---
>
> >### **Comprehensive baseline comparisons for memory/time assessment**
>
> **Table:** Comparison of GPU Memory and Time taken for all methods for ImageNet-C as desired class dataset.
>
> | Method       | Category           | GPU Memory (GB) | Time (sec/image) |
> |-------------|-------------------|-----------------|------------------|
> | ZS-Eval     | Training-free     | 5.71 (1X)       | 0.009 (1X)       |
> | TDA         | Training-free     | 5.71 (1X)       | 0.016 (1.78X)    |
> | TPT         | Prompt-tuning     | 23.24 (4.07X)   | 1.000 (111.1X)   |
> | TPT-C       | Prompt-tuning     | 23.24 (4.07X)   | 1.000 (111.1X)   |
> | (K+1)PC     | LayerNorm-tuning  | 5.73 (1.004X)   | 0.032 (3.56X)    |
> | UniEnt      | LayerNorm-tuning  | 5.73 (1.004X)   | 0.032 (3.56X)    |
> | ROSITA      | LayerNorm-tuning  | 5.73 (1.004X)   | 0.028 (3.11X)    |
>
> The above table clearly shows the relative efficiency of each method, emphasizing that **ROSITA maintains near training-free efficiency while significantly outperforming heavier prompt-tuning methods like TPT.**
> Table VII and Figure 4 in the paper provide a comparative analysis of GPU memory and computational time required for representative OSTTA methods for different datasets. Prompt-tuning approaches like **TPT** demand significantly higher memory (23.24 GB for 1000 classes) due to the need for storing intermediate activations and performing backward passes through the text encoder. In contrast, **ROSITA** only requires **5.73 GB**, which is comparable to training-free methods such as ZS-Eval (5.71 GB), despite incorporating adaptation. This balance of efficiency and effectiveness makes ROSITA a practical choice for real-world OSTTA deployment, where computational constraints are critical.
>
> ---
>
> >### **Conceptual justification for $L_U$ (Equation 7).**
>
> The ReDUCe loss is designed to improve the separation between desired class samples ($C_d$) and undesired class samples ($C_u$), ensuring better classification and open-set recognition. While most adaptation methods primarily focus on improving classification within a fixed set of categories, ROSITA **explicitly incorporates undesired class samples into the adaptation process** through Equation 7 as follows:
>
> **1. Identifying Reliable Undesired Class Samples:**
> Since naïvely adapting using all test samples can be harmful, **only reliable undesired class samples** are used for adaptation. A sample is classified as a **reliable undesired class sample** if its confidence score $s_t$ falls **below** a predefined threshold $\mu_u$, derived from the Linear Discriminant Analysis (LDA)-based score distribution.
>
> **2. Contrastive Learning for Undesired Class Samples (Equation 7):**
>
> The proposed loss $\mathcal{L}_U$ is highly effective for **open-set recognition** for the following reasons:
>
> 1. **Explicit Handling of unseen class samples**: Without proper modeling, a model might misclassify unseen samples as one of the known classes. By identifying reliable undesired samples and maintaining a feature bank ($\mathcal{M}_u$), ReDUCe loss ensures that novel samples are correctly treated as distinct from desired classes.
>
> 2. **Feature Space Separation for Open-Set Recognition**: This loss **clusters undesired class samples together** while pushing them away from desired classes. This contrastive approach improves feature consistency among unknown samples and reinforces their separation from the known categories, making it easier to reject out-of-distribution inputs.
>
> 3. **Robust Test-Time Adaptation in Open-Set Scenarios**: By adapting only to **reliable** samples, ReDUCe prevents harmful adaptation to incorrect or ambiguous data. This enhances model robustness in real-world deployment, where domain shifts and novel samples frequently appear, ensuring that the model maintains discriminability between known and unknown categories.

---

> > ### Author Response · Authors · 2025-02-27
> > **Response to Reviewer cGsT (7/7)**
> >
> > >### **Insights on Gradient Analysis**
> >
> > The gradient analysis of the ReDUCe loss provides key insights into how the method enhances open-set recognition by explicitly shaping the feature space. The main takeaways are:
> >
> > 1. **Attractive and Repulsive Forces**: The gradient expressions for both $L_D$ (desired samples) and $L_U$ (undesired samples) reveal a balance of **attraction** towards positive neighbors and **repulsion** from negative samples. This ensures that desired and undesired samples form **well-separated clusters**, reinforcing their discriminability.
> >
> > 2. **Adaptive Hard Negative Mining**: The repulsion force is weighted by the softmax probability $ p(z^-) $, meaning that harder negatives (those more similar to the test sample) exert a stronger push. This **dynamically enhances class separation**, ensuring robust feature space adaptation.
> >
> > 3. **Explicit Open-Set Modeling**: Unlike traditional closed-set test-time adaptation losses, the gradient structure here ensures that **undesired samples are not merely ignored but explicitly modeled**. This prevents the model from collapsing to known classes, making it **better at rejecting novel open-set samples** while maintaining confident classification of known classes.
> >
> > -------
> >
> > >### **Writing Suggestions**
> >
> > Thank you for the suggestions, we will revise the manuscript carefully correcting the mentioned issues.
> >
> > ------
> >
> > [1] Liu, Yuejiang, et al. "TTT++: When Does Self-Supervised Test-Time Training Fail or Thrive?." NeurIPS, 2021.
> >
> > [2] Chen, Dian, et al. "Contrastive Test-Time Adaptation." CVPR, 2022.
> >
> > [3]  Li et al. " On the robustness of open-world test-time training: Self-training with dynamic prototype expansion", ICCV 2023.
> >
> > [4] Z. Gao et al. "Unified Entropy Optimization for Open-Set Test-Time Adaptation", CVPR 2024.
> >
> > [5] Shu et al., " Test-time prompt tuning for zero-shot generalization in vision-language models", NeurIPS 2022.
> >
> > [6] Hassan et asl. "Align Your Prompts: Test-Time Prompting with Distribution Alignment for Zero-Shot Generalization", NeurIPS 2023.
> >
> > [7] Karmanov et al., "Efficient test-time adaptation of vision-language models", CVPR 2024.
> >
> > [8] Zhang et al., "Dual Prototype Evolving for Test-Time Generalization of Vision-Language Models", NeurIPS 2024.
> >
> > [9] Zhao et al., "Tuning layernorm in attention: Towards efficient multi-modal llm finetuning", ICLR 2024.
> >
> > [10] Ting Chen et al., "A simple framework for contrastive learning of visual representations", ICML 2020.
> >
> > [11] Kaiming He et al., "Momentum contrast for unsupervised visual representation learning", CVPR 2020.
> >
> > [12] Schneider et al., "Improving robustness against common corruptions by covariate shift adaptation", NeurIPS 2020.
> >
> > [13] Wang et al., "Tent: Fully Test-Time Adaptation by Entropy Minimization", ICLR 2021.
> >
> > [14] Vaze et al., "Open-set recognition: A good closed-set classifier is all you need", ICLR 2022.
> >
> > [15] Press et al. “RDumb: A simple approach that questions our progress in continual test-time adaptation,” NeurIPS 2023.

---

> > > ### Comment · Reviewer_cGsT · 2025-03-11
> > > **Great rebuttal!**
> > >
> > > I am pleased to report that most of my concerns have been addressed. I want to express my sincere gratitude to the authors. In particular, I wish to acknowledge their considerable efforts in conducting additional experiments and providing clarifications during the rebuttal period. I am confident that our discussion has significantly enhanced the quality of the manuscript.

---

### Author Response · Authors · 2025-02-28
**Manuscript Revision**

Dear Reviewers,

Thank you for the insightful feedback, which has helped us significantly to strengthen our work. We have revised the manuscript to incorporate your writing suggestions and additional experimental analyses done as part of this review process.

Here, we summarize the major changes done in the manuscript:

1. Discussion on **Effect of ReDUCe loss in representation space (Appendix C.3).**
2. **Open Set Single Image CTTA Experiments (Appendix D.3).**
3. **Open-set Continuously Changing Corruptions Benchmark (Appendix D.4).**
4. Additional experiments on **Varying OOD ratio (Appendix D.5).**
5. **Failure case Analysis (Appendix E).**
6. **Broader Impact Concerns (Appendix F).**

As the discussion phase draws to a close, we look forward to addressing any remaining questions or concerns you might have. Thank you once again for your time and valuable input.

---

> ### Author Response · Authors · 2025-03-03
> **Request to review the responses and Manuscript revision**
>
> Dear reviewers,
>
> As the discussion phase ends soon, we request you to review our responses to the concerns raised. We have also revised the manuscript based on the suggestions and included the additional experiments done.

---

### Decision · Action_Editor_Pc78 · 2025-04-17

**Recommendation:** Accept with minor revision

**Comment:**

The paper proposes ROSITA, a test-time adaptation framework composed of a new loss function suitable for open-set test-time adaptation, paired with a study of the optimal parameters to fine-tune at test.

I would like to follow the recommendation of all reviewers to accept the paper, however minor revisions are required to incorporate key results of the author-reviewer discussion phase more prominently into the main paper. Three important reasons for revisions I see are:

- The results from the rebuttal ([see here](https://openreview.net/forum?id=72YVabBErN&noteId=2Fctne0uAa)) are currently not fully worked into the paper and right now only briefly mentioned in page 12, “Appendix”. It would be good if the authors extended their discussion section and commented on the key insights generated during the review/discussion phase.
- Limitations of the benchmark (e.g. using MNIST prominently as distractor samples) and the reply the authors gave on this point during the rebuttal phase should be worked into the main paper. It is also a limitation that the authors did not benchmark the performance of the method in full absence of distractor samples (i.e., in standard TTA settings), which could be made transparent.
- The repository should be updated to reflect the state of the experiments post-rebuttal, to meet the claim of “establishing a comprehensive benchmark for Open-set Single image Test-Time Adaptation”, or the claim should be tuned down.

**Audience:**

The reviewers unanimously agree that the paper targets the TMLR audience, and I agree.

**Claims And Evidence:**

The paper studies an important test-time adaptation setting under dynamical distribution shifts, which is evaluated on a benchmark derived from established datasets used in the field. One reviewer explicitly highlighted that post-rebuttal, the considered benchmarks were appropriately extended.

Two points seem to be important post-discussion:

**Comparison to prior work, benchmark design.** The abstract claims, “[...] making them unsuitable for real-world open-set scenarios”, which is phrased too strongly, and unnecessarily downweights contributions of existing TTA papers. I suggest to tune this sentence a bit, especially given that some of the distractors picked in the paper differ substantially from the test domain (e.g. MNIST vs. Image Datasets as cGsT
 noted), even though this became a benchmark in the field as the [authors noted in their rebuttal](https://openreview.net/forum?id=72YVabBErN&noteId=ImEt718OLA).

In some settings, e.g. CCC, which was added in the rebuttal, the choice of using MNIST is especially debatable as the dataset would allow easily to sample additional domains for the distractor samples that are harder for the model to distinguish, while MNIST induces a very strong distribution shift between the considered dataset and the distractor samples. While the additional experiments were regarded as sufficiently comprehensive, I would ask the authors to update the discussion section of their paper with what they also noted in their rebuttal, i.e., that future work should consider other forms of distractors besides the settings here motivated by previous benchmarks.

**Benchmarking and Evaluation** The abstract further claims, “We address this limitation by establishing a comprehensive benchmark for Open-set Single image Test-Time Adaptation using Vision-Language Models.” which is accurate, but the code (https://github.com/ostta/ROSITA/blob/main/docs/DATASETS.md) should be updated to reflect the new settings updated during the rebuttal, e.g. CCC. I would also encourage the authors to update this README to reflect how models should be evaluated/add pointers to their codebase in this file for their camera ready paper.

As part of the benchmarking effort, reviewers suggested adding techniques like LORA. However, I believe that the claim “However, they are more sensitive to learning rate and optimizer choice” (Appendix C.5) is not sufficiently well supported by experimental data. From Table 14, it is unclear to me how this conclusion can be drawn. The effect is simply that LORA requires a generally lower learning rate, which is acceptable. I would ask the authors to revise/tune down this sentence. The LORA results should also become more prominent in the paper.
Besides these open points that came up during review, the claims made by the paper seem to be well supported after considering the rebuttal.

---

> ### Author Response · Authors · 2025-05-09
> **Official Comment by Authors**
>
> Dear Action Editor and Reviewers,
>
> Thank you for reviewing our paper and providing insightful feedback. The constructive comments provided and the discussion through the rebuttal period helped us improved the paper significantly. We thank your decision towards acceptance of our paper with minor revision. We have uploaded the Camera Ready version, revising the manuscript based on the suggestions provided by the Action Editor. We list the changes made below for your reference:
>
> - Revised abstract.
> - Included discussion on LoRA (Section 4. and Appendix C.5).
> - Included CCC results in the main paper (Table 5.)
> - Updated the limitations (Research Question 9.)
> - Updated code repository to include CCC dataset, the README file as suggested.
>
> Thank you once again for your time and effort in evaluating our work.
>
> Authors

---

> > ### Comment · Action_Editor_Pc78 · 2025-05-09
> >
> > Dear authors, thanks for the edits. The performed updates address my concerns. I think especially the revised abstract and modification to research question 9 better position the work now.